# Measure-Theoretic Anti-Causal Representation Learning

**Arman Behnam**
Department of Computer Secience
Illinois Institute of Technology
Chicago, Illinois, USA
abehnam@hawk.illinoistech.edu

**Binghui Wang**
Department of Computer Secience
Illinois Institute of Technology
Chicago, Illinois, USA
bwang70@illinoistech.edu

## Abstract

Causal representation learning in the anti-causal setting—labels cause features rather than the reverse—presents unique challenges requiring specialized approaches. We propose Anti-Causal Invariant Abstractions (ACIA), a novel measure-theoretic framework for anti-causal representation learning. ACIA employs a two-level design: low-level representations capture how labels generate observations, while high-level representations learn stable causal patterns across environment-specific variations. ACIA addresses key limitations of existing approaches by: (1) accommodating prefect and imperfect interventions through interventional kernels, (2) eliminating dependency on explicit causal structures, (3) handling high-dimensional data effectively, and (4) providing theoretical guarantees for out-of-distribution generalization. Experiments on synthetic and real-world medical datasets demonstrate that ACIA consistently outperforms state-of-the-art methods in both accuracy and invariance metrics. Furthermore, our theoretical results establish tight bounds on performance gaps between training and unseen environments, confirming the efficacy of our approach for robust anti-causal learning. Code is available at https://github.com/ArmanBehnam/ACIA.

## 1 Introduction

Causal representation learning discovers causal relationships underlying data rather than statistical associations [47]. At its core, causal representation learning seeks to identify *high-level causal variables* from low-level observations, bridging the gap between statistical pattern recognition and causal reasoning. Learning these causal variables offers transformative potential for artificial intelligence systems that can reason about cause and effect.

A particularly challenging yet promising domain is learning representations in the *anti-causal* setting, where the causal direction is reversed from traditional prediction tasks. Figure 1 diagrams the anti-causal setting, where $Y$ (target) is the causal variable, $X$ (observation) the observable variables, $E$ (Environment) the environment variable introducing spurious correlations, $U$ is a confounder which affects both $X$ and $Y$, and $Z$ (latent variable) is an unmeasured intermediary. The orange arrows represent direct paths to the observed variables $X$, and blue arrows represent confounding effects.

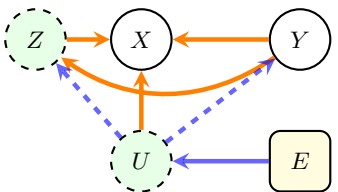

Figure 1: Anti-Causal Diagram: causal (orange), spurious (blue), and confounding (dashed blue) dependencies.

Consider a disease diagnosis from chest X-rays across different hospitals [11]. A disease ($Y$) causes observable symptoms and measurements ($X$), with the relationship represented as $Y \rightarrow X$. The confounding factors $U$ (e.g., age and sex) affect both disease and symptoms. Environmental factors $E$ (e.g., hospital-specific

39th Conference on Neural Information Processing Systems (NeurIPS 2025).

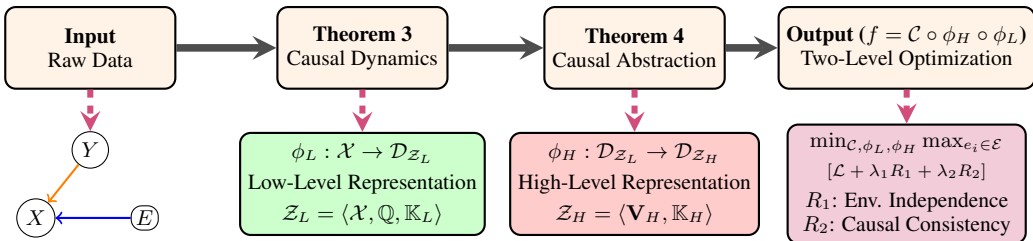

Figure 2: ACIA: Anti-Causal Invariant Abstraction Framework.

protocols) introduce spurious correlations by creating hospital-specific variations, forming the anti-causal structure $Y \to X \leftarrow E$. The orange arrows therefore depict the true disease-to-symptom mechanism ($Y \to X$), while the paths involving $E$ and $U$ introduce noise. This anti-causal structure requires specialized methods to disentangle causal mechanisms from environmental artifacts.

Early works [46, 22] formalized anti-causal learning and showed that traditional methods fail in this setting. Follow-up works can be categorized into three main approaches: *1) Intervention-based causal learning* [59, 10, 35] that models causal effects through interventions; *2) Structure-based causal methods* [61, 21, 33, 49, 50] explicitly models causal structures through Directed Acyclic Graphs (DAGs), requiring complete knowledge of the underlying Structural Causal Model (SCM); *3) Invariant learning methods* [23, 63, 18, 62, 57, 71, 56] that seeks representations invariant across distributions. See more related work in Appendix A.

However, existing methods face several critical limitations. First, intervention-based approaches [59, 10] *assume perfect interventions*—where intervened variables are completely disconnected from their causes—a restrictive assumption rarely satisfied in real-world scenarios. Second, structure-based methods rely on *explicit structural dependencies* through SCM [49, 50] poses significant hurdles when the underlying SCM is unknown. Third, distribution-invariant approaches' assumptions on *independent and identically distributed data or known test distributions* limit their generalization capabilities [71, 56]. Fundamentally, these limitations arise because the single-level representations learned by these methods cannot simultaneously capture the causal mechanism from $Y$ to $X$ while filtering spurious correlations from $E$ to $X$ in anti-causal structure [5, 27].

We develop Anti-Causal Invariant Abstraction (ACIA), a two-level representation learning method (Figure 2), to address above limitations. ACIA is inspired by recent causal representation learning studies [18, 59, 2, 66] and measure-theoretic causality [39]. Specifically, ACIA addresses these limitations through three key innovations: Firs, we introduce a *generalized intervention model that accommodates both perfect and imperfect interventions via the designed interventional kernels*. Second, ACIA *learns directly from raw input without requiring explicit SCM specification*. Third, we introduce *environment-invariant regularizers* that enable stable identification of invariant causal variables across environments. All together, ACIA involves:

- **causal dynamics** to learn *low-level representations* directly from data—without requiring explicit DAGs/SCMs—that capture the anti-causal structure by encoding how labels generate observable features while preserving environment-specific variations. E.g., in medical diagnosis, this reflects how diseases ($Y$) manifest as symptoms and measurements ($X$) in X-ray images, encompassing both disease-related patterns and hospital-specific factors. The learnt low-level representations support reasoning under both perfect and imperfect interventions, enabled by the interventional kernels.

- **causal abstraction** to learn *high-level representations* that distill environment-invariant causal features from the low-level representations. These abstractions generalize across environments by discarding spurious environmental correlations while preserving label-relevant mechanisms. For example, this involves identifying patterns that are consistently associated with specific diseases regardless of the hospital environment.

- **theoretical guarantees** that establish convergence rates, out-of-distribution/domain generalization bounds, and environmental robustness for the learned representations.

We extensively evaluate ACIA on multiple synthetic and real-world datasets with perfect and imperfect interventions. For instance, our results demonstrate ACIA achieves almost perfect accuracy (e.g., 99%) on widely-studied CMNIST and RMNIST synthetic datasets with perfect environment independence and intervention robustness, significantly outperforming SOTA baselines. Most notably, on the real-

world Camelyon17 medical dataset, ACIA achieves 84.40% accuracy—an 19% improvement over the best baseline (65.5% by LECI [18])—while maintaining competitive environment independence and low-level invariance metrics. These results validate our theoretical framework's ability to learn robust anti-causal representations across both synthetic and real-world settings, with particularly strong performance gains in scenarios with complex environmental variations.

## 2 Background on Measure Theory and Causality

A measurable space $(\Omega, \mathscr{F}, \mu)$ consists of a sample space $\Omega$, a $\sigma$-algebra $\mathscr{F}$ of measurable sets, and a probability measure $\mu$. Within the causal context, $\Omega$ represents possible states of the world, $\mathscr{F}$ represents events we can measure, and $\mu$ assigns probabilities to these events. The important notations in this paper are summarized in Table 4 in Appendix.

### 2.1 Measure Theory

**Definition 1** (Environment Measurable Space). *Given a finite set of environments $\mathcal{E}$, each $e \in \mathcal{E}$ is associated with a measurable input space $(\mathcal{X}_e, \mathscr{F}_{\mathcal{X}_e})$, a measurable output space $(\mathcal{Y}_e, \mathscr{F}_{\mathcal{Y}_e})$, and a probability measure $P_e$ on the product space $(\mathcal{X}_e \times \mathcal{Y}_e, \mathscr{F}_{\mathcal{X}_e} \otimes \mathscr{F}_{\mathcal{Y}_e})$.*

**Definition 2** (Data Space). *For each environment $e \in \mathcal{E}$, the data space is a tuple $(D_e, \mathscr{F}_{D_e}, p_e)$ where $D_e = \{(x_j^e, y_j^e)\}_{j=1}^{|D_e|}$ is a finite collection of input-output pairs from environment $e$, $x_j^e$ are elements of the input space $\mathcal{X}_e$, $y_j^e$ are elements of the output space $\mathcal{Y}_e$, $T_e$ is the index set that defines the component-wise sample space structure for environment $e$. Specifically, it indexes the components of the product space such that $\Omega_e = \times_{t \in T_e} E_t$, where each $E_t$ represents a measurable component space at index $t$, and $p_e$ is a probability measure on $D_e$ defining the distribution of $(x_j^e, y_j^e)$.*

**Definition 3** (Representation). *A representation is a measurable function $\phi : \mathcal{X} \to \mathcal{R}$ mapping inputs to a latent space $\mathcal{R}$, where $(\mathcal{R}, \mathscr{F}_{\mathcal{R}})$ is a measurable space.*

A representation is causal if it captures the underlying causal mechanisms generating the data.

**Definition 4** (Kernel [25]). *A kernel $K$ is a function $K : \Omega \times \mathscr{F} \to [0, 1]$ such that: 1. For each fixed $\omega \in \Omega$, the mapping $A \mapsto K(\omega, A)$ is a probability measure on $(\Omega, \mathscr{F})$; 2. For each fixed $A \in \mathscr{F}$, the mapping $\omega \mapsto K(\omega, A)$ is $\mathscr{F}$-measurable.*

Intuitively, $K(\omega, A)$ represents the probability of $A$ conditioned on the information encoded in $\omega$. Properties of kernels being used in this work are discussed in Appendix B.1.

### 2.2 Causality

In the measure-theoretic framework, interventions modify kernels rather than structural equations [40], enabling unified treatment of both perfect and imperfect interventions.

**Definition 5** (Intervention [26]). *An intervention is a measurable mapping $\mathbb{Q}(\cdot|\cdot) : \mathscr{H} \times \Omega \to [0, 1]$ that modifies causal kernels by modifying the underlying probability structure. There are two types of intervention in causal representation learning:*

1. *A* hard (or perfect) intervention *sets $\mathbb{Q}(A|\omega) = \mathbb{Q}(A)$, independent of $\omega$;*

2. *A* soft (or imperfect) intervention *allows $\mathbb{Q}(A|\omega)$ to depend on $\omega$.*

The do-operator $do(X = x)$ represents setting $X$ to value $x$, breaking causal arrows into $X$. Note that $P(Y|do(X = x))$ differs from conditional probability $P(Y|X = x)$, which observes $X = x$ while preserving all causal relationships.

**Definition 6** (Causal Independence [40]). *Variables $X$ and $Y$ are causally independent given $Z$, denoted $X \perp\!\!\!\perp_c Y|Z$, if $P(Y|do(X = x), Z) = P(Y|Z)$ for all $x$ in the support of $X$, and $P(X|do(Y = y), Z) = P(X|Z)$ for all $y$ in the support of $Y$.*

**Definition 7** (Causal Space [39]). *For an environment $e$, a causal space is a tuple $(\Omega_e, \mathscr{H}_e, P_e, K_e)$, where $\Omega_e = \times_{t \in T_e} E_t$ is the sample space, $P_e$ is the probability measure on $(\Omega_e, \mathscr{H}_e)$, and $K_e$ is a kernel function for environment $e$. For each $t \in T_e$, $\mathscr{A}_t$ is the $\sigma$-algebra on component space $E_t$, and the overall $\sigma$-algebra $\mathscr{H}_e = \otimes_{t \in T_e} \mathscr{A}_t$ is the tensor product of these component $\sigma$-algebras.*

This definition is the backbone of measure-theoretic causality in this paper, inspired from [39].

# 3 ACIA: Measure-Theoretic Anti-Causal Representation Learning

This section presents our theoretical framework ACIA for anti-causal representation learning. Our theoretical analysis relies on the following assumptions: The number of training environments $|\mathcal{E}_{\text{train}}| = K < \infty$. For any spurious feature $s$ correlated with $Y$ in $e_i$, there exists $e_j \in \mathcal{E}_{\text{train}}$ where $s \perp\!\!\!\perp Y$. We have access to samples from interventional distributions $P(X|do(Y = y))$ via intervention $\mathbb{Q}$, and representations are $L$-Lipschitz continuous with respect to appropriate metrics.

## 3.1 Problem Formulation

We formalize the anti-causal representation learning problem as follows: given a causal structure where label $Y$ causes the observation $X$ and environment $E$ also influences $X$ ($Y \to X \leftarrow E$), our goal is to learn representations that capture the causal generative invariant from $Y$ to $X$ [1]. Given observations from environments $\mathcal{E} = \{e_i\}_{i=1}^n$ with corresponding datasets $\mathcal{D} = \{D_{e_i}\}_{i=1}^n$, we aim to learn two-level representations:

- a *low-level representation* $\phi_L : \mathcal{D} \subset \mathcal{X} \to \mathcal{Z}_L$ that extracts features from raw data to uncover the anti-causal structure including both $Y \to X$ and $E \to X$.

- a *high-level representation* $\phi_H : \mathcal{Z}_L \to \mathcal{Z}_H$ that distills environment-invariant causal features from $\phi_L$, i.e., enforcing $\phi_H(\phi_L(X)) \perp E \mid Y$.

The predictor $\mathcal{C} : \mathcal{Z}_H \to \mathcal{Y}$ then maps high-level representations to labels. With a loss function $\ell : \mathcal{Y} \times \mathcal{Y} \to \mathbb{R}_+$ defined across all environments $\mathcal{E}$, the full model $f = \mathcal{C} \circ \phi_H \circ \phi_L$ can be trained in an end-to-end fashion.

More specifically, we introduce **causal dynamic** (Thm.3) to facilitate learning low-level representations $\mathcal{Z}_L$ by jointly optimizing the loss with a causal structure consistency regularizer ($R_2$ in Eqn.6), where minimizing it encourages the low-level representations to align with the true causal mechanisms underlying the data. On top of $\mathcal{Z}_L$, we further introduce **causal abstraction** (Thm.4) to learn high-level representations, guided by another environment independence regularizer ($R_1$ in Eqn.7). This regularizer measures the discrepancy between the expected high-level representations across environments conditioned on the label $Y$. Minimizing it can remove environment-specific information while retaining label-relevant causal features.

**Two-level design rationale.** This hierarchical structure enables us to do two procedures : (1) *Interventional effect calculation*: $\phi_L$ captures how the anti-causal setting responds to interventions, handling both perfect and imperfect intervention scenarios; (2) *Information bottleneck*: $\phi_H$ retains only label-relevant invariants while discarding environment-specific noise.

## 3.2 The Theoretical Framework

Our framework establishes product causal space (Def. 8) on measure-theoretic causality to handle anti-causal learning across multiple environments; causal kernel (Def.10) and interventional kernel (Thm.2) to characterize anti-causal structures (Thm.1). Building on this foundation, we develop causal dynamics (Thm.3) to learn low-level representations that extract anti-causal relationships from the raw data. On top of it, we further develop causal abstractions (Thm.4) to learn environment-invariant high-level representations. Figure 2 shows the detailed procedure of our framework. We first introduce necessary definitions below.

**Definition 8** (Product Causal Space). *Given causal spaces $(\Omega_{e_i}, \mathcal{H}_{e_i}, \mathbb{P}_{e_i}, K_{e_i})$ and $(\Omega_{e_j}, \mathcal{H}_{e_j}, \mathbb{P}_{e_j}, K_{e_j})$ for environments $e_i$ and $e_j$, a product causal space is a tuple $(\Omega, \mathcal{H}, \mathbb{P}, \mathbb{K})$ where $\Omega = \Omega_{e_1} \times \Omega_{e_2}$ is the sample space for combined environments $e_i$ and $e_j$, $\mathcal{H} = \mathcal{H}_{e_i} \otimes \mathcal{H}_{e_j}$ is the product $\sigma$-algebra, and $\mathbb{P} = \mathbb{P}_{e_i} \otimes \mathbb{P}_{e_j}$ is the product measure. $\mathbb{K}_S = \{K_S : S \in \mathscr{P}(T)\}$ is a family of causal kernels with $\mathscr{P}$ the power set function and $T = T_{e_i} \cup T_{e_j}$ is the union of index sets.*

This construction enables joint reasoning across environments while preserving individual causal structures. For analysis of environment subsets, we utilize sub-$\sigma$-algebras:

**Definition 9** (Sub-$\sigma$-algebra). *Given a product causal space $(\Omega, \mathcal{H}, \mathbb{P}, \mathbb{K})$, for any subset $S \subseteq T$, the sub-$\sigma$-algebra $\mathcal{H}_S$ is generated by measurable rectangles $A_i \times A_j$ where $A_i \in \mathcal{H}_{e_i}$ and $A_j \in \mathcal{H}_{e_j}$ corresponding to the events in the time indices $S$.*

Product causal spaces merge causal space of different environments by tensor products of $\sigma$-algebras. Properties of causal sub-structures are discussed in Appendix B.2. Next, we define the causal kernel to represent the causal structure.

---

[1] A *causal generative invariant* is a stable function $f : \mathcal{Y} \to \mathcal{X}$ by which $Y$ produces $X$, formally represented as $X = f(Y, \epsilon)$ where $\epsilon$ represents noise, and $f$ is invariant across environments.

**Definition 10** (Causal Kernel). *A causal kernel $K_S \in \mathbb{K}_S$ for index set $S \in \mathscr{P}(T)$ is a function $K_S : \Omega \times \mathscr{H} \to [0,1]$. For fixed $\omega \in \Omega$, $K_S(\omega, \cdot)$ is a probability measure on $(\Omega, \mathscr{H})$, and for fixed $A \in \mathscr{H}$, $K_S(\cdot, A)$ is $\mathscr{H}_S$-measurable, where $\mathscr{H}_S$ is sub-$\sigma$-algebra in $S$.*

Intuitively, $K_S(\omega, A)$ is the conditional probability of event $A$ given causal information encoded in $\omega$, restricted to environments indexed by $S$. This enables characterization of anti-causal structures:

**Theorem 1** (Anti-Causal Kernel Characterization). *For an anti-causal structure with arbitrary feature space $\mathcal{X}$, label space $\mathcal{Y}$, and environments $\mathcal{E}$, the causal kernel satisfies:*

$$K_S(\omega, A) = \int_{\mathcal{Y}} P(X \in A \mid Y = y, E \in S)\, d\mu_Y(y) \tag{1}$$

*where $\mu_Y$ is the marginal measure on $\mathcal{Y}$.*

This characterization captures how labels $Y$ generate observations $X$ across environment subsets $S$, integrating over all possible label values weighted by their marginal probabilities. Realization of anti-causal kernels is discussed in Appendix B.3. Moreover, the independence property of the anti-causal kernel is as follows:

**Corollary 1** (Independence Property of Anti-Causal Kernel). *In an anti-causal structure, for any $\omega, \omega' \in \Omega$ with identical $Y$-component and for all $A \in \mathscr{H}_{\mathcal{X}}$, $B \in \mathscr{H}_Y$, $S \in \mathscr{P}(T)$:*

$$K_S(\omega, \{A|B\}) = K_S(\omega', \{A|B\}) \tag{2}$$

This independence property reveals that conditional kernels depend only on the label $Y$, not on environment-specific information in $\omega$. In Appendix B.4, we prove that causal events show kernel values varying with $\omega$, while anti-causal events maintain invariance under specific subset removals from the conditioning set.

We now characterize how interventions modify causal kernels, enabling unified treatment of both perfect and imperfect interventions.

**Theorem 2** (Interventional Kernel). *Let $(\Omega, \mathscr{H}, \mathbb{P}, \mathbb{K})$ be a product causal space. For any subset $S \in \mathscr{P}(T)$ and intervention $\mathbb{Q} : \mathscr{H} \times \Omega \to [0,1]$, there exists a unique interventional kernel:*

$$K_S^{do(\mathcal{X}, \mathbb{Q})}(\omega, A) = \int_{\Omega} K_S(\omega, d\omega') \mathbb{Q}(A|\omega') \tag{3}$$

*provided that the integral is a Lebesgue integral w.r.t. the measure induced by $K_S(\omega, \cdot)$ on $(\Omega, \mathscr{H})$. In addition, $K_S(\omega, \cdot)$ is $\sigma$-finite for each $\omega \in \Omega$, and $\mathbb{Q}(A|\cdot)$ is $\mathscr{H}$-measurable for each $A \in \mathscr{H}$.*

*We emphasize that this construction encapsulates both intervention types: hard interventions where $\mathbb{Q}(A|\omega') = \mathbb{Q}(A)$ is constant across $\omega'$, and soft interventions where $\mathbb{Q}(A|\omega')$ varies with $\omega'$. Soundness and causal explanation of interventional kernels are discussed in Appendix C.5. The distinctive property of anti-causal structures is that interventions on $X$ do not affect $Y$, while interventions on $Y$ change the distribution of $X$ (asymmetric response to interventions).*

**Corollary 2** (Interventional Kernel Invariance). *In anti-causal structure, interventional kernels satisfy the following invariance criteria:*

*1. $K_S^{do(X)}(\omega, \{Y \in B\}) = K_S(\omega, \{Y \in B\})$ for all measurable sets $B \subseteq \mathcal{Y}$, meaning intervening on $X$ does not change the distribution of $Y$.*

*2. $K_S^{do(Y)}(\omega, \{X \in A\}) \neq K_S(\omega, \{X \in A\})$ for some measurable sets $A \subseteq \mathcal{X}$, meaning intervening on $Y$ changes the distribution of $X$, which is characteristic of an anti-causal relationship.*

Building on the kernel framework, we now develop our approach to learning low-level representations. We adopt the *causal dynamics* perspective [2, 68], which identifies latent causal relationships from observed data under distribution shifts—precisely the setting in anti-causal learning across environments. Our low-level representation mapping $\phi_L$ implements causal dynamics by learning how labels $Y$ generate observations $X$ while preserving environment-specific information. Unlike traditional approaches that immediately pursue invariance, $\phi_L$ intentionally captures both the causal pathway ($Y \to X$) and environmental influences ($E \to X$), providing rich features for subsequent abstraction by $\phi_H$. We formally describe causal dynamic below:

**Theorem 3** (Causal Dynamic and its Kernels). *Given environments $\mathcal{E}$, a causal dynamic $\mathcal{Z}_L = \langle \mathcal{X}, \mathbb{Q}, \mathbb{K}_L \rangle$ can be constructed under the following conditions:*
*1. The product causal space $(\Omega, \mathscr{H}, \mathbb{P}, \mathbb{K})$ is complete and separable.*

2. *The empirical measure $\mathbb{Q}_n$ converges to the true measure $\mathbb{Q}$, i.e., $\sup_{A \in \mathscr{H}} |\mathbb{Q}_n(A) - \mathbb{Q}(A)| \xrightarrow{a.s.} 0$.*
3. *The causal kernel $K_S^{\mathcal{Z}_L}(\omega, A) = \int_{\Omega'} K_S(\omega', A) \, d\mathbb{Q}(\omega')$ exists and is well-defined for all $S \subseteq T$.*

*Further, the causal dynamic kernels are then given by: $\mathbb{K}_L = \{K_S^{\mathcal{Z}_L}(\omega, A) : S \in \mathscr{P}(T), A \in \mathscr{H}\}$.*

This theorem establishes the mathematical foundation for low-level representation learning. The conditions ensure: (1) the underlying probability spaces are well-behaved, (2) finite sample approximations converge to the true distributions, and (3) the integration over empirical data produces valid kernels. The resulting causal dynamic kernels $\mathbb{K}_L$ capture how causal relationships manifest across all possible environment combinations. In addition, the integration over empirical distribution $\mathbb{Q}$ enables unified modeling of both perfect and imperfect interventions. Perfect interventions correspond to point masses in $\mathbb{Q}$, while imperfect interventions use continuous distributions, providing flexibility for real-world scenarios where interventions are rarely perfect, inspired from [2].

We now define the low-level representation mapping $\phi_L : \mathcal{X} \to \mathcal{Z}_L$ with $\mathcal{Z}_L$ established in Thm. 3. We denote $\{\phi_L(\mathcal{X}(\omega_j))\}$ as the collection of low-level representations for a set of samples.

While causal dynamics operate on the raw input space to capture anti-causal relationships, **causal abstraction** further distills these representations into more abstract invariants. Previously, abstraction referred to the process of mapping complex, detailed representations to simpler ones that preserve only the relevant information [17, 7]. In our framework, causal abstraction specifically integrates over the domain of low-level representations to form high-level kernels that capture environment-invariant relationships. This integration serves as an information bottleneck, filtering out environment-specific features while retaining label-relevant causal feature (also demonstrated by our empirical results).

**Theorem 4** (Causal Abstraction and its Kernel). *Let $\mathcal{X}$ be the input space and $\mathscr{H}_\mathcal{X}$ be its $\sigma$-algebra. Assume a measure $\mu$ on the domain of low-level representations $\mathcal{D}_{\mathcal{Z}_L}$. Then, the high-level representation $\mathcal{Z}_H = \langle \mathbf{V}_H, \mathbb{K}_H \rangle$ can be constructed with kernel:*

$$K_S^{\mathcal{Z}_H}(\omega, A) = \int_{\mathcal{D}_{\mathcal{Z}_L}} K_S^{\mathcal{Z}_L}(\omega, A) \, d\mu(z) \tag{4}$$

*The set of high-level causal kernels is then given by: $\mathbb{K}_H = \{K_S^{\mathcal{Z}_H}(\omega, A) : S \in \mathscr{P}(T), A \in \mathscr{H}_\mathcal{X}\}$.*

The high-level representation mapping is defined as $\phi_H : \mathcal{Z}_L \to \mathcal{Z}_H$ with $\mathcal{Z}_H$ established in Thm. 4. We denote $\mathbf{V}_H = \{\phi_H(\phi_L(\mathcal{X}(\omega_j)))\}$ as the resulting high-level representations for a set of samples.

### 3.3 Objective Function of ACIA

ACIA's objective function bases on the theoretical results in Sec.3.2. Specifically, the kernel independence property (Cor.1) motivates the environment independence regularizer $R_1$, while the intervention invariance criteria (Cor.2) guides the design of the causal structure consistency regularizer $R_2$. The optimization achieves the causal dynamics construction (Thm.3) for $\phi_L$ and causal abstraction (Thm.4) for $\phi_H$, ensuring learned representations satisfy the anti-causal structure characterized in Thm.1.

Let $\mathcal{C}$ be a classifier and $\ell$ be a loss function. Our objective function of ACIA is defined as:

$$\min_{\mathcal{C}, \phi_L, \phi_H} \max_{e_i \in \mathcal{E}} \left[ \int_\Omega \ell((\mathcal{C} \circ \phi_H \circ \phi_L)(\mathcal{X}(\omega)), Y(\omega)) \, d\mathbb{P}_{e_i}(\omega) + \lambda_1 R_1 + \lambda_2 R_2 \right] \tag{5}$$

$$R_1 = \sum_{e_i, e_j \in \mathcal{E}, i \neq j} \left\| \int_\mathcal{Y} \int_\Omega \phi_H(\phi_L(\mathcal{X}(\omega))) \, d\mathbb{P}_{e_i}(\omega|y) \, d\mu_Y(y) - \int_\mathcal{Y} \int_\Omega \phi_H(\phi_L(\mathcal{X}(\omega))) \, dP_{e_j}(\omega|y) \, d\mu_Y(y) \right\|_2 \tag{6}$$

$$R_2 = \sum_{e_i \in \mathcal{E}} \left\| \int_\mathcal{Y} y \, d\mathbb{P}_{e_i}(y|\phi_H(\phi_L(\mathcal{X}(\omega)))) - \int_\mathcal{Y} y \, dK_{\{e_i\}}^{do(Y)}(\omega, dy) \right\|_2 \tag{7}$$

*Remark 1:* The minmax formulation in Eqn.5 enforces worst-case robustness across environments. This formulation is supported by our out-of-distribution (OOD) generalization bound (Thm.7). Without it, the learned representations often fail to disentangle environmental factors effectively, as has been validated in prior work on invariant representation learning (IRM[5], Rex[27], VRex[27] ).

*Remark 2:* Our two regularizers $R_1$ and $R_2$ enforce key invariance properties essential for robust anti-causal representation learning[2]:

---

[2] $R_1$ and $R_2$ are also inspired by the invariant representation learning methods such as IRM [5]. Their connections are discussed in Appendix F.3.

- $R_1$ enforces environment independence of representations. It measures the discrepancy between the expected high-level representations across different environments, conditioned on the label $Y$. Minimizing $R_1$ encourages: $\int_\Omega \phi_H(\phi_L(\mathcal{X}(\omega)))\, d\mathbb{P}_{e_i}(\omega|y) \approx \int_\Omega \phi_H(\phi_L(\mathcal{X}(\omega)))\, d\mathbb{P}_{e_j}(\omega|y)$ for all environment pairs $(e_i, e_j)$ and labels $y$, promoting the invariance $\phi_H(\phi_L(\mathcal{X})) \perp\!\!\!\perp E \mid Y$.

- $R_2$ enforces causal structure consistency. It compares the expected value of $Y$ given the high-level representation $\int_\mathcal{Y} y\, d\mathbb{P}_{e_i}(y|\phi_H(\mathcal{X}(\omega)))$ and that of $Y$ under intervention: $\int_\mathcal{Y} y\, dK^{do(Y)}_{\{e_i\}}(\omega, dy)$. Minimizing $R_2$ encourages the representation to be aligned with the true causal structure.

**Theoretical Performance of ACIA:** We also analyze the theoretical performance of ACIA, e.g., convergence property (Thm.6), generalization bound in terms of sample complexity (Thm.7) and interventional kernels (Thm.8) in the anti-causal setting, and environmental robustness (Thm.9) (which shows bounded distributional shifts between training and testing environments, providing robust anti-causal representations). **All proofs are deferred to Appendix D.**

### 3.4 ACIA Algorithm Details

The complete ACIA algorithm include three components (see details of Alg.1-Alg.3).

i) Alg.1 constructs the low-level representation $\phi_L$ by building causal spaces for each environment $e_i \in \mathcal{E}$ and their product spaces, computing causal kernels $K_S$, and ultimately outputting the low-level causal dynamics $\mathcal{Z}_L = \langle \mathcal{X}, \mathbb{Q}, \mathbb{K}_L \rangle$ as established in Thm.3.

ii) Alg.2 takes the set of low-level representations $\phi_L = \{\mathcal{Z}_{L_k}\}_{k=1}^K$ outputted by Alg.1 and constructs the high-level abstraction $\mathcal{Z}_H = \langle \mathbf{V}_H, \mathbb{K}_H \rangle$ by integrating kernels across the low-level representation domain $\mathcal{D}_{\mathcal{Z}_L}$ as derived in Thm.4.

iii) Alg.3 integrates both algorithms by taking the outputs $\phi_L$ and $\phi_H$ as inputs, implementing the core optimization procedure defined in Eqn.5. Particularly, it jointly optimizes both representations while enforcing environment independence through $R_1$ and causal structure consistency through $R_2$.

**Practical Implementation:** $R_1$ is estimated via conditional distribution comparisons across environments, while $R_2$ is approximated through alignment between predicted and interventional distributions. The verification of OOD guarantees (Thms. 7 and 8) involves checking: (i) invariance of $\phi_L$ across environments conditioned on $Y$, (ii) environment independence $\phi_H(\phi_L(X)) \perp E|Y$, and (iii) bounded distributional shifts between training and test environments.

**Computational Complexity:** The objective function in Eqn.5 can be iteratively solved using stochastic gradient with time complexity of $O\left(\frac{nd}{\epsilon^2} \log\left(\frac{1}{\delta}\right)\right)$ and space complexity of $O(|\mathcal{E}|d + d^2)$, where $\epsilon$ is desired precision, $\delta$ is failure probability, and $d$ is dimension of the representation space.

## 4 Experiments

### 4.1 Experimental Setup

**Datasets and Models:** We test four datasets in anti-causal settings: Colored MNIST (CMNIST), Rotated MNIST (RMNIST), Ball Agent [9], and Camelyon17 [6]. In CMNIST and RMNIST, digit labels cause specific image features: colors and rotations (environment), respectively. Ball Agent is a physical simulation environment where ball positions (continuous labels) cause pixel observations, with controlled interventions affecting object dynamics; Camelyon17 is a real medical dataset where tumor presence (label) causes tissue patterns in pathology images, with hospital-specific staining protocols creating environmental variations. These datasets test various aspects of ACIA: discrete vs. continuous labels and perfect vs. imperfect interventions. *Details of (building) these datasets are in Appendix E.1. The model architecture and hyperparameter settings of ACIA are in Appendix E.2.*

**Evaluation Metrics:** We use four metrics to measure predictive performance and causal properties.

1. *Test Accuracy:* Fraction of test samples correctly predicted by our predictor. 2. *Environment Independence (EI):* It measures the degree to which high-level representations remain independent of environment-specific information while preserving label-relevant information. Specifically, we compute mutual information between high-level representations and environment labels, conditioned on class labels. At the end, we weight them by class frequency and calculate their summation. Lower values indicate better environment independence.

3. *Low-level Invariance (LLI or $R_1$):* It quantifies stability of low-level representations across environments. We measure the variance of representations across different environments. At the end, we calculate their average across feature dimensions. Lower values indicate greater invariance.

Table 1: Comparisons with baselines across four datasets and metrics.

| Method | CMNIST | | | | RMNIST | | | | Ball Agent | | | | Camelyon17 | | | |
|---|---|---|---|---|---|---|---|---|---|---|---|---|---|---|---|---|
| | Acc↑ | EI↓ | LLI↓ | IR↓ | Acc↑ | EI↓ | LLI↓ | IR↓ | Acc↑ | EI↓ | LLI↓ | IR↓ | Acc↑ | EI↓ | LLI↓ | IR↓ |
| GDRO[44] | 92.00 | 1.85 | 0.80 | 0.91 | 63.00 | 16.03 | 4.10 | 1.53 | 66.00 | 1.04 | 0.69 | 0.75 | 58.00 | 1.32 | 0.87 | 0.83 |
| MMD[29] | 94.00 | 1.22 | 1.73 | 1.13 | 92.00 | 6.88 | 15.62 | 0.69 | 68.50 | 1.13 | 0.87 | 1.82 | 60.00 | 4.43 | 2.16 | 1.12 |
| CORAL[53] | 89.00 | 1.48 | 2.06 | 1.30 | 91.00 | 4.02 | 9.56 | 0.31 | 70.50 | 1.23 | 1.92 | 1.84 | 41.00 | 1.62 | 2.45 | 1.01 |
| DANN[16] | 45.00 | 0.03 | 0.86 | 0.20 | 38.50 | 12.82 | 3.85 | 1.47 | 61.00 | 1.35 | 0.96 | 0.89 | 39.00 | 0.68 | 1.40 | 1.95 |
| IRM[5] | 85.00 | 1.43 | 0.83 | 1.08 | 85.50 | 19.03 | 6.64 | 3.17 | 56.00 | 0.89 | 0.67 | 1.71 | 52.00 | 1.95 | 1.76 | 2.45 |
| Rex[27] | 73.00 | 0.69 | 1.41 | 1.80 | 80.50 | 0.69 | 10.69 | 0.96 | 54.50 | 1.05 | 0.11 | 7.65 | 39.00 | 0.68 | 1.40 | 1.95 |
| VREx[27] | 95.50 | 1.71 | 1.09 | 0.77 | 93.50 | 2.41 | 2.77 | 1.03 | 74.00 | 0.93 | 0.78 | 0.73 | 54.50 | 1.98 | 1.78 | 1.02 |
| ACTIR[23] | 78.50 | 0.64 | 0.97 | 1.80 | 72.00 | 0.23 | 18.79 | 0.19 | 69.00 | 0.88 | 0.02 | 0.58 | 60.50 | 0.60 | 0.63 | 0.80 |
| CausalDA[62] | 83.50 | 0.41 | 0.85 | 12.23 | 87.50 | 0.62 | 0.91 | 16.44 | 45.50 | 1.20 | 0.85 | 1.22 | 55.50 | 0.55 | 1.60 | 10.55 |
| LECI[18] | 70.00 | 0.83 | 0.40 | 0.67 | 82.00 | 0.29 | 2.91 | 0.04 | 71.20 | **0.46** | 0.39 | 0.05 | 65.50 | **0.23** | 0.50 | 0.45 |
| **ACIA** | **99.20** | **0.00** | **0.01** | **0.02** | **99.10** | **0.00** | **0.03** | **0.01** | **99.98** | *0.52* | **0.03** | **0.03** | **84.40** | *0.28* | **0.42** | **0.43** |

4. *Intervention Robustness (IR or $R_2$):* It evaluates model robustness under interventions by comparing the difference between observational and interventional distributions. Specifically, we first obtain probability confidence scores for original and intervened samples, and then calculate KL divergence between these distributions. Lower values indicate higher robustness.

**Baselines:** We compare AICA against 10 baseline methods spanning three main categories: (1) *Robust optimization methods*: GDRO [44] optimizes the worst-group performance under distribution shifts. (2) *Distribution/Domain-invariant learning*: MMD [29] minimizes distributional distances, CORAL [53] aligns feature correlations, DANN [16] uses adversarial training, IRM [5] enforces invariant predictors, VREx [27] uses risk extrapolation with different variance penalties. (3) *Causal representation learning methods:* CausalDA [62] incorporates causal structure discovery for invariant representation learning. ACTIR [23] specifically targets anti-causal settings, and LECI [18] learns environment-wise causal independence through graph decomposition.

## 4.2 Experimental Results

### 4.2.1 Results under Perfect Intervention

Table 1 comprehensively presents the comparison results of ACIA with existing baselines. These results validate our measure-theoretic framework's ability to capture and exploit anti-causal structures in synthetic and real-world settings. In particular, the results highlight several key findings:

1. **Our ACIA performs the best and significantly outperforms baselines.** For instance, on CMNIST and RMNIST, ACIA achieves an accuracy of 99.00%+, perfect environment independence (0.00), almost perfect interventional robustness (0.02 and 0.01) and low-level invariance (0.01 and 0.03), significantly surpassing others baseline. On Ball Agent, our ACIA achieves 99.72% accuracy, and almost perfect low-level invariance and interventional robustness. On the real-world Camelyon17, ACIA achieves the best test accuracy 87.00% and retains the underlying causal properties.

2. **Causal dynamic construction (Theorem 3) is confirmed** by the low-level invariance in our results. This matches the theoretical expectation of environment-independent feature learning.

3. **Interventional kernel invariance (Corollary 2) is empirically validated** via the intervention robustness score, implying the distinction between observational and interventional distributions.

4. **Anti-Causal OOD generalization bound (Theorem 7 in Appendix) is substantiated** by the test accuracy improvements over the compared baselines across all datasets.

### 4.2.2 Results under Imperfect Intervention

Perfect/Hard intervention completely disconnects intervened variables from their causes, while imperfect/soft intervention modifies causal mechanisms and maintain partial original dependencies. This experiment aims to validate the effectiveness of our ACIA against imperfect intervention on the studied datasets, with details of constructing imperfect intervention discussed in Appendix E.3.

Table 2 shows the results. We can see that imperfect intervention achieves similar results on the four datasets and metrics as perfect intervention. This supports our theoretical claim that ACIA can effectively handle both perfect and imperfect interventions via its interventional kernel formulation.

### 4.2.3 Visualizing Learnt Representations

**Results on CMNIST:** Figure 3 demonstrates ACIA's ability to learn representations that perfectly capture the anti-causal structure and predict the test data. *(1) First panel:* Low-level representations

Table 2: ACIA performance under imperfect intervention.

| Dataset | CMNIST | | | | RMNIST | | | | Ball Agent | | | | Camelyon17 | | | |
|---|---|---|---|---|---|---|---|---|---|---|---|---|---|---|---|---|
| Metric | Acc | EI | LLI | IR | Acc | EI | LLI | IR | Acc | EI | LLI | IR | Acc | EI | LLI | IR |
| Value | 99.4 | 0.00 | 0.01 | 0.03 | 99.0 | 0.01 | 0.03 | 0.01 | 99.7 | 0.44 | 0.06 | 0.06 | 84.4 | 0.30 | 0.44 | 0.45 |

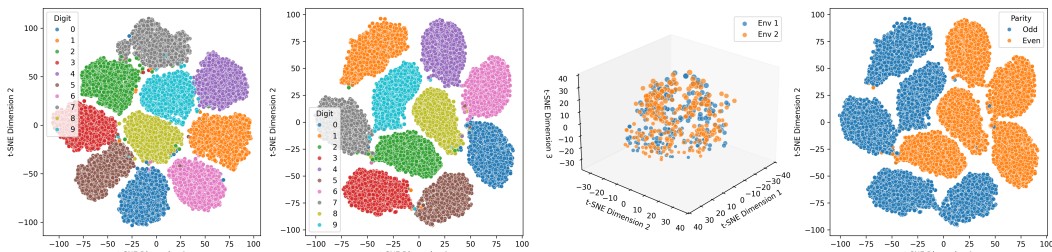

Figure 3: t-SNE visualization of ACIA representations on CMNIST. From left to right: (1) Low-level representations show initial digit clustering with color influence; (2) High-level representations show improved digit separation; (3) Environment visualization demonstrate removal of environment-specific information; (4) Parity analysis reveals clear separation between even and odd digits.

show clear digit-based clustering while retaining certain environment information (note that some colored images are mapped to the digit cluster that they are not belonging to); *2) Second panel:* High-level representations improve digit cluster separation with clearer boundaries; *3) Third panel:* The environment visualization displays colored images from different environments are mixed, confirming the removal of environment-specific information; and *4) Fourth panel:* The parity visualization reveals how ACIA organizes digits based on their mathematical properties—the alternating pattern between even digits (orange) and odd digits (blue) confirms ACIA preserves meaningful numerical relationships while eliminating spurious color correlations. This organization aligns with findings from [24] showing that neural networks capture abstract number properties beyond visual features.

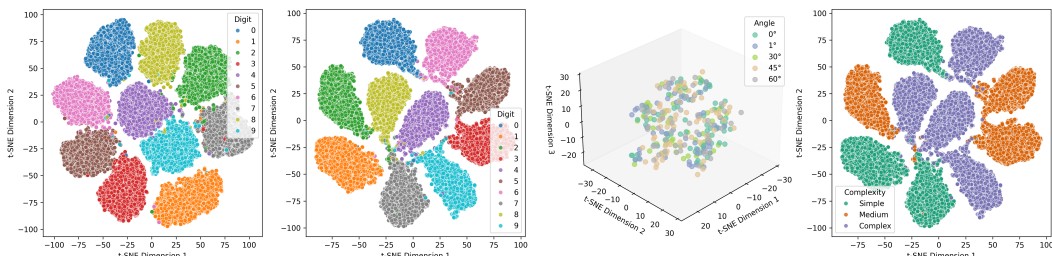

Figure 4: t-SNE visualization of ACIA representations on RMNIST. From left to right: (1) Low-level representations show digit clustering but with rotation influence; (2) High-level representations with better digit boundaries; (3) Rotation angle visualization shows uniform distribution across the representation space; (4) Digit complexity reveals semantic organization by structural properties.

**Results on RMNIST:** See Figure 4. Similarly, (1) Low-level representations show clear digit-based clustering but keep certain rotation information; (2) High-level representations with more distinct boundaries; (3) The rotation angle visualization displays uniform coloring across the entire representation space, confirming successful abstraction of rotation-specific information; and (4) The digit complexity visualization reveals semantic organization where digits with similar structural properties cluster together. *Simple (0,1,7)*: minimal stroke count (typically 1-2), more rotation-invariant features, and lower topological complexity; *Medium (2,3,5)*: moderate stroke count (typically 2-3), mixed curves/lines, and intermediate visual density; and *Complex (4,6,8,9)*: most stroke count (3+), multiple curves/intersections, and higher topological complexity [65].

**Results on Ball Agent:** See Figure 5. (1) Low-level representations display position-based organization with considerable mixing between position values; (2) High-level representations show more pronounced position-based clustering with clearer boundaries, demonstrating improved abstraction of spatial information; (3) The intervention visualization displays a categorical distribution of intervention patterns (None, Single, Double, Multiple); *their details are shown in Appendix E.1.3*), revealing

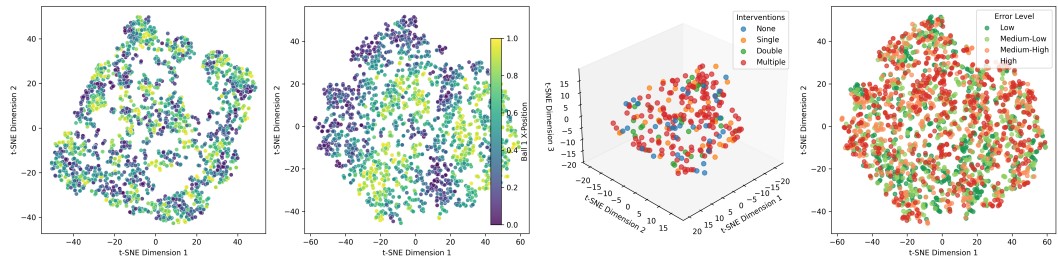

Figure 5: t-SNE visualization of ACIA representations on Ball Agent. From left to right: (1) Low-level representations display position-based organization with environmental mixing, (2) High-level representations show more pronounced position clustering; (3) Intervention visualization categorized by intervention patterns, and (4) Prediction error shows areas of high accuracy (green) versus areas requiring improvement (red).

how different intervention types affect the latent space structure; and (4) The prediction error visualization shows areas of high accuracy (green) versus areas requiring improvement (red), confirming that position-relevant information is preserved while achieving partial invariance to interventions.

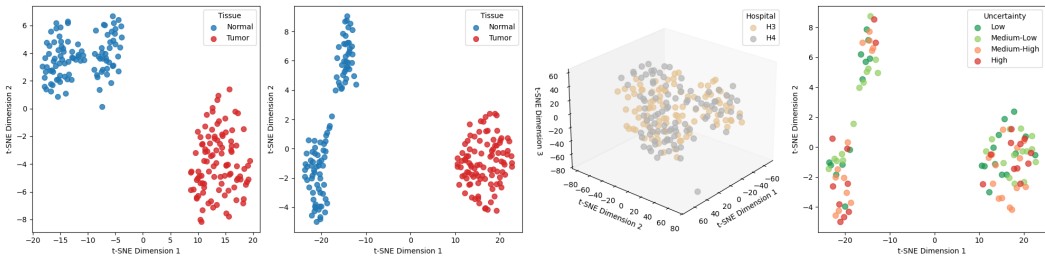

Figure 6: t-SNE visualization of ACIA representations on Camelyon17. From left to right: (1) Low-level representations show partial tumor/normal tissue separation, (2) High-level representations with improved class boundaries, (3) Hospital visualization demonstrates mixing of environment-specific features, and (4) Uncertainty analysis highlights regions of high confidence (green/yellow) versus regions requiring more evidence (red).

**Results on Camelyon17:** In Figure 6, (1) Low-level representations show separation between tumor and normal tissue samples, but with certain mixing; (2) High-level representations demonstrate more pronounced clustering with clearer boundaries between tissue types, particularly visible in the left-right separation; (3) The hospital visualization displays significant mixing between hospital sources despite their different staining protocols, confirming reduction of environment-specific information; and (4) The uncertainty visualization highlights regions where the model maintains high confidence (green/yellow) versus areas requiring more evidence (red). More details are in Appendix E.1.4.

## 5 Conclusion

We presented ACIA, a measure-theoretic framework for anti-causal representation learning. ACIA provides: (1) a unified interventional kernel formulation that accommodates both perfect and imperfect interventions without requiring explicit causal structure knowledge; (2) a novel causal dynamic that captures anti-causal structure from raw observations, together with a causal abstraction that distills environment-invariant relationships; (3) a principled optimization framework based on a min–max objective with causal regularizers; and (4) provable out-of-distribution generalization guarantees that bound the performance gap between training and unseen environments. Overall, ACIA opens new directions for causal representation learning in settings where traditional assumptions do not hold. In future, we plan to generalize ACIA to handle more complex causal structures (such as confounded-descendant or mixed causal-anticausal scenarios [63]).

## Acknowledgments

We thank the anonymous reviewers for their valuable and constructive feedback. This work was supported in part by the Cisco Research Award and by the National Science Foundation under Grant Nos. ECCS-2216926, CCF-2331302, CNS-2241713, and CNS-2339686.

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

# Appendix

Table 3: Summarizing Causal and Non-Causal Invariant Representation Learning Methods

| Method | Anti-causal Structure | SCM Requirements | Imperfect Interventions | Intervention Inference | Nonparametric | High-dim Data | OOD |
|---|---|---|---|---|---|---|---|
| Distribution/Domain-invariant Learning | | | | | | | |
| (C-)ADA [31] | ✗ | ✗ | ✗ | ✗ | ✗ | ✓ | ✗ |
| Domain adaptation [54] | ✗ | ✗ | ✗ | ✗ | ✗ | ✓ | ✓ |
| DDAIG [75] | ✗ | ✗ | ✗ | ✗ | ✗ | ✓ | ✓ |
| L2A-OT [76] | ✗ | ✗ | ✗ | ✗ | ✗ | ✓ | ✓ |
| ERM [56] | ✗ | ✗ | ✗ | ✗ | ✗ | ✓ | ✗ |
| DOMAINBED [19] | ✗ | ✗ | ✗ | ✗ | ✗ | ✓ | ✓ |
| StableNet [73] | ✗ | ✗ | ✗ | ✗ | ✗ | ✓ | ✓ |
| SagNets [37] | ✗ | ✗ | ✗ | ✗ | ✗ | ✓ | ✓ |
| SWAD [12] | ✗ | ✗ | ✗ | ✗ | ✗ | ✓ | ✗ |
| FACT [67] | ✗ | ✗ | ✗ | ✗ | ✓ | ✓ | ✓ |
| Evaluation Protocol [71] | ✗ | ✗ | ✗ | ✗ | ✗ | ✓ | ✓ |
| Ratatouille [43] | ✗ | ✗ | ✗ | ✗ | ✗ | ✓ | ✓ |
| XRM [42] | ✗ | ✗ | ✗ | ✗ | ✓ | ✓ | ✗ |
| FeAT [13] | ✗ | ✗ | ✗ | ✗ | ✓ | ✓ | ✓ |
| AIA [52] | ✗ | ✗ | ✗ | ✗ | ✓ | ✓ | ✓ |
| IRM [5] | ✗ | ✗ | ✗ | ✗ | ✗ | ✓ | ✗ |
| Rex [27] | ✗ | ✗ | ✗ | ✗ | ✓ | ✓ | ✓ |
| CI to Spurious [57] | ✗ | ✗ | ✗ | ✗ | ✓ | ✓ | ✓ |
| Information Bottleneck [1] | ✗ | ✗ | ✗ | ✗ | ✓ | ✓ | ✓ |
| CausalDA [62] | ✓ | ✗ | ✗ | ✗ | ✓ | ✓ | ✓ |
| Transportable Rep [23] | ✓ | ✗ | ✗ | ✗ | ✓ | ✓ | ✓ |
| Structure-based Causal Representation Learning | | | | | | | |
| DISRL [62] | ✗ | ✓ | ✗ | ✓ | ✓ | ✓ | ✓ |
| Causal Disentanglement [50] | ✗ | ✓ | ✗ | ✗ | ✗ | ✓ | ✓ |
| ICP [41] | ✗ | ✓ | ✗ | ✗ | ✓ | ✗ | ✗ |
| ICP for nonlinear [20] | ✗ | ✓ | ✗ | ✗ | ✓ | ✗ | ✓ |
| Active ICP [15] | ✗ | ✓ | ✗ | ✗ | ✓ | ✗ | ✓ |
| CSG [30] | ✗ | ✓ | ✗ | ✗ | ✓ | ✓ | ✓ |
| LECI [18] | ✗ | ✓ | ✓ | ✗ | ✓ | ✓ | ✓ |
| KCDC [36] | ✗ | ✓ | ✓ | ✗ | ✓ | ✓ | ✓ |
| Separation & Risk [34] | ✗ | ✓ | ✗ | ✗ | ✓ | ✓ | ✓ |
| Intervention-based Causal Learning | | | | | | | |
| Nonparametric ICR [59] | ✗ | ✓ | ✓ | ✓ | ✓ | ✓ | ✓ |
| General Nonlinear Mixing [10] | ✗ | ✓ | ✓ | ✗ | ✓ | ✓ | ✓ |
| Weakly supervised [9] | ✗ | ✓ | ✓ | ✓ | ✓ | ✓ | ✓ |
| iCaRL [32] | ✗ | ✓ | ✗ | ✓ | ✓ | ✓ | ✓ |
| CIRL [33] | ✗ | ✓ | ✓ | ✓ | ✓ | ✓ | ✓ |
| ICRL [2] | ✗ | ✓ | ✗ | ✗ | ✓ | ✓ | ✓ |
| LCA [48] | ✗ | ✓ | ✓ | ✓ | ✗ | ✓ | ✓ |
| ICA [64] | ✗ | ✓ | ✓ | ✓ | ✗ | ✓ | ✓ |
| AIT [45] | ✗ | ✓ | ✓ | ✓ | ✗ | ✓ | ✓ |
| **ACIA** | ✓ | ✗ | ✓ | ✓ | ✓ | ✓ | ✓ |

# A  More Related Work

In Table 3, we broadly summarize causal and non-causal invariant representation learning methods.

## A.1  Distribution/Domain-invariant Learning

Early non-causal methods like (C-)ADA [31] and DDAIG [75] focused on domain adaptation strategies, while more recent works such as FeAT [13] and AIA [52] have developed sophisticated objective functions for distribution shift robustness. Domain adaptation methods [54], L2A-OT [76], ERM [56], DOMAINBED [19], Adversarial and Pre-training [70], StableNet [73], SagNets [37], SWAD [12], Theoretical Framework [69], FACT [67], Evaluation Protocol [71], Ratatouille [43], XRM [42], IRM [5], Rex [27], CI to Spurious [57], Information Bottleneck [1], CausalDA [62], and Transportable Rep [23] also fall under this category as they aim to learn representations invariant across different distributions or domains.

## A.2  Intervention-based Causal Learning

Foundational works in the causal domain include Nonparametric ICR [59] which jointly learns encoders and intervention targets with SCM-based structures and General Nonlinear Mixing [10] which addresses non-linear relationships in latent spaces, which model causal effects through interventions. Weakly supervised methods [9], iCaRL [32], CIRL [33], and LCA [48] also leverage interventions for learning causal representations. ICRL [2] also falls under this category. Weak distributional

| Name | Symbol | Name | Symbol |
|---|---|---|---|
| Environment | $e_i$ | Product sample space | $\Omega = \Omega_{e_i} \times \Omega_{e_j}$ |
| Set of environments | $\mathcal{E}$ | Product $\sigma$-algebra | $\mathscr{H} = \mathscr{H}_{e_i} \otimes \mathscr{H}_{e_j}$ |
| Environment sample space | $\Omega_{e_i}$ | Product probability measure | $\mathbb{P} = \mathbb{P}_{e_i} \otimes P_{e_j}$ |
| $\sigma$-algebra on $\Omega_{e_i}$ | $\mathscr{H}_{e_i}$ | Product causal kernel family | $\mathbb{K} = \{K_S : S \in \mathscr{P}(T)\}$ |
| Probability measure | $\mathbb{P}_{e_i}$ | Input space | $\mathcal{X}$ |
| Causal kernel | $K_{e_i}$ | Low-level latent space domain | $\mathcal{D}_{\mathbb{Z}_L}$ |
| Environment causal space | $(\Omega_{e_i}, \mathscr{H}_{e_i}, \mathbb{P}_{e_i}, K_{e_i})$ | High-level latent space | $\mathcal{D}_{Z_H}$ |
| Causal product space | $(\Omega, \mathscr{H}, \mathbb{P}, \mathbb{K})$ | Label space | $\mathcal{Y}$ |
| Sub-$\sigma$-algebra | $\mathscr{H}_S$ | Low-level representation | $\phi_L : \mathcal{X} \to \mathcal{D}_{\mathbb{Z}_L}$ |
| Index set | $T = T_{e_i} \cup T_{e_j}$ | High-level representation | $\phi_H : \mathcal{D}_{\mathbb{Z}_L} \to \mathcal{D}_{Z_H}$ |
| Interventional kernel | $K_S^{do(\mathcal{X},\mathbb{Q})}(\omega, A)$ | Predictor | $\mathcal{C} : \mathcal{D}_{Z_H} \to \mathcal{Y}$ |
| Intervention measure | $\mathbb{Q}(\cdot\|\cdot)$ | Full predictive model | $f = \mathcal{C} \circ \phi_H \circ \phi_L$ |
| Marginal measure on $\mathcal{Y}$ | $\mu_Y$ | Loss function | $\ell : \mathcal{Y} \times \mathcal{Y} \to \mathbb{R}_+$ |
| Causal dynamic | $\mathcal{Z}_L = \langle \mathcal{X}, \mathbb{Q}, \mathbb{K}_L \rangle$ | Environment independence reg. | $R_1$ |
| Causal abstraction | $\mathcal{Z}_H = \langle \mathbf{V}_H, \mathbb{K}_H \rangle$ | Causal structure alignment reg. | $R_2$ |
| Set of low-level kernels | $\mathbb{K}_L = \{K_S^{\mathcal{Z}_L}(\omega, A)\}$ | Regularization parameters | $\lambda_1, \lambda_2$ |
| Set of high-level kernels | $\mathbb{K}_H = \{K_S^{\mathcal{Z}_H}(\omega, A)\}$ | Conditional mutual information | $I(X; E = e \mid Y)$ |

Table 4: Key Notations in Anti-Causal Representation Learning Framework

Table 5: Comparison of prior information requirements across causal representation learning methods

| Method | Causal Structure | SCM Knowledge | Intervention Type |
|---|---|---|---|
| ACTIR[23] | Anti-causal $Y \to X \leftarrow E$ | Variable roles only | Perfect only |
| CausalDA[62] | DAG structure | Variable types | Perfect only |
| LECI[18] | Partial connectivity | Variable relationships | Perfect only |
| **ACIA** | **Anti-causal $Y \to X \leftarrow E$** | **Variable roles only** | **Both perfect and imperfect** |

invariances [3] considers perfect interventions for single-node, and addresses multi-node imperfect interventions by identifying latent variables whose distributional properties remain stable Independent Component Analysis (ICA) [64] focus on unsupervised identification of latent causal variables through component analysis, and operate the disentanglement through taxonomic distance measures and graph-based analysis. AIT [45] builds on SCMs with explicit DAG assumptions, and primarily focuses on standard causal direction.

## A.3 Structure-based Causal Representation Learning

Methods like DISRL [62] and Causal Disentanglement [50] explicitly model causal structures. Foundational works like ICP [41] and its nonlinear extension [20], as well as CSG [30] and LECI [18] which focuses on identifying causal subgraphs while removing spurious correlations, also incorporate causal structure but often require explicit Directed Acyclic Graphs (DAGs) or focus on identifying causal subgraphs. KCDC [36] employs kernel methods primarily for causal discovery and orientation, and focuses on statistical independence tests through kernel measures. Anti-causal separation and risk invariance [34] inputs are generated as functions of target labels and protected attributes. They use conventional causal modeling with DAGs and do-calculus.

## A.4 Comparison of Prior Information Requirements

Table 5 compares the prior knowledge requirements across state-of-the-art causal representation learning methods. Our results demonstrate that despite requiring less prior information, ACIA outperforms these methods.

- "Variable roles only" refers to knowing which variable is the target ($Y$), observations ($X$), and environmental factors ($E$).

- "Variable types" means knowing the data type of each variable, such as whether it is binary, categorical, or drawn from a specific noise distribution.

- "Variable relationship" refers to knowing the causal structure (i.e., the directionality in the causal DAG) among the variables.

# B  Properties

## B.1  Properties of Kernel

For each $S \in \mathscr{P}(T)$, the kernel $K_S : \Omega \times \mathscr{H} \to [0, 1]$ extends from the component kernels as follows:

1. For measurable rectangles $A_i \times A_j$ with $A_i \in \mathscr{H}_{e_i}$ and $A_j \in \mathscr{H}_{e_j}$, and $\omega = (\omega_i, \omega_j) \in \Omega$:

$$K_S(\omega, A_i \times A_j) = K_{e_i}(\omega_i, A_i) \cdot K_{e_j}(\omega_j, A_j) \tag{8}$$

2. For general measurable sets $A \in \mathscr{H}$, by the Carathéodory extension theorem:

$$K_S(\omega, A) = \mathbb{E}[\mathbf{1}_A \mid \mathscr{H}_S](\omega) \tag{9}$$

where $\mathscr{H}_S$ is the sub-$\sigma$-algebra corresponding to indices in $S$.

## B.2  Properties of Product Causal Space Sub-$\sigma$-algebra

Intuition: Sub-$\sigma$-algebras capture partial information from subsets of environments. These properties ensure our hierarchical structure is well-behaved and consistent across different environment combinations. The following propositions formalize these characteristics.

**Proposition 1** (Properties of Sub-$\sigma$-algebras). *Let $(\Omega, \mathscr{H}, \mathbb{P}, \mathbb{K})$ be a product causal space and $\mathscr{H}_S$ be a sub-$\sigma$-algebra for $S \subseteq T$. Then: (i) $\mathscr{H}_S \subseteq \mathscr{H}$ for all $S \subseteq T$ (ii) If $S_1 \subseteq S_2 \subseteq T$, then $\mathscr{H}_{S_1} \subseteq \mathscr{H}_{S_2}$ (iii) $\mathscr{H}_T = \mathscr{H}$*

*Proof.* We prove each statement separately:

(i) By construction, $\mathscr{H}_S$ is generated by measurable rectangles $A_i \times A_j$ where $A_i \in \mathscr{H}_{e_i}$ and $A_j \in \mathscr{H}_{e_j}$ corresponding to events in the time indices $S$. Since $\mathscr{H} = \mathscr{H}_{e_i} \otimes \mathscr{H}_{e_j}$ is the product $\sigma$-algebra that contains all measurable rectangles, we have $\mathscr{H}_S \subseteq \mathscr{H}$ by definition.

(ii) Let $S_1 \subseteq S_2 \subseteq T$. Any measurable rectangle generating $\mathscr{H}_{S_1}$ corresponds to events in time indices from $S_1$. Since $S_1 \subseteq S_2$, these same rectangles are also in the generating set of $\mathscr{H}_{S_2}$. By the minimality property of $\sigma$-algebras, $\mathscr{H}_{S_1} \subseteq \mathscr{H}_{S_2}$.

(iii) When $S = T$, the generating rectangles of $\mathscr{H}_S$ include all possible measurable rectangles from $\mathscr{H}_{e_i}$ and $\mathscr{H}_{e_j}$ that can be formed from the complete set of time indices. These rectangles generate $\mathscr{H} = \mathscr{H}_{e_i} \otimes \mathscr{H}_{e_j}$, so $\mathscr{H}_T = \mathscr{H}$. $\qquad\square$

**Proposition 2** (Probability Measure Restriction). *For any $S \subseteq T$, the restriction $\mathbb{P}|_{\mathscr{H}_S}$ of the product probability measure to $\mathscr{H}_s$ is a well-defined probability measure, and for $S_1 \subseteq S_2 \subseteq T$:*

$$\mathbb{P}|_{\mathscr{H}_{S_2}}(A) = \mathbb{P}|_{\mathscr{H}_{S_1}}(A) \text{ for all } A \in \mathscr{H}_{S_1}$$

*Proof.* First, we establish that $\mathbb{P}|_{\mathscr{H}_S}$ is a well-defined probability measure. $\mathscr{H}_S$ is a $\sigma$-algebra by construction, $\mathbb{P}|_{\mathscr{H}_S}(A) = \mathbb{P}(A)$ for all $A \in \mathscr{H}_S$, $\mathbb{P}|_{\mathscr{H}_S}(\Omega) = \mathbb{P}(\Omega) = 1$, and $\mathbb{P}|_{\mathscr{H}_S}$ inherits countable additivity from $\mathbb{P}$.

To prove the consistency of the restrictions for $S_1 \subseteq S_2 \subseteq T$, let $A \in \mathscr{H}_{S_1}$. Since $S_1 \subseteq S_2$, by Proposition 1(ii), we have $A \in \mathscr{H}_{S_2}$.

For any measurable rectangle $A = A_i \times A_j$ where $A_i \in \mathscr{H}_{e_i}$ and $A_j \in \mathscr{H}_{e_j}$ corresponding to events in time indices $S_1$:

$$\mathbb{P}|_{\mathscr{H}_{S_1}}(A) = \mathbb{P}(A) = \mathbb{P}_{e_i}(A_i) \cdot \mathbb{P}_{e_j}(A_j) \quad \text{(by definition of product measure)} = \mathbb{P}|_{\mathscr{H}_{S_2}}(A)$$

This equality extends to all sets in $\mathscr{H}_{S_1}$ by the uniqueness of measure extension. Therefore:

$$\mathbb{P}|_{\mathscr{H}_{S_2}}(A) = \mathbb{P}|_{\mathscr{H}_{S_1}}(A) \text{ for all } A \in \mathscr{H}_{S_1}$$

$\qquad\square$

**Proposition 3** (Monotonicity of Information). *For $S_1 \subseteq S_2 \subseteq T$ and any $\mathscr{H}$-measurable random variable $X$:*

$$\mathbb{E}[\mathbb{E}[X|\mathscr{H}_{S_2}]|\mathscr{H}_{S_1}] = \mathbb{E}[X|\mathscr{H}_{S_1}]$$

*Proof.* Let $S_1 \subseteq S_2 \subseteq T$ and let $X$ be any $\mathscr{H}$-measurable random variable. By Proposition 1(ii), we have $\mathscr{H}_{S_1} \subseteq \mathscr{H}_{S_2}$. This nested relationship between the sub-$\sigma$-algebras is crucial for applying the tower property of conditional expectation. By the tower property of conditional expectation, for nested $\sigma$-algebras $\mathscr{G}_1 \subseteq \mathscr{G}_2 \subseteq \mathscr{F}$, we have:

$$\mathbb{E}[\mathbb{E}[X|\mathscr{G}_2]|\mathscr{G}_1] = \mathbb{E}[X|\mathscr{G}_1]$$

Applying this to our sub-$\sigma$-algebras $\mathscr{H}_{S_1} \subseteq \mathscr{H}_{S_2} \subseteq \mathscr{H}$:

$$\mathbb{E}[\mathbb{E}[X|\mathscr{H}_{S_2}]|\mathscr{H}_{S_1}] = \mathbb{E}[X|\mathscr{H}_{S_1}]$$

This result has an information-theoretic interpretation: conditioning on a larger $\sigma$-algebra ($\mathscr{H}_{S_2}$) provides more refined information than conditioning on a smaller one ($\mathscr{H}_{S_1}$). The tower property shows that the expected value of this refined information, when further conditioned on the smaller $\sigma$-algebra, equals the direct conditioning on the smaller $\sigma$-algebra. □

### B.3 Properties of Anti-Causal Kernels

Based on Theorem.1, for a more precise characterization of the causal kernel that underlies these event properties, we provide the following remark relating kernels to conditional probabilities in different settings.

**Remark 1** (Characterization of Causal Kernels). *For a causal space $(\Omega, \mathscr{H}, \mathbb{P}, \mathbb{K})$, the relationship between causal kernels and conditional probabilities is characterized as follows:*

*1. For any $S \in \mathscr{P}(T)$, the causal kernel induces conditional probabilities:*

$$K_S(\omega, A) = \mathbb{E}[\mathbf{1}_A \mid \mathscr{H}_S](\omega)$$

*where $\mathbf{1}_A$ is the indicator function of set $A$.*

*2. Regular conditional probabilities (i.e., versions that are measurable in $\omega$) arise as a special case:*

$$K_S(\omega, A) = \mathbb{P}(A \mid \mathscr{H}_S)(\omega)$$

*3. In anti-causal structures:*

$$K_S(\omega, A) = \mathbb{P}(A \mid Y = y, E \in S)$$

*where $y$ is the $Y$-component of $\omega$.*

### B.4 Properties of Causal and Anti-Causal Events

We can understand the causal and anti-causal events kernel differences as follows:

**Proposition 4** (Causal and Anti-Causal Event Properties). *For a product causal space with kernel $K_S$ and any measurable event $A \in \mathscr{H}$:*

*1. For a causal event $A$: $K_S(\omega, A) \neq \mathbb{P}(A)$ for all $\omega \in \Omega$*

*2. For an anti-causal event $A$: $K_S(\omega, A) = K_{S\setminus U}(\omega, A)$ for all $\omega \in \Omega$*

*where $U \subseteq S \in \mathscr{P}(T)$. Based on Def.5, in our settings, an intervention is a measurable mapping $\mathbb{Q}(\cdot|\cdot) : \mathscr{H} \times \Omega \to [0,1]$. Hard intervention is $\mathbb{Q}(A|\omega') = P(X \in A|do(Y = y'))$, and soft intervention is $\mathbb{Q}(A|\omega') = P(X \in A|Y = y', E \in S)$ where $y'$ denotes the $Y$-component of $\omega'$.*

*Proof.* We establish the distinct properties of causal and anti-causal events through their behavior under the causal kernel.

**Part 1: Causal events**

Let $A$ be causally dependent on variables in $\mathscr{H}_S$. By definition of causal dependence, there exist $\omega, \omega' \in \Omega$ such that $K_S(\omega, A) \neq K_S(\omega', A)$. Since $\mathbb{P}(A) = \int_\Omega K_S(\omega, A)\, d\mathbb{P}(\omega)$ is a fixed constant, we cannot have $K_S(\omega, A) = \mathbb{P}(A)$ for all $\omega \in \Omega$. Therefore $K_S(\omega, A) \neq \mathbb{P}(A)$ for some $\omega$.

Therefore, $K_S(\omega, A) \neq \mathbb{P}(A)$ for some $\omega \in \Omega$, which is consistent with the causal structure of $A$. Since $\mathbb{P}(A) = \int_\Omega K_S(\omega, A)\, d\mathbb{P}(\omega)$ is a weighted average of $K_S(\omega, A)$ over all $\omega$, we cannot have $K_S(\omega, A) = \mathbb{P}(A)$ for all $\omega$.

**Part 2: Anti-causal events**

Let $A \in \mathscr{H}$ be an anti-causal event, and let $U \subseteq S \in \mathscr{P}(T)$. We need to show that $K_S(\omega, A) = K_{S \setminus U}(\omega, A)$ for all $\omega \in \Omega$. By definition, an anti-causal event is one whose probability is invariant to certain interventions. Specifically, removing a subset $U$ from the conditioning information $S$ does not change the kernel's value if $A$ is anti-causal with respect to $U$.

From the definition of causal kernels:

$$K_S(\omega, A) = \mathbb{E}[\mathbf{1}_A \mid \mathscr{H}_S](\omega), \quad K_{S \setminus U}(\omega, A) = \mathbb{E}[\mathbf{1}_A \mid \mathscr{H}_{S \setminus U}](\omega)$$

For an anti-causal event $A$, the information in $\mathscr{H}_U$ (corresponding to indices in $U$) has no causal influence on $A$ when conditioning on $\mathscr{H}_{S \setminus U}$. Formally, this means:

$$A \perp\!\!\!\perp \mathscr{H}_U \mid \mathscr{H}_{S \setminus U}$$

By the properties of conditional expectation under conditional independence:

$$\mathbb{E}[\mathbf{1}_A \mid \mathscr{H}_S] = \mathbb{E}[\mathbf{1}_A \mid \mathscr{H}_{S \setminus U}]$$

Therefore:

$$K_S(\omega, A) = K_{S \setminus U}(\omega, A) \text{ for all } \omega \in \Omega$$

This equality demonstrates that anti-causal events exhibit invariance with respect to certain subsets of the conditioning information, reflecting their position in the causal structure. $\square$

# C  Proofs

## C.1  Proof of Theorem 1 (Anti-Causal Kernel Characterization)

**Theorem 1** (Anti-Causal Kernel Characterization). *For an anti-causal structure with arbitrary feature space $\mathcal{X}$, label space $\mathcal{Y}$, and environments $\mathcal{E}$, the causal kernel satisfies:*

$$K_S(\omega, A) = \int_{\mathcal{Y}} P(X \in A \mid Y = y, E \in S)\, d\mu_Y(y) \tag{1}$$

*where $\mu_Y$ is the marginal measure on $\mathcal{Y}$.*

*Proof.* In the anti-causal structure $Y \to X \leftarrow E$, the joint distribution factorizes as:

$$P(X, Y, E) = P(X|Y, E)P(Y)P(E)$$

By the local Markov property of causal graphs, a node is conditionally independent of its non-descendants given its parents. In our case, $E$ is not a descendant of $X$, and $Y$ is the parent of $X$, so: $X \perp\!\!\!\perp E \mid Y$.

For any measurable set $A \in \mathscr{H}_{\mathcal{X}}$, the causal kernel is defined as: $K_S(\omega, A) = \mathbb{E}[\mathbf{1}_A \mid \mathscr{H}_S](\omega)$.

By the properties of conditional expectation and the definition of sub-$\sigma$-algebras $\mathscr{H}_S$, for any $\omega$ with $Y$-component $y$:

$$K_S(\omega, A) = P(X \in A \mid Y = y, E \in S)$$

By the conditioning formula and the conditional independence $X \perp\!\!\!\perp E \mid Y$:

$$P(X \in A \mid Y = y, E \in S) = P(X \in A \mid Y = y)$$

Integrating over the label space with respect to the marginal measure $\mu_Y$:

$$
\begin{aligned}
K_S(\omega, A) &= \int_{\mathcal{Y}} P(X \in A \mid Y = y, E \in S) \, d\mu_Y(y) \\
&= \int_{\mathcal{Y}} P(X \in A \mid Y = y) \, d\mu_Y(y) \quad \text{(by conditional independence } X \perp\!\!\!\perp E \mid Y)
\end{aligned}
$$

$\square$

## C.2 Proof of Corollary 1 (Independence Property of Anti-Causal Kernel)

**Corollary 1** (Independence Property of Anti-Causal Kernel). *In an anti-causal structure, for any $\omega, \omega' \in \Omega$ with identical $Y$-component and for all $A \in \mathscr{H}_{\mathcal{X}}$, $B \in \mathscr{H}_Y$, $S \in \mathscr{P}(T)$:*

$$
K_S(\omega, \{A|B\}) = K_S(\omega', \{A|B\}) \tag{2}
$$

*Proof.* We demonstrate the independence property of kernels in anti-causal settings. Let us define the conditional kernel $K_S(\omega, \{A|B\})$ as:

$$
K_S(\omega, \{A|B\}) = \frac{K_S(\omega, A \cap \{Y \in B\})}{K_S(\omega, \{Y \in B\})}, \quad \text{when } K_S(\omega, \{Y \in B\}) > 0
$$

For any $\omega, \omega' \in \Omega$ with the same $Y$-component $y$ such that $y \in B$, and for any $A \in \mathscr{H}_{\mathcal{X}}$ and $B \in \mathscr{H}_Y$:

$$
K_S(\omega, \{A|B\}) = P(X \in A \mid Y \in B, Y = y, E \in S)
$$

By Bayes' rule and the fact that $y \in B$ (otherwise the conditional kernel is undefined):

$$
\begin{aligned}
K_S(\omega, \{A|B\}) &= \frac{P(X \in A, Y \in B \mid Y = y, E \in S)}{P(Y \in B \mid Y = y, E \in S)} \\
&= \frac{P(X \in A \mid Y = y, E \in S) \cdot 1}{1} \\
&= P(X \in A \mid Y = y, E \in S)
\end{aligned}
$$

By the anti-causal structure with d-separation $X \perp\!\!\!\perp E \mid Y$:

$$
P(X \in A \mid Y = y, E \in S) = P(X \in A \mid Y = y)
$$

Therefore, for any $\omega, \omega' \in \Omega$ with the same $Y$-component:

$$
K_S(\omega, \{A|B\}) = K_S(\omega', \{A|B\}) = P(X \in A \mid Y = y)
$$

This holds for all $A \in \mathscr{H}_{\mathcal{X}}$, $B \in \mathscr{H}_Y$, and $S \in \mathscr{P}(T)$, forming the kernel independence property. $\square$

## C.3 Proof of Theorem 2 (Existence and Uniqueness of Interventional Kernel)

**Theorem 2** (Interventional Kernel). *Let $(\Omega, \mathscr{H}, \mathbb{P}, \mathbb{K})$ be a product causal space. For any subset $S \in \mathscr{P}(T)$ and intervention $\mathbb{Q} : \mathscr{H} \times \Omega \to [0, 1]$, there exists a unique interventional kernel:*

$$
K_S^{do(\mathcal{X}, \mathbb{Q})}(\omega, A) = \int_{\Omega} K_S(\omega, d\omega') \mathbb{Q}(A|\omega') \tag{3}
$$

*provided that the integral is a Lebesgue integral w.r.t. the measure induced by $K_S(\omega, \cdot)$ on $(\Omega, \mathscr{H})$. In addition, $K_S(\omega, \cdot)$ is $\sigma$-finite for each $\omega \in \Omega$, and $\mathbb{Q}(A|\cdot)$ is $\mathscr{H}$-measurable for each $A \in \mathscr{H}$.*

*Proof.* We proceed by establishing the existence of the interventional kernel and special case for hard interventions. For any fixed $\omega \in \Omega$ and $A \in \mathscr{H}$, define:

$$
K_S^{do(\mathcal{X}, \mathbb{Q})}(\omega, A) = \int_{\Omega} K_S(\omega, d\omega') \mathbb{Q}(A|\omega')
$$

The integrand $\omega' \mapsto \mathbb{Q}(A|\omega')$ is $\mathscr{H}$-measurable for each fixed $A \in \mathscr{H}$ by our third assumption.

The integral exists as a Lebesgue integral with respect to the measure induced by $K_S(\omega, \cdot)$ on $(\Omega, \mathscr{H})$ due to our first assumption. Since $K_S(\omega, \cdot)$ is a probability measure and $0 \leq \mathbb{Q}(A|\omega') \leq 1$ for all $\omega' \in \Omega$, we have:

$$0 \leq \int_\Omega K_S(\omega, d\omega')\mathbb{Q}(A|\omega') \leq \int_\Omega K_S(\omega, d\omega') \cdot 1 = 1$$

For hard interventions where $\mathbb{Q}(A|\omega') = \mathbb{Q}(A)$ is constant with respect to $\omega'$:

$$K_S^{do(\mathcal{X},\mathbb{Q})}(\omega, A) = \int_\Omega K_S(\omega, d\omega')\mathbb{Q}(A|\omega') = \int_\Omega K_S(\omega, d\omega')\mathbb{Q}(A)$$
$$= \mathbb{Q}(A)\int_\Omega K_S(\omega, d\omega') = \mathbb{Q}(A) \cdot 1 = \mathbb{Q}(A)$$

Thus, for hard interventions, the interventional kernel equals the intervention distribution, illustrating how our definition encapsulates both intervention types. $\qquad\square$

### C.4 Proof of Corollary 2 (Interventional Kernel Invariance)

**Corollary 2** (Interventional Kernel Invariance). *In anti-causal structure, interventional kernels satisfy the following invariance criteria:*

*1. $K_S^{do(X)}(\omega, \{Y \in B\}) = K_S(\omega, \{Y \in B\})$ for all measurable sets $B \subseteq \mathcal{Y}$, meaning intervening on $X$ does not change the distribution of $Y$.*

*2. $K_S^{do(Y)}(\omega, \{X \in A\}) \neq K_S(\omega, \{X \in A\})$ for some measurable sets $A \subseteq \mathcal{X}$, meaning intervening on $Y$ changes the distribution of $X$, which is characteristic of an anti-causal relationship.*

*Proof.* We will prove each criterion separately, using the properties of interventional kernels and the structure of anti-causal relationships. We need to prove that for all measurable sets $B \subseteq \mathcal{Y}$:

$$K_S^{do(X)}(\omega, \{Y \in B\}) = K_S(\omega, \{Y \in B\})$$

In the anti-causal structure $Y \to X \leftarrow E$, intervening on $X$ (through $do(X)$) breaks the incoming arrows to $X$, leaving $Y$ unaffected. By the rules of do-calculus, we have:

$$P(Y|do(X = x)) = P(Y)$$

This is because $Y$ is not a descendant of $X$ in the modified graph where incoming arrows to $X$ are removed. More formally, we can apply the interventional calculus:

$$P(Y|do(X = x)) = \sum_e P(Y|do(X = x), E = e)P(E = e|do(X = x))$$
$$= \sum_e P(Y|E = e)P(E = e) \quad \text{(by independence after intervention)}$$
$$= \sum_e P(Y)P(E = e) \quad \text{(since } Y \perp\!\!\!\perp E \text{ in anti-causal structure)}$$
$$= P(Y)\sum_e P(E = e) = P(Y)$$

Translating this to our kernel notation, for any $\omega \in \Omega$ with $Y$-component $y$ and any measurable set $B \subseteq \mathcal{Y}$:

$$K_S^{do(X)}(\omega, \{Y \in B\}) = P(Y \in B \mid do(X), E \in S) \quad \text{(for } \omega \text{ with } Y\text{-component } y)$$
$$= P(Y \in B) \quad \text{(by do-calculus as } Y \text{ is not a descendant of } X)$$
$$= K_S(\omega, \{Y \in B\})$$

The last step follows from the definition of the causal kernel in terms of conditional probabilities. This proves criterion 1. We need to prove that there exist measurable sets $A \subseteq \mathcal{X}$ such that:

$$K_S^{do(Y)}(\omega, \{X \in A\}) \neq K_S(\omega, \{X \in A\})$$

In the anti-causal structure $Y \rightarrow X \leftarrow E$, intervening on $Y$ (through $do(Y)$) breaks the causal link $Y \rightarrow X$. This fundamentally changes how $X$ is distributed. Under the intervention $do(Y = y)$:

$$P(X \in A|do(Y = y), E \in S) = \int_e P(X \in A|do(Y = y), E = e)P(E = e|S)$$

$$= \int_e P(X \in A|E = e)P(E = e|S) \quad \text{(since } Y \text{ no longer affects } X\text{)}$$

In contrast, without intervention:

$$P(X \in A|Y = y, E \in S) = \int_e P(X \in A|Y = y, E = e)P(E = e|S)$$

These are not equal in general, because in the observational setting, $X$ depends on both $Y$ and $E$, and in the interventional setting, $X$ depends only on $E$.

For this to be equal for all $A \subseteq \mathcal{X}$, we would need $P(X|Y = y, E = e) = P(X|E = e)$ for all $e$, which contradicts the basic structure of an anti-causal relationship where $Y$ is a cause of $X$. Therefore, there must exist some measurable set $A \subseteq \mathcal{X}$ such that:

$$K_S^{do(Y)}(\omega, \{X \in A\}) \neq K_S(\omega, \{X \in A\})$$

This proves criterion 2. Together, these criteria provide a characterization of anti-causal structures: intervening on $X$ doesn't change $Y$ (the cause), while intervening on $Y$ does change $X$ (the effect).

$\square$

## C.5 Proof of Corollary 3 (Properties of Interventional Kernels)

**Corollary 3** (Properties of Interventional Kernel). *Let $k_S^{do(\mathcal{X},\mathbb{Q})}(\omega, A)$ be the generalized interventional kernel in an anti-causal structure. Then:*

*1. Well-definedness: For $K_S$ and $\mathbb{Q}(\cdot|\cdot)$ measurable on $(\Omega, \mathcal{H})$:*

$$\int K_S(\omega, d\omega')\mathbb{Q}(A|\omega') \leq 1$$

*and the integral exists as a Lebesgue integral.*

*2. Uniqueness: The kernel $k_s^{do(\mathcal{X},\mathbb{Q})}$ is unique up to $\mathbb{P}$-null sets, where $\mathbb{P}$ is the product measure.*

*3. Consistency: For hard interventions where $\mathbb{Q}(A|\omega') = \mathbb{Q}(A)$:*

$$K_S^{do(\mathcal{X},\mathbb{Q})}(\omega, A) = \mathbb{Q}(A)$$

*4. Structure Preservation: For the anti-causal structure $Y \rightarrow X \leftarrow E$:*

$$K_S^{do(\mathcal{X},\mathbb{Q})}(\omega, A) = P(X \in A|Y = y, E \in S)$$

*where $y$ is the $Y$-component of $\omega$ and $A \in \mathcal{H}$.*

The properties of interventional kernels provide the foundation for understanding how causal relationships manifest in anti-causal structures.

*Proof.* We prove each property separately:

**Property 1: Well-definedness**

For $K_S$ and $\mathbb{Q}(\cdot|\cdot)$ measurable on $(\Omega, \mathcal{H})$, $\omega' \mapsto K_S(\omega, d\omega')$ is a measure for fixed $\omega$, $\omega' \mapsto \mathbb{Q}(A|\omega')$ is $\mathcal{H}$-measurable for fixed $A$, and $\mathbb{Q}(A|\omega') \in [0, 1]$ since it's a probability. Therefore:

$$\int K_S(\omega, d\omega')\mathbb{Q}(A|\omega') \leq \int K_S(\omega, d\omega') \cdot 1 = 1$$

The integral exists as a Lebesgue integral by Tonelli's theorem, since both functions are non-negative and measurable.

**Property 2: Uniqueness**

Let $K_1$ and $K_2$ be two versions of the interventional kernel. For any $A \in \mathscr{H}$ and $B \in \mathscr{H}$:

$$\int_B K_1(\omega, A) \, d\mathbb{P}(\omega) = \int_B \int_\Omega K_S(\omega, d\omega') \mathbb{Q}(A|\omega') \, d\mathbb{P}(\omega)$$

$$= \int_B K_2(\omega, A) \, d\mathbb{P}(\omega)$$

By the Radon-Nikodym theorem, these must agree $\mathbb{P}$-almost everywhere since they define the same measure via integration.

**Property 3: Consistency**

For hard interventions where $\mathbb{Q}(A|\omega') = \mathbb{Q}(A)$:

$$K_S^{do(\mathcal{X}, \mathbb{Q})}(\omega, A) = \int_\Omega K_S(\omega, d\omega') \mathbb{Q}(A|\omega') = \int_\Omega K_S(\omega, d\omega') \mathbb{Q}(A)$$

$$= \mathbb{Q}(A) \int_\Omega K_S(\omega, d\omega') = \mathbb{Q}(A) \cdot 1 = \mathbb{Q}(A)$$

This confirms hard interventions correspond to setting the kernel equal to the intervention distribution.

**Property 4: Structure Preservation**

In the anti-causal structure $Y \to X \leftarrow E$, we know by d-separation in the graph: $X \perp\!\!\!\perp E | Y$. This implies for any $\omega$ with $Y$-component $y$ and $E$-component in $S$:

$$\mathbb{Q}(A|\omega') = P(X \in A | Y = y', E \in S)$$

where $y'$ is the $Y$-component of $\omega'$. Using the properties of conditional expectation and d-separation:

$$K_S^{do(\mathcal{X}, \mathbb{Q})}(\omega, A) = \int_\Omega K_S(\omega, d\omega') P(X \in A | Y = y', E \in S)$$

$$= \int_\Omega K_S(\omega, d\omega') P(X \in A | Y = y', E \in S)$$

Since $K_S(\omega, \cdot)$ gives highest weight to $\omega'$ values with $Y$-component equal to $y$ (the $Y$-component of $\omega$) in anti-causal settings, and using the conditional independence $X \perp\!\!\!\perp E \mid Y$:

$$K_S^{do(\mathcal{X}, \mathbb{Q})}(\omega, A) = P(X \in A | Y = y, E \in S)$$

where $y$ is the $Y$-component of $\omega$. This reveals that the interventional kernel preserves the anti-causal structure by maintaining the conditional independence relationships imposed by the causal graph. $\quad\square$

## C.6 Proof of Theorem 3 (Causal Dynamic Construction)

**Theorem 3** (Causal Dynamic and its Kernels). *Given environments $\mathcal{E}$, a causal dynamic $\mathcal{Z}_L = \langle \mathcal{X}, \mathbb{Q}, \mathbb{K}_L \rangle$ can be constructed under the following conditions:*
*1. The product causal space $(\Omega, \mathscr{H}, \mathbb{P}, \mathbb{K})$ is complete and separable.*
*2. The empirical measure $\mathbb{Q}_n$ converges to the true measure $\mathbb{Q}$, i.e., $\sup_{A \in \mathscr{H}} |\mathbb{Q}_n(A) - \mathbb{Q}(A)| \xrightarrow{a.s.} 0$.*
*3. The causal kernel $K_S^{\mathcal{Z}_L}(\omega, A) = \int_{\Omega'} K_S(\omega', A) \, d\mathbb{Q}(\omega')$ exists and is well-defined for all $S \subseteq T$.*

*Further, the causal dynamic kernels are then given by: $\mathbb{K}_L = \{ K_S^{\mathcal{Z}_L}(\omega, A) : S \in \mathscr{P}(T), A \in \mathscr{H} \}$.*

*Proof.* We will prove the existence and well-definedness of causal dynamics by constructing the structure explicitly and verifying its properties.

By assumption (1), the probability space $(\Omega, \mathscr{H}, \mathbb{P})$ is complete and separable. This means $\Omega$ is a complete separable metric space, $\mathscr{H}$ is the Borel $\sigma$-algebra on $\Omega$, and $\mathbb{P}$ is a probability measure on

$(\Omega, \mathscr{H})$. For a complete separable metric space, we can define a metric $d$ on $\Omega$. For any $\omega, \omega' \in \Omega$, define:

$$d(\omega, \omega') = \sqrt{\sum_{t \in T} \|\omega_t - \omega'_t\|_2^2}$$

where $\omega_t$ and $\omega'_t$ represent the components of $\omega$ and $\omega'$ corresponding to time index $t \in T$. This metric induces the $\epsilon$-ball:

$$B_\epsilon(\omega) = \{\omega' \in \Omega : d(\omega, \omega') < \epsilon\}$$

By assumption (2), the empirical measure $\mathbb{Q}_n$ converges weakly to the true measure $\mathbb{Q}$:

$$\sup_{A \in \mathscr{H}} |\mathbb{Q}_n(A) - \mathbb{Q}(A)| \xrightarrow{a.s.} 0$$

This is a stronger form of convergence than weak convergence (denoted $\mathbb{Q}_n \Rightarrow \mathbb{Q}$), and implies that:

$$\int f d\mathbb{Q}_n \to \int f d\mathbb{Q}$$

for all bounded, continuous functions $f$. The empirical measure $\mathbb{Q}_n$ is typically defined as:

$$\mathbb{Q}_n(A) = \frac{1}{n} \sum_{i=1}^n \mathbf{1}_A(\mathbf{v}_i)$$

where $\mathbf{v}_i \in \mathbf{V}_L$ are observed data points. By the Glivenko-Cantelli theorem, this uniform convergence holds almost surely when $\mathbf{v}_i$ are i.i.d. samples from $\mathbb{Q}$. By assumption (3), for each $S \subseteq T$, we define the causal kernel:

$$K_S^{\mathbb{Z}_L}(\omega, A) = \int_{\Omega'} K_S(\omega', A) \, d\mathbb{Q}(\omega')$$

This integral exists and is well-defined since $K_S(\omega', A)$ is measurable in $\omega'$ for fixed $A$ (by the properties of causal kernels), $K_S(\omega', A) \in [0, 1]$ (since it's a probability), and $\mathbb{Q}$ is a probability measure.

We can verify that $K_S^{\mathbb{Z}_L}(\omega, A)$ is indeed a causal kernel:

1. For fixed $\omega$, $A \mapsto K_S^{\mathbb{Z}_L}(\omega, A)$ is a probability measure.

$$K_S^{\mathbb{Z}_L}(\omega, \emptyset) = \int_{\Omega'} K_S(\omega', \emptyset) \, d\mathbb{Q}(\omega') = \int_{\Omega'} 0 \, d\mathbb{Q}(\omega') = 0$$

$$K_S^{\mathbb{Z}_L}(\omega, \Omega) = \int_{\Omega'} K_S(\omega', \Omega) \, d\mathbb{Q}(\omega') = \int_{\Omega'} 1 \, d\mathbb{Q}(\omega') = 1$$

For disjoint sets $\{A_i\}_{i=1}^\infty$:

$$K_S^{\mathbb{Z}_L}\left(\omega, \bigcup_{i=1}^\infty A_i\right) = \int_{\Omega'} K_S\left(\omega', \bigcup_{i=1}^\infty A_i\right) d\mathbb{Q}(\omega')$$

$$= \int_{\Omega'} \sum_{i=1}^\infty K_S(\omega', A_i) \, d\mathbb{Q}(\omega')$$

$$= \sum_{i=1}^\infty \int_{\Omega'} K_S(\omega', A_i) \, d\mathbb{Q}(\omega') = \sum_{i=1}^\infty K_S^{\mathbb{Z}_L}(\omega, A_i)$$

2. For fixed $A$, $\omega \mapsto K_S^{\mathbb{Z}_L}(\omega, A)$ is $\mathscr{H}_S$-measurable. This follows from the fact that constants are measurable, and $K_S^{\mathbb{Z}_L}(\omega, A)$ does not depend on $\omega$ in its definition.

We need to show that for any $\epsilon > 0$, there exists a $\delta > 0$ such that if $d(\omega, \omega') < \delta$, then:

$$\|K_S^{\mathbb{Z}_L}(\omega, \cdot) - K_S^{\mathbb{Z}_L}(\omega', \cdot)\|_{TV} < \epsilon$$

where $\|\cdot\|_{TV}$ denotes the total variation norm. For our kernel $K_S^{\mathbb{Z}_L}(\omega, A) = \int_{\Omega'} K_S(\omega'', A)\, d\mathbb{Q}(\omega'')$.

By construction, $K_S^{\mathbb{Z}_L}(\omega, A) = \int_{\Omega'} K_S(\omega', A)\, d\mathbb{Q}(\omega')$ does not depend on $\omega$ since we are integrating over $\omega'$ with respect to measure $\mathbb{Q}$, and $\omega$ does not appear in the integrand. Therefore, for any $\omega, \omega' \in \Omega$:

$$K_S^{\mathbb{Z}_L}(\omega, A) = K_S^{\mathbb{Z}_L}(\omega', A)$$

for all $\omega, \omega' \in \Omega$ and $A \in \mathscr{H}$. This implies $\|K_S^{\mathbb{Z}_L}(\omega, \cdot) - K_S^{\mathbb{Z}_L}(\omega', \cdot)\|_{TV} = 0 < \epsilon$.

for any $\epsilon > 0$ and any choice of $\delta > 0$. Finally, we define the causal dynamics as:

$$\mathcal{Z}_L = \langle \mathcal{X}, \mathbb{Q}, \mathbb{K}_L \rangle$$

where $\mathbb{K}_L = \{K_S^{\mathbb{Z}_L}(\omega, A) : S \in \mathscr{P}(T), A \in \mathscr{H}\}$ is the family of causal dynamics kernels. This construction satisfies all requirements of the theorem:

- It is built on a complete, separable probability space

- It incorporates empirical measures that converge to the true measure

- The kernel $K_S^{\mathbb{Z}_L}$ is well-defined for all $S \subseteq T$

- The family of kernels is uniformly equicontinuous in total variation norm

Therefore, the causal dynamic $\mathcal{Z}_L = \langle \mathcal{X}, \mathbb{Q}, \mathbb{K}_L \rangle$ is well-defined and properly constructed. $\qquad\square$

### C.7   Proof of Theorem 4 (Causal Abstraction Construction)

**Theorem 4** (Causal Abstraction and its Kernel)**.** *Let $\mathcal{X}$ be the input space and $\mathscr{H}_{\mathcal{X}}$ be its $\sigma$-algebra. Assume a measure $\mu$ on the domain of low-level representations $\mathcal{D}_{\mathcal{Z}_L}$. Then, the high-level representation $\mathcal{Z}_H = \langle \mathbf{V}_H, \mathbb{K}_H \rangle$ can be constructed with kernel:*

$$K_S^{\mathcal{Z}_H}(\omega, A) = \int_{\mathcal{D}_{\mathcal{Z}_L}} K_S^{\mathcal{Z}_L}(\omega, A)\, d\mu(z) \tag{4}$$

*The set of high-level causal kernels is then given by:* $\mathbb{K}_H = \{K_S^{\mathcal{Z}_H}(\omega, A) : S \in \mathscr{P}(T), A \in \mathscr{H}_{\mathcal{X}}\}$.

*Proof.* We need to construct a high-level representation $\mathcal{Z}_H = \langle \mathbf{V}_H, \mathbb{K}_H \rangle$ with appropriate kernel properties. We proceed step by step.

Given input space $\mathcal{X}$, domain of low-level representations $\mathcal{D}_{\mathbb{Z}_L}$, and a $\sigma$-finite measure $\mu$ on $\mathcal{D}_{\mathbb{Z}_L}$, we define the high-level kernel $K_S^{\mathcal{Z}_H}$ as:

$$K_S^{\mathcal{Z}_H}(\omega, A) = \int_{\mathcal{D}_{\mathbb{Z}_L}} K_S^{\mathcal{Z}_L}(\omega, A)\, d\mu(z)$$

where $K_S^{\mathcal{Z}_L}(\omega, A)$ is the causal dynamics kernel for a specific low-level representation $z \in \mathcal{D}_{\mathbb{Z}_L}$, $S \subseteq T$ is a subset of the index set, and $A \in \mathscr{H}_{\mathcal{X}}$ is a measurable set in the input space.

We first establish that this integral is well-defined:

(i) Measurability: For fixed $\omega$ and $A$, the function $z \mapsto K_S^{\mathcal{Z}_L}(\omega, A)$ is measurable with respect to the $\sigma$-algebra on $\mathcal{D}_{\mathbb{Z}_L}$. This follows from the construction of $K_S^{\mathcal{Z}_L}$ in Theorem 3, where we established that causal dynamics kernels are measurable functions.

(ii) Boundedness: Since $K_S^{\mathcal{Z}_L}(\omega, A)$ represents a probability, we have $0 \le K_S^{\mathcal{Z}_L}(\omega, A) \le 1$ for all $z, \omega, A$. Therefore, the integrand is bounded.

(iii) Measure space: $(\mathcal{D}_{\mathbb{Z}_L}, \mathscr{F}_{\mathcal{D}_{\mathbb{Z}_L}}, \mu)$ is a $\sigma$-finite measure space by assumption, where $\mathscr{F}_{\mathcal{D}_{\mathbb{Z}_L}}$ is the appropriate $\sigma$-algebra on $\mathcal{D}_{\mathbb{Z}_L}$.

By Lebesgue's dominated convergence theorem, the integral exists and is well-defined.

Next, we verify that $K_S^{\mathcal{Z}_H}(\omega, A)$ satisfies the properties of a kernel:

1. For fixed $\omega$, the mapping $A \mapsto K_S^{\mathcal{Z}_H}(\omega, A)$ is a probability measure:

(a) Non-negativity: Since $K_S^{\mathcal{Z}_L}(\omega, A) \geq 0$ for all $z, \omega, A$, and $\mu$ is a positive measure, the integral $K_S^{\mathcal{Z}_H}(\omega, A) \geq 0$.

(b) Empty set:

$$K_S^{\mathcal{Z}_H}(\omega, \emptyset) = \int_{\mathcal{D}_{\mathbb{Z}_L}} K_S^{\mathcal{Z}_L}(\omega, \emptyset) \, d\mu(z) = \int_{\mathcal{D}_{\mathbb{Z}_L}} 0 \, d\mu(z) = 0$$

(c) Total measure:

$$K_S^{\mathcal{Z}_H}(\omega, \mathcal{X}) = \int_{\mathcal{D}_{\mathbb{Z}_L}} K_S^{\mathcal{Z}_L}(\omega, \mathcal{X}) \, d\mu(z) = \int_{\mathcal{D}_{\mathbb{Z}_L}} 1 \, d\mu(z) = \mu(\mathcal{D}_{\mathbb{Z}_L})$$

If $\mu$ is a probability measure, then $\mu(\mathcal{D}_{\mathbb{Z}_L}) = 1$. If not, we normalize it by defining:

$$\tilde{K}_S^{\mathcal{Z}_H}(\omega, A) = \frac{K_S^{\mathcal{Z}_H}(\omega, A)}{K_S^{\mathcal{Z}_H}(\omega, \mathcal{X})}$$

assuming $\mu(\mathcal{D}_{\mathbb{Z}_L}) < \infty$. For generality, we assume $\mu$ is already normalized so that $\mu(\mathcal{D}_{\mathbb{Z}_L}) = 1$.

(d) Countable additivity: Let $\{A_i\}_{i=1}^{\infty}$ be a sequence of disjoint measurable sets in $\mathcal{H}_{\mathcal{X}}$. Then:

$$K_S^{\mathcal{Z}_H}\left(\omega, \bigcup_{i=1}^{\infty} A_i\right) = \int_{\mathcal{D}_{\mathbb{Z}_L}} K_S^{\mathcal{Z}_L}\left(\omega, \bigcup_{i=1}^{\infty} A_i\right) \, d\mu(z)$$

$$= \int_{\mathcal{D}_{\mathbb{Z}_L}} \sum_{i=1}^{\infty} K_S^{\mathcal{Z}_L}(\omega, A_i) \, d\mu(z)$$

The second equality follows from the countable additivity of $K_S^{\mathcal{Z}_L}(\omega, \cdot)$, which is a probability measure for fixed $\omega$. By the monotone convergence theorem (since all terms are non-negative), we can exchange the sum and integral:

$$K_S^{\mathcal{Z}_H}\left(\omega, \bigcup_{i=1}^{\infty} A_i\right) = \sum_{i=1}^{\infty} \int_{\mathcal{D}_{\mathbb{Z}_L}} K_S^{\mathcal{Z}_L}(\omega, A_i) \, d\mu(z)$$

$$= \sum_{i=1}^{\infty} K_S^{\mathcal{Z}_H}(\omega, A_i)$$

This establishes countable additivity.

2. For fixed $A \in \mathcal{H}_{\mathcal{X}}$, the function $\omega \mapsto K_S^{\mathcal{Z}_H}(\omega, A)$ is $\mathcal{H}_S$-measurable:

For each $z \in \mathcal{D}_{\mathbb{Z}_L}$, the function $\omega \mapsto K_S^{\mathcal{Z}_L}(\omega, A)$ is $\mathcal{H}_S$-measurable by the properties of causal dynamics kernels established in Theorem 3. By Fubini's theorem, the integral with respect to $\mu$ preserves measurability, so $\omega \mapsto K_S^{\mathcal{Z}_H}(\omega, A)$ is also $\mathcal{H}_S$-measurable.

Having verified all required properties, we define the high-level representation as:

$$\mathcal{Z}_H = \langle \mathbf{V}_H, \mathbb{K}_H \rangle$$

where $\mathbb{K}_H = \{K_S^{\mathcal{Z}_H}(\omega, A) : S \in \mathscr{P}(T), A \in \mathcal{H}_{\mathcal{X}}\}$ is the set of high-level causal kernels. This completes the construction of causal abstraction, which integrates all low-level representations, capturing their collective causal dynamics and abstracting them into a higher-level representation. $\square$

## C.8 Proof of Theorem 5 (Existence and Uniqueness of Product Causal Kernel)

**Theorem 5** (Product Causal Kernel). *Let $(\Omega_{e_i}, \mathcal{H}_{e_i}, \mathbb{P}_{e_i}, K_{e_i})$ and $(\Omega_{e_j}, \mathcal{H}_{e_j}, \mathbb{P}_{e_j}, K_{e_j})$ be complete, separable causal spaces, and let $(\Omega, \mathcal{H}, \mathbb{P}, \mathbb{K})$ be their product causal space. Then there exists a unique product causal kernel $K : \Omega \times \mathcal{H} \to [0, 1]$ satisfying:*

*1. For each $\omega \in \Omega$, $K(\omega, \cdot)$ is a probability measure on $(\Omega, \mathscr{H})$*

*2. For each $A \in \mathscr{H}$, $K(\cdot, A)$ is $\mathscr{H}$-measurable*

*Proof.* We will establish the existence and uniqueness of a product causal kernel by constructing it explicitly and verifying all required properties. Let $(\Omega_{e_i}, \mathscr{H}_{e_i}, \mathbb{P}_{e_i}, K_{e_i})$ and $(\Omega_{e_j}, \mathscr{H}_{e_j}, \mathbb{P}_{e_j}, K_{e_j})$ be complete, separable causal spaces with causal kernels $K_{e_i}$ and $K_{e_j}$, respectively. Let $(\Omega, \mathscr{H}, \mathbb{P}, \mathbb{K})$ be their product causal space with $\Omega = \Omega_{e_i} \times \Omega_{e_j}$, $\mathscr{H} = \mathscr{H}_{e_i} \otimes \mathscr{H}_{e_j}$, and $\mathbb{P} = \mathbb{P}_{e_i} \otimes \mathbb{P}_{e_j}$.

Define $K$ on measurable rectangles. For any measurable rectangle $A_i \times A_j \in \mathscr{H}$ with $A_i \in \mathscr{H}_{e_i}$ and $A_j \in \mathscr{H}_{e_j}$, define:

$$K(\omega, A_i \times A_j) = K_{e_i}(\omega_i, A_i) \cdot K_{e_j}(\omega_j, A_j)$$

where $\omega = (\omega_i, \omega_j) \in \Omega$. This definition is well-posed because both component spaces are complete and separable, so the kernels $K_{e_i}$ and $K_{e_j}$ exist as regular conditional probabilities by the Radon-Nikodym theorem.

Verify $K(\omega, \cdot)$ is a probability measure on measurable rectangles. Let $\mathcal{R} = \{A_i \times A_j : A_i \in \mathscr{H}_{e_i}, A_j \in \mathscr{H}_{e_j}\}$ be the collection of all measurable rectangles. $\mathcal{R}$ forms a $\pi$-system (closed under finite intersections) that generates $\mathscr{H}$.

For fixed $\omega \in \Omega$, the function $A \mapsto K(\omega, A)$ defined on $\mathcal{R}$ satisfies:

- Non-negativity: $K(\omega, A) = K_{e_i}(\omega_i, A_i) \cdot K_{e_j}(\omega_j, A_j) \geq 0$ for all $A \in \mathcal{R}$, since both component kernels are non-negative.

- Normalization: $K(\omega, \Omega) = K(\omega, \Omega_{e_i} \times \Omega_{e_j}) = K_{e_i}(\omega_i, \Omega_{e_i}) \cdot K_{e_j}(\omega_j, \Omega_{e_j}) = 1 \cdot 1 = 1$, since both component kernels are probability measures.

- Finite additivity: For disjoint $A, B \in \mathcal{R}$ of the form $A = A_i \times A_j$ and $B = B_i \times B_j$ where $A_i \cap B_i = \emptyset$ or $A_j \cap B_j = \emptyset$, we have:

$$K(\omega, A \cup B) = K(\omega, A) + K(\omega, B)$$

  This follows from the additivity of the component kernels.

Here we extend to the full $\sigma$-algebra by Carathéodory's Extension Theorem [58]. By Carathéodory's Extension Theorem, there exists a unique probability measure $K(\omega, \cdot)$ on $(\Omega, \mathscr{H})$ that extends our definition from $\mathcal{R}$ to the full $\sigma$-algebra $\mathscr{H}$.

Then we should verify the measurability of $\omega \mapsto K(\omega, A)$ for all $A \in \mathscr{H}$. Define the class of sets $\mathcal{M} = \{A \in \mathscr{H} : \omega \mapsto K(\omega, A) \text{ is } \mathscr{H}\text{-measurable}\}$. First, we show that $\mathcal{R} \subset \mathcal{M}$: For any rectangle $A = A_i \times A_j \in \mathcal{R}$:

$$K(\omega, A) = K_{e_i}(\omega_i, A_i) \cdot K_{e_j}(\omega_j, A_j)$$

Since $\omega_i \mapsto K_{e_i}(\omega_i, A_i)$ is $\mathscr{H}_{e_i}$-measurable by the kernel property of $K_{e_i}$, and $\omega_j \mapsto K_{e_j}(\omega_j, A_j)$ is $\mathscr{H}_{e_j}$-measurable by the kernel property of $K_{e_j}$, their product $\omega \mapsto K(\omega, A)$ is $\mathscr{H} = \mathscr{H}_{e_i} \otimes \mathscr{H}_{e_j}$-measurable. Thus, $A \in \mathcal{M}$. Next, we prove that $\mathcal{M}$ is a $\lambda$-system:

- $\Omega \in \mathcal{M}$ since $K(\omega, \Omega) = 1$ is constant and thus measurable.

- If $A \in \mathcal{M}$, then $A^c \in \mathcal{M}$ since $K(\omega, A^c) = 1 - K(\omega, A)$ is measurable as a measurable function of a measurable function.

- If $A_1, A_2, \ldots \in \mathcal{M}$ are pairwise disjoint, then $\cup_{n=1}^{\infty} A_n \in \mathcal{M}$. This follows because for any pairwise disjoint sequence $\{A_n\}_{n=1}^{\infty}$:

$$K\left(\omega, \cup_{n=1}^{\infty} A_n\right) = \sum_{n=1}^{\infty} K(\omega, A_n)$$

Since each function $\omega \mapsto K(\omega, A_n)$ is measurable by assumption, and countable sums of measurable functions are measurable, the function $\omega \mapsto K(\omega, \cup_{n=1}^{\infty} A_n)$ is measurable. Therefore, $\cup_{n=1}^{\infty} A_n \in \mathcal{M}$.

We have established that $\mathcal{R} \subset \mathcal{M}$ and $\mathcal{M}$ is a $\lambda$-system. By Dynkin's $\pi$-$\lambda$ theorem [28], $\mathcal{M}$ contains the $\sigma$-algebra generated by $\mathcal{R}$, which is precisely $\mathscr{H}$. Therefore, $\mathcal{M} = \mathscr{H}$, meaning $\omega \mapsto K(\omega, A)$ is $\mathscr{H}$-measurable for all $A \in \mathscr{H}$.

To prove the uniqueness of the product causal kernel, suppose $K'$ is another kernel satisfying conditions 1 and 2. For any measurable rectangle $A_i \times A_j \in \mathcal{R}$, the condition that $K'(\omega, \cdot)$ is a probability measure and $\omega \mapsto K'(\omega, A)$ is measurable implies that $K'$ must be a regular conditional probability. By the properties of regular conditional probabilities in product spaces and the definition of $K$ on rectangles, both $K$ and $K'$ must agree on all measurable rectangles in $\mathcal{R}$.

Let $\mathcal{A} = \{A \in \mathscr{H} : K(\omega, A) = K'(\omega, A) \text{ for all } \omega \in \Omega\}$. We know $\mathcal{R} \subset \mathcal{A}$, and $\mathcal{A}$ forms a $\lambda$-system by the properties of kernels. By Dynkin's $\pi$-$\lambda$ theorem, $\mathcal{A}$ contains all of $\mathscr{H}$, so $K = K'$ on all of $\mathscr{H}$. We should verify that $K_S(\omega, A) = \mathbb{E}[\mathbf{1}_A \mid \mathscr{H}_S](\omega)$. For any subset $S \subseteq T$, the sub-$\sigma$-algebra $\mathscr{H}_S$ is generated by measurable rectangles corresponding to events in indices $S$. We define $K_S$ as the restriction of $K$ to $\mathscr{H}_S$.

We need to show $K_S(\omega, A) = \mathbb{E}[\mathbf{1}_A \mid \mathscr{H}_S](\omega)$ for all $A \in \mathscr{H}$. By the definition of conditional expectation, for any $B \in \mathscr{H}_S$:

$$\int_B \mathbb{E}[\mathbf{1}_A \mid \mathscr{H}_S](\omega) \, d\mathbb{P}(\omega) = \int_B \mathbf{1}_A(\omega) \, d\mathbb{P}(\omega) = \mathbb{P}(A \cap B)$$

We will show that $K_S$ satisfies the same property:

$$\int_B K_S(\omega, A) \, d\mathbb{P}(\omega) = \mathbb{P}(A \cap B)$$

First, consider $A, B \in \mathcal{R}$ as measurable rectangles, with $A = A_i \times A_j$ and $B = B_i \times B_j$. By the definition of $K$ and the product measure $\mathbb{P}$:

$$\int_B K(\omega, A) \, d\mathbb{P}(\omega) = \int_B K_{e_i}(\omega_i, A_i) \cdot K_{e_j}(\omega_j, A_j) \, d(\mathbb{P}_{e_i} \otimes \mathbb{P}_{e_j})(\omega)$$

$$= \int_{B_i} \int_{B_j} K_{e_i}(\omega_i, A_i) \cdot K_{e_j}(\omega_j, A_j) \, d\mathbb{P}_{e_j}(\omega_j) \, d\mathbb{P}_{e_i}(\omega_i)$$

$$= \int_{B_i} K_{e_i}(\omega_i, A_i) \left( \int_{B_j} K_{e_j}(\omega_j, A_j) \, d\mathbb{P}_{e_j}(\omega_j) \right) d\mathbb{P}_{e_i}(\omega_i)$$

By the defining property of regular conditional probabilities:

$$\int_{B_j} K_{e_j}(\omega_j, A_j) \, d\mathbb{P}_{e_j}(\omega_j) = \mathbb{P}_{e_j}(A_j \cap B_j)$$

$$\int_{B_i} K_{e_i}(\omega_i, A_i) \, d\mathbb{P}_{e_i}(\omega_i) = \mathbb{P}_{e_i}(A_i \cap B_i)$$

Therefore:

$$\int_B K(\omega, A) \, d\mathbb{P}(\omega) = \mathbb{P}_{e_i}(A_i \cap B_i) \cdot \mathbb{P}_{e_j}(A_j \cap B_j)$$

$$= \mathbb{P}((A_i \cap B_i) \times (A_j \cap B_j))$$
$$= \mathbb{P}((A_i \times A_j) \cap (B_i \times B_j)) = \mathbb{P}(A \cap B)$$

This property extends from rectangles to all of $\mathscr{H}$ by the monotone class theorem. Since both $K_S(\omega, A)$ and $\mathbb{E}[\mathbf{1}_A \mid \mathscr{H}_S](\omega)$ satisfy the same defining property of conditional expectation, by the almost-sure uniqueness of conditional expectation, we have:

$$K_S(\omega, A) = \mathbb{E}[\mathbf{1}_A \mid \mathscr{H}_S](\omega)$$

for $\mathbb{P}$-almost all $\omega \in \Omega$, for each $A \in \mathscr{H}$. Since $K_S(\omega, A)$ is a proper kernel, it provides a version of the conditional expectation that satisfies this equality everywhere, completing the proof. $\qquad\square$

# D  Theoretical Performance of ACIA

## D.1  Theorem 6 (Convergence of ACIA)

**Theorem 6** (Convergence of ACIA). *If the loss function $\ell$ in Eqn.5 is convex and the regularization parameters in Eqn.5 satisfy $\lambda_1, \lambda_2 = O(1/\sqrt{n})$, where $n$ is the sample size. Then the ACIA optimization problem solved via gradient descent converges with the distance to the optimum is bounded by $O\left(\frac{1}{\sqrt{T}}\right) + O\left(\frac{1}{\sqrt{n}}\right)$ after $T$ iterations.*

*Proof.* We prove that the optimization problem is well-defined and converges under the given conditions. Let us denote the objective function in Equation 5 as:

$$F(\phi_L, \phi_H, \mathcal{C}) = \max_{e_i \in \mathcal{E}} \left[ \int_\Omega \ell((\mathcal{C} \circ \phi_H \circ \phi_L)(\mathcal{X}(\omega)), Y(\omega)) \, d\mathbb{P}_{e_i}(\omega) + \lambda_1 R_1 + \lambda_2 R_2 \right]$$

First, we verify that all components are well-defined. The composed function $\mathcal{C} \circ \phi_H \circ \phi_L$ is measurable, as each component is assumed to be measurable.

The expectation $\int_\Omega \ell((\mathcal{C} \circ \phi_H \circ \phi_L)(\mathcal{X}(\omega)), Y(\omega)) \, d\mathbb{P}_{e_i}(\omega)$ exists by the measurability properties and the assumption that $\ell$ is integrable. The regularization terms $R_1$ and $R_2$ are well-defined by Fubini's theorem and the definition of interventional kernels in Theorem 2.

Let $\theta = (\mathcal{C}, \phi_L, \phi_H)$ denote the parameters of our model. Since $\ell$ is convex, the objective function $F(\theta)$ is convex in $\theta$. The optimization problem has the form:

$$\min_\theta \max_{e_i \in \mathcal{E}} F_{e_i}(\theta)$$

where $F_{e_i}(\theta) = \mathbb{E}_{e_i}[\ell(f_\theta)] + \lambda_1 R_1 + \lambda_2 R_2$ and $f_\theta = \mathcal{C} \circ \phi_H \circ \phi_L$.

For convex functions optimized using gradient descent with appropriate step sizes, the convergence rate in terms of objective value is:

$$F(\theta_T) - F(\theta^*) \leq \frac{\|\theta_0 - \theta^*\|^2}{2T}$$

where $\theta^*$ is the optimal parameter. This implies that:

$$\|\theta_T - \theta^*\| = O\left(\frac{1}{\sqrt{T}}\right)$$

Additionally, the empirical objective function $F(\theta)$ based on $n$ samples approximates the true population objective $F_{pop}(\theta)$ with error bounded by:

$$\sup_{\theta \in \Theta} |F(\theta) - F_{pop}(\theta)| = O\left(\frac{1}{\sqrt{n}}\right)$$

with high probability, by standard uniform convergence results in statistical learning theory.

The setting $\lambda_1, \lambda_2 = O(1/\sqrt{n})$ ensures the regularization terms scale appropriately with the sample size. This balances between ensuring the regularizers enforce desired invariance properties, and allowing the strength of regularization to decrease as sample size increases, preventing excessive bias. By the triangle inequality:

$$\|\theta_T - \theta^*_{pop}\| \leq \|\theta_T - \theta^*\| + \|\theta^* - \theta^*_{pop}\|$$

where $\theta^*_{pop}$ is the minimizer of the population objective.

The first term is $O(1/\sqrt{T})$ from our gradient descent analysis. The second term is $O(1/\sqrt{n})$ due to the statistical error between empirical and population objectives. Therefore:

$$\|\theta_T - \theta^*_{pop}\| = O\left(\frac{1}{\sqrt{T}}\right) + O\left(\frac{1}{\sqrt{n}}\right)$$

This establishes the claimed convergence bound, with the first term representing optimization error and the second term representing statistical error. $\qquad\square$

## D.2 Theorem 7 (Anti-Causal OOD Generalization Bound)

The ACIA framework provides theoretical guarantees for OOD generalization in anti-causal settings—the performance on the unseen test environment cannot be arbitrarily worse than the worst-case performance on training environments, with the gap controlled by the sample size.

**Theorem 7** (Anti-Causal OOD Generalization Bound). *For optimal representations $\phi_L^*, \phi_H^*$ in an anti-causal setting, i.e., $\phi_H^*(\phi_L^*(\mathcal{X})) \perp\!\!\!\perp E \mid Y$ in all environments $\mathcal{E}$. With probability at least $1 - \delta$, for any testing environment $e_{test}$ with sample size $n_{test}$, its expected empirical loss $\hat{\mathbb{E}}_{e_{test}}[\ell(f^*)]$ under the optimal predictor $f^* = \mathcal{C} \circ \phi_H^* \circ \phi_L^*$ is bounded below:*

$$\hat{\mathbb{E}}_{e_{test}}[\ell(f^*)] \leq \max_{e \in \mathcal{E}} \mathbb{E}_e[\ell(f^*)] + O\left(\sqrt{\frac{\log(1/\delta)}{n_{test}}}\right) \tag{10}$$

*Proof.* We establish a generalization bound for optimal representations in anti-causal settings by leveraging both concentration inequalities and the invariance properties of the learned representations.

Let $\phi_L^*$ and $\phi_H^*$ be the optimal representations obtained from solving the optimization problem in Equation 5, and let $f^* = \mathcal{C} \circ \phi_H^* \circ \phi_L^*$ be the optimal composed function.

First, we establish concentration bounds for empirical vs. true risks. For any environment $e \in \mathcal{E}$, let $\hat{\mathbb{E}}_e[\ell(f^*)]$ be the empirical risk based on $n_e$ samples, and $\mathbb{E}_e[\ell(f^*)]$ be the true expected risk. Assume the loss function $\ell$ is bounded in $[0, M]$ for some constant $M > 0$.

By Hoeffding's inequality [8], for any $\delta_e > 0$, with probability at least $1 - \delta_e$:

$$\left|\hat{\mathbb{E}}_e[\ell(f^*)] - \mathbb{E}_e[\ell(f^*)]\right| \leq M\sqrt{\frac{\log(2/\delta_e)}{2n_e}}$$

Next, we analyze the relationship between test and training environments under anti-causal structure. By the optimality of $\phi_L^*$ and $\phi_H^*$, the environment independence property is satisfied: $\phi_H^*(\phi_L^*(\mathcal{X})) \perp\!\!\!\perp E \mid Y$. This implies that the distribution of high-level representations depends only on the label $Y$ and not on the environment $E$ when conditioned on $Y$.

This implies the optimal representation extracts the invariant causal mechanism $Y \rightarrow X$ while filtering out the spurious correlation $E \rightarrow X$. In other words, for any environments $e_1, e_2$ (including a test environment $e_{test}$), we have:

$$P_{e_1}(Z_H|Y) = P_{e_2}(Z_H|Y) = P(Z_H|Y)$$

where $Z_H = \phi_H^*(\phi_L^*(\mathcal{X}))$ is the high-level representation.

We now relate the expected loss in test environment to those in training environments. The expected loss in any environment $e$ can be written as:

$$\mathbb{E}_e[\ell(f^*)] = \int_{\mathcal{Y}} \int_{\mathcal{Z}_H} \ell(\mathcal{C}(z_H), y) \, dP(z_H|y) \, dP_e(y)$$

Due to the environment independence property, $P(z_H|y)$ is the same across all environments. Therefore, the differences in expected loss across environments arise only from differences in the label distribution $P_e(y)$. Define the maximum expected loss across training environments:

$$\mathbb{E}_{\max}[\ell(f^*)] = \max_{e \in \mathcal{E}} \mathbb{E}_e[\ell(f^*)]$$

Now, we bound the empirical risk in the test environment. For a test environment $e_{test}$, we have:

$$\hat{\mathbb{E}}_{e_{test}}[\ell(f^*)] = \mathbb{E}_{e_{test}}[\ell(f^*)] + \left(\hat{\mathbb{E}}_{e_{test}}[\ell(f^*)] - \mathbb{E}_{e_{test}}[\ell(f^*)]\right)$$

$$\leq \mathbb{E}_{e_{test}}[\ell(f^*)] + \left|\hat{\mathbb{E}}_{e_{test}}[\ell(f^*)] - \mathbb{E}_{e_{test}}[\ell(f^*)]\right|$$

Due to the anti-causal structure and the invariance property of the optimal representations, the expected loss in the test environment is related to the losses in training environments. Specifically,

since the learned representations capture the invariant causal mechanism from $Y$ to $X$ while removing environment-specific effects, we have:

$$\mathbb{E}_{e_{\text{test}}}[\ell(f^*)] \leq \mathbb{E}_{\max}[\ell(f^*)] + \epsilon_{\text{struct}}$$

where $\epsilon_{\text{struct}}$ is a small error arising from structural differences between test and training environments.

Finally, we apply union bound to combine the bounds. We set $\delta_e = \delta/|\mathcal{E}|$ for each training environment $e \in \mathcal{E}$ and $\delta_{\text{test}} = \delta/2$ for the test environment. By the union bound, with probability at least $1 - \delta$:

$$\hat{\mathbb{E}}_{e_{\text{test}}}[\ell(f^*)] \leq \mathbb{E}_{\max}[\ell(f^*)] + \epsilon_{\text{struct}} + M\sqrt{\frac{\log(4/\delta)}{2n_{\text{test}}}}$$

$$\leq \max_{e \in \mathcal{E}} \mathbb{E}_e[\ell(f^*)] + \max_{e \in \mathcal{E}} M\sqrt{\frac{\log(2|\mathcal{E}|/\delta)}{2n_e}} + \epsilon_{\text{struct}} + M\sqrt{\frac{\log(4/\delta)}{2n_{\text{test}}}}$$

For optimal representations that successfully enforce the invariance property, $\epsilon_{\text{struct}}$ approaches zero as the regularization strength is appropriately tuned. Under the assumption that the test environment follows the same anti-causal structure as the training environments, we can simplify to:

$$\hat{\mathbb{E}}_{e_{\text{test}}}[\ell(f^*)] \leq \max_{e \in \mathcal{E}} \mathbb{E}_e[\ell(f^*)] + O\left(\sqrt{\frac{\log(1/\delta)}{n_e}}\right)$$

$\square$

### D.3 Theorem 8 (Anti-Causal OOD Gap w.r.t. Kernels)

The below theorem implies that performance gaps cannot be smaller than the fundamental difference between the interventional distribution and observational distribution.

**Theorem 8** (Anti-Causal OOD Gap w.r.t. Kernels). *The gap between test performance and training performance w.r.t kernels is bounded as:*

$$|\mathbb{E}_{e_{test}}[\ell(f^*)] - \mathbb{E}_{e_{train}}[\ell(f^*)]| \geq \min_{e \in \mathcal{E}} \|K_{\{e\}}^{do(X)}(\omega) - K_{\{e\}}(\omega)\|_{\mathcal{H}} \tag{11}$$

*where $K_{\{e\}}^{do(X)}$ is the interventional kernel and $K_{\{e\}}$ is the observational kernel.*

*Proof.* We analyze the fundamental lower bound on the gap between test and training performance by relating it to the difference between interventional and observational distributions in causal settings.

First, we express expected losses in terms of probability distributions. For any environment $e$, the expected loss under the optimal predictor $f^* = \mathcal{C} \circ \phi_H^* \circ \phi_L^*$ can be written as:

$$\mathbb{E}_e[\ell(f^*)] = \int_{\Omega} \ell(f^*(\mathcal{X}(\omega)), Y(\omega)) \, d\mathbb{P}_e(\omega) \tag{12}$$

This expectation depends on the joint distribution of $(\mathcal{X}, Y)$ in environment $e$, which can be represented through the causal kernel $K_{\{e\}}$.

Then, we relate performance gap between test and training environments to distributional differences.

$$|\mathbb{E}_{e_{\text{test}}}[\ell(f^*)] - \mathbb{E}_{e_{\text{train}}}[\ell(f^*)]| = \left| \int_{\Omega} \ell(f^*(\mathcal{X}(\omega)), Y(\omega)) \, d(\mathbb{P}_{e_{\text{test}}} - \mathbb{P}_{e_{\text{train}}})(\omega) \right|$$

The measure $(\mathbb{P}_{e_{test}} - \mathbb{P}_{e_{train}})$ represents the difference between the joint probability distributions in test and training environments.

$$\int_{\Omega} f(\omega) \, d(\mathbb{P}_{e_{test}} - \mathbb{P}_{e_{train}})(\omega) = \int_{\Omega} \int_{\Omega'} f(\omega') \, d(K_{\{e_{test}\}}(\omega, d\omega') - K_{\{e_{train}\}}(\omega, d\omega')) \, d\mathbb{P}(\omega)$$

Now, we establish a dual representation using integral probability metrics. For a function class $\mathcal{H}$ containing functions of the form $h(\omega) = \ell(f^*(\mathcal{X}(\omega)), Y(\omega))$, we can express the distributional difference as an integral probability metric:

$$\|K_{\{e_{\text{test}}\}} - K_{\{e_{\text{train}}\}}\|_{\mathcal{H}} = \sup_{h \in \mathcal{H}} \left| \int h \, d(K_{\{e_{\text{test}}\}} - K_{\{e_{\text{train}}\}}) \right|$$

Here, we relate environmental differences to causal structure using interventional kernels. In the anti-causal setting, the causal structure implies that the observational distribution $K_{\{e\}}$ differs from the interventional distribution $K_{\{e\}}^{do(X)}$ due to the spurious correlation introduced by $E$. The difference $\|K_{\{e\}}^{do(X)} - K_{\{e\}}\|_{\mathcal{H}}$ quantifies the strength of this spurious correlation in environment $e$. Intuitively, if this difference is large, then $E$ has a strong influence on $X$ in that environment.

Next, we apply the variational characterization of integral probability metrics. For any function $h \in \mathcal{H}$, we have:

$$\left| \int h \, d(K_{\{e_{\text{test}}\}} - K_{\{e_{\text{train}}\}}) \right| \geq \min_{e \in \mathcal{E}} \left| \int h \, d(K_{\{e\}}^{do(X)} - K_{\{e\}}) \right|$$

This inequality holds because the interventional distribution $K_{\{e\}}^{do(X)}$ represents the distribution when the spurious correlation $E \to X$ is removed. The minimum across all training environments represents the smallest possible spurious effect that must be overcome in generalization.

Ultimately, we complete the proof using the specific loss function. Let $h^*(\omega) = \ell(f^*(\mathcal{X}(\omega)), Y(\omega))$ be the composed loss function. Then:

$$|\mathbb{E}_{e_{\text{test}}}[\ell(f^*)] - \mathbb{E}_{e_{\text{train}}}[\ell(f^*)]| = \left| \int h^* \, d(K_{\{e_{\text{test}}\}} - K_{\{e_{\text{train}}\}}) \right| \geq \min_{e \in \mathcal{E}} \left| \int h^* \, d(K_{\{e\}}^{do(X)} - K_{\{e\}}) \right|$$

Since this holds for the specific function $h^*$, and the integral probability metric takes the supremum over all $h \in \mathcal{H}$, we have:

$$|\mathbb{E}_{e_{\text{test}}}[\ell(f^*)] - \mathbb{E}_{e_{\text{train}}}[\ell(f^*)]| \geq \min_{e \in \mathcal{E}} \|K_{\{e\}}^{do(X)} - K_{\{e\}}\|_{\mathcal{H}}$$

This establishes the lower bound on the performance gap in terms of the minimum distance between interventional and observational kernels across training environments. $\square$

## D.4 Theorem 9 (Environmental Robustness)

Having established generalization bounds and performance gaps, we now examine how these translate to environmental robustness guarantees.

**Theorem 9** (Environmental Robustness). *Denote the learnt two-level representations by ACIA as $\phi_L^*$ and $\phi_H^*$. Then for any new environment $e_{new}$, the distributional distance $d_{\mathcal{H}}(\mathbb{P}_{e_{new}} \mathbb{P}_{\mathcal{E}})$ between $\mathbb{P}_{e_{new}}$ and $\mathbb{P}_{\mathcal{E}}$ over the function class $\mathcal{H}$ containing all predictors is bounded by*

$$d_{\mathcal{H}}(\mathbb{P}_{e_{new}} \mathbb{P}_{\mathcal{E}}) \leq \delta_1 + \delta_2,$$

*where $\mathbb{P}_{\mathcal{E}} = \frac{1}{|\mathcal{E}|} \sum_{e \in \mathcal{E}} \mathbb{P}_e$ is the mixture distribution of training environments $\mathcal{E}$; $\delta_1$ and $\delta_2$ respectively measure the degree of invariance violation in below conditions 1 and 2.*

1. *The high-level representation is environment-independent: $\phi_H(\phi_L^*(\mathcal{X})) \perp E \mid Y$*

2. *The low-level representation is invariant: $\Pr(\phi_L^*(\mathcal{X}) \mid Y)$ is constant across environments*

*Proof.* Let $\mathbb{P}_{\mathcal{E}} = \frac{1}{|\mathcal{E}|} \sum_{e \in \mathcal{E}} \mathbb{P}_e$ be the mixture distribution of training environments $\mathcal{E}$, and $\mathbb{P}_{e_{new}}$ be the distribution of a new environment $e_{new}$. Let $\mathcal{H}$ be the function class containing all predictors of the form $h = g \circ \phi_H^* \circ \phi_L^*$ for some measurable function $g$. The distributional distance $d_{\mathcal{H}}(\mathbb{P}_{e_{new}}, \mathbb{P}_{\mathcal{E}})$ is defined as an integral probability metric (IPM) [51]:

$$d_{\mathcal{H}}(\mathbb{P}_{e_{new}}, \mathbb{P}_{\mathcal{E}}) = \sup_{h \in \mathcal{H}} |\mathbb{E}_{e_{new}}[h] - \mathbb{E}_{\mathcal{E}}[h]|$$

Next, we express expectations in terms of representations. For any function $h \in \mathcal{H}$, we can express expectations in terms of the learned representations:

$$\mathbb{E}_e[h] = \int_{\mathcal{Y}} \int_{\mathcal{Z}_H} h(z_H) \, d\mathbb{P}_e(z_H|y) \, d\mathbb{P}_e(y)$$

where $z_H = \phi_H^*(\phi_L^*(\mathcal{X}))$ is the high-level representation.

By condition (1), the high-level representation is environment-independent given $Y$: $\phi_H^*(\phi_L^*(\mathcal{X})) \perp E \mid Y$, which means for any environments $e_1, e_2$ (including a new environment $e_{new}$):

$$\mathbb{P}_{e_1}(z_H|y) = \mathbb{P}_{e_2}(z_H|y).$$

Assume certain level of violation on condition (1), we rewrite

$$\mathbb{P}_{e_1}(z_H|y) = \mathbb{P}_{e_2}(z_H|y) + \Delta(z_H, y, e_1, e_2)$$

where $\Delta(z_H, y, e_1, e_2)$ is a function quantifying the violation of perfect environment independence, with $\|\Delta\|_\infty \leq \delta_1$ for some small $\delta_1 \geq 0$ that depends on the degree to which property (1) is satisfied.

By condition (2), the low-level representation is invariant, $\Pr(\phi_L^*(\mathcal{X}) \mid Y)$ is constant across environments. This implies the distribution of $z_L = \phi_L^*(\mathcal{X})$ conditioned on $Y$ is same across environments. Assume the degree of violation on condition (2) is $\delta_2$.

We now bound the difference in expectations. For any $h \in \mathcal{H}$, the difference in expectations between environments is:

$$|\mathbb{E}_{e_{new}}[h] - \mathbb{E}_{\mathcal{E}}[h]| = \left| \int_{\mathcal{Y}} \int_{\mathcal{Z}_H} h(z_H)\, d\mathbb{P}_{e_{new}}(z_H|y)\, d\mathbb{P}_{e_{new}}(y) - \int_{\mathcal{Y}} \int_{\mathcal{Z}_H} h(z_H)\, d\mathbb{P}_{\mathcal{E}}(z_H|y)\, d\mathbb{P}_{\mathcal{E}}(y) \right|$$

$$= \left| \int_{\mathcal{Y}} \int_{\mathcal{Z}_H} h(z_H)\, d(\mathbb{P}_{e_{new}}(z_H|y) - \mathbb{P}_{\mathcal{E}}(z_H|y))\, d\mathbb{P}_{e_{new}}(y) \right.$$

$$\left. + \int_{\mathcal{Y}} \int_{\mathcal{Z}_H} h(z_H)\, d\mathbb{P}_{\mathcal{E}}(z_H|y)\, d(\mathbb{P}_{e_{new}}(y) - \mathbb{P}_{\mathcal{E}}(y)) \right|$$

By the triangle inequality:

$$|\mathbb{E}_{e_{new}}[h] - \mathbb{E}_{\mathcal{E}}[h]| \leq \left| \int_{\mathcal{Y}} \int_{\mathcal{Z}_H} h(z_H)\, d(\mathbb{P}_{e_{new}}(z_H|y) - \mathbb{P}_{\mathcal{E}}(z_H|y))\, d\mathbb{P}_{e_{new}}(y) \right|$$

$$+ \left| \int_{\mathcal{Y}} \int_{\mathcal{Z}_H} h(z_H)\, d\mathbb{P}_{\mathcal{E}}(z_H|y)\, d(\mathbb{P}_{e_{new}}(y) - \mathbb{P}_{\mathcal{E}}(y)) \right|$$

We apply the bounds on invariance violations. From condition (1), for each $y \in \mathcal{Y}$, we have:

$$\|\mathbb{P}_{e_{new}}(z_H|y) - \mathbb{P}_{\mathcal{E}}(z_H|y)\|_{TV} \leq \delta_1$$

where $\|\cdot\|_{TV}$ denotes the total variation distance.

Using condition (2) and the anti-causal structure, the difference in label distributions is also bounded:

$$\|\mathbb{P}_{e_{new}}(y) - \mathbb{P}_{\mathcal{E}}(y)\|_{TV} \leq \delta_2$$

To bridge the theoretical and practical results gap, we apply Hölder's inequality to bound the expectation difference. Assuming bounded functions $\|h\|_\infty \leq 1$, and using Hölder's inequality:

$$|\mathbb{E}_{e_{new}}[h] - \mathbb{E}_{\mathcal{E}}[h]| \leq \int_{\mathcal{Y}} \|h\|_\infty \cdot \|\mathbb{P}_{e_{new}}(z_H|y) - \mathbb{P}_{\mathcal{E}}(z_H|y)\|_{TV}\, d\mathbb{P}_{e_{new}}(y) + \|h\|_\infty \cdot \|\mathbb{P}_{e_{new}}(y) - \mathbb{P}_{\mathcal{E}}(y)\|_{TV}$$

$$\leq \delta_1 \cdot \int_{\mathcal{Y}} d\mathbb{P}_{e_{new}}(y) + \delta_2 = \delta_1 + \delta_2$$

By taking the supremum over function class $\mathcal{H}$, we have:

$$d_{\mathcal{H}}(\mathbb{P}_{e_{new}}, \mathbb{P}_{\mathcal{E}}) = \sup_{h \in \mathcal{H}} |\mathbb{E}_{e_{new}}[h] - \mathbb{E}_{\mathcal{E}}[h]| \leq \delta_1 + \delta_2.$$

This establishes that the distributional discrepancy between any new environment and the mixture of training environments is bounded by a term that depends only on how well these invariance properties are satisfied w.r.t the learnt two-level representations by ACIA. □

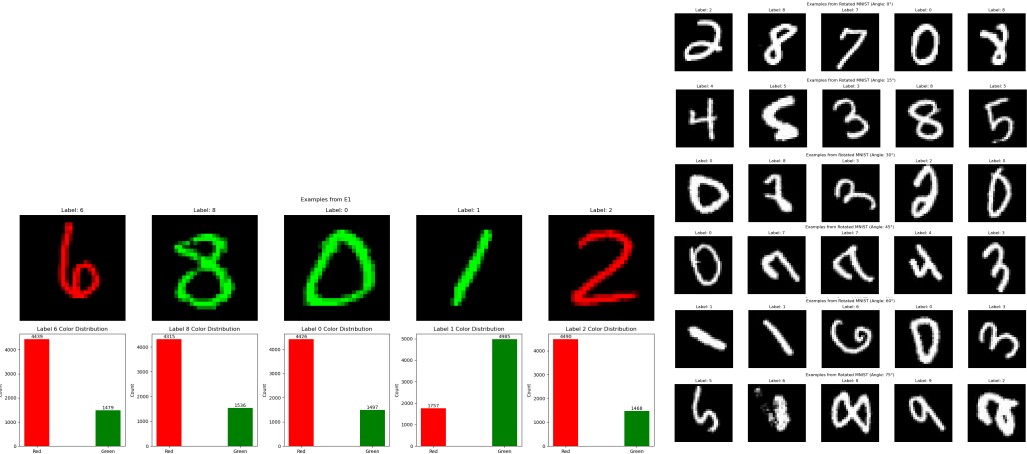

Figure 7: MNIST-based anti-causal datasets: (Left) CMNIST by color; (Righ) RMNIST by rotation.

# E  Experimental Setup and Results

## E.1  Datasets for Anti-Causal Learning

### E.1.1  Colored MNIST (CMNIST)

Let $Y \in \{0, 1, ..., 9\}$ be the digit label, $X_0 \in \mathbb{R}^{28 \times 28}$ be the original grayscale MNIST image, $E \in \{e_1, e_2\}$ be the environment variable, and $C \in \{R, G\}$ be the color variable (Red or Green). The generation process is as follows:

1. Sample $(Y, X_0)$ from the original MNIST dataset.

2. Generate the color $C$ based on $Y$ and $E$:

$$P(C = R|Y, E) = \begin{cases} 0.75 & \text{if } Y \text{ is even and } E = e_1 \\ 0.25 & \text{if } Y \text{ is odd and } E = e_1 \\ 0.25 & \text{if } Y \text{ is even and } E = e_2 \\ 0.75 & \text{if } Y \text{ is odd and } E = e_2 \end{cases}$$

3. Generate the colored image $X \in \mathbb{R}^{28 \times 28 \times 3}$:
   If $C = R$: $X[:, :, 0] = X_0, X[:, :, 1] = 0, X[:, :, 2] = 0$
   If $C = G$: $X[:, :, 0] = 0, X[:, :, 1] = X_0, X[:, :, 2] = 0$

**Why is this an anti-causal problem?**

1. The digit label $Y$ causes the color $C$, which is part of the observed features $X$. This is opposite to the typical prediction task where we try to predict $Y$ from $X$.

2. The underlying causal graph is: $Y \rightarrow X \leftarrow E$. This is a classic anti-causal structure where the label $Y$ is a cause of the features $X$, and the environment $E$ influences $X$.

**Preparation of the Colored MNIST**

According to Table 6, environment $e_1$ contains 60,000 images while $e_2$ contains 10,000 images, maintaining roughly a 6:1 ratio. A clear pattern emerges in the color distribution: even digits (0,2,4,6,8) in $e_1$ are predominantly colored red ( 75%), while odd digits (1,3,5,7,9) are predominantly green ( 75%). This pattern is inverted in $e_2$, where even digits are mostly green ( 75%) and odd digits are mostly red ( 75%). Despite these strong digit-color correlations within each environment, the overall color distribution remains relatively balanced in both environments - $e_1$ has 49.54% red and 50.46% green, while $e_2$ has 51.22% red and 48.78% green.

This design creates a spurious correlation between digits and colors that varies across environments, making it an ideal dataset for testing anti-causal representation learning methods.

Table 6: Distribution of digits and colors in Colored MNIST Environments $e_1$ and $e_2$

| Digit | Environment $e_1$ | | | Environment $e_2$ | | |
|---|---|---|---|---|---|---|
| | Count | Red | Green | Count | Red | Green |
| 0 | 5923 | 4426 (74.73%) | 1497 (25.27%) | 980 | 284 (28.98%) | 696 (71.02%) |
| 1 | 6742 | 1757 (26.06%) | 4985 (73.94%) | 1135 | 881 (77.62%) | 254 (22.38%) |
| 2 | 5958 | 4490 (75.36%) | 1468 (24.64%) | 1032 | 280 (27.13%) | 752 (72.87%) |
| 3 | 6131 | 1548 (25.25%) | 4583 (74.75%) | 1010 | 753 (74.55%) | 257 (25.45%) |
| 4 | 5842 | 4365 (74.72%) | 1477 (25.28%) | 982 | 244 (24.85%) | 738 (75.15%) |
| 5 | 5421 | 1382 (25.49%) | 4039 (74.51%) | 892 | 658 (73.77%) | 234 (26.23%) |
| 6 | 5918 | 4439 (75.01%) | 1479 (24.99%) | 958 | 239 (24.95%) | 719 (75.05%) |
| 7 | 6265 | 1561 (24.92%) | 4704 (75.08%) | 1028 | 782 (76.07%) | 246 (23.93%) |
| 8 | 5851 | 4315 (73.75%) | 1536 (26.25%) | 974 | 222 (22.79%) | 752 (77.21%) |
| 9 | 5949 | 1438 (24.17%) | 4511 (75.83%) | 1009 | 779 (77.21%) | 230 (22.79%) |
| Total | 60000 | 29721 (49.54%) | 30279 (50.46%) | 10000 | 5122 (51.22%) | 4878 (48.78%) |

**How to create the causal space from CMNIST**

1. We first define the sample space $\Omega$:

$$\Omega = \{(Y, X_0, E, C, X) \mid Y \in \{0, \ldots, 9\}, \mathbf{V}_{L_0} \in \mathbb{R}^{28 \times 28}, E \in \{e_1, e_2\}, C \in \{R, G\}, \mathbf{V}_L \in \mathbb{R}^{28 \times 28 \times 3}\}$$

2. We then define the $\sigma$-algebras. $\mathscr{H}_{\mathbf{V}_L}$: $\sigma$-algebra on $\mathbf{V}_L$ (the colored images). and $\mathscr{H}_Y$: $\sigma$-algebra on $Y$ (the digit labels).

3. Then define environments and datasets, causal spaces for each environment, and the product causal space based on Alg.1.

4. Next, we construct the causal kernel $\mathbb{K}$: For the anti-causal CMNIST, construct $\mathbb{K}$ to reflect the causal structure $Y \to X \leftarrow E$. For example, for just one environment $e_i$ (so not dependant on $e_i$ in the formula anymore):

$$K_S(\omega, A) = \begin{cases} P(C = R \mid Y = y) & \text{if } A \text{ corresponds to red images} \\ P(C = G \mid Y = y) & \text{if } A \text{ corresponds to green images} \end{cases}$$

where $P(C = R \mid Y = y)$ is given by the generation process:

- 0.75 if $Y$ is even and $E = e_1$, or $Y$ is odd and $E = e_2$
- 0.25 if $Y$ is odd and $E = e_1$, or $Y$ is even and $E = e_2$

For $S \in \mathscr{P}(T)$, $\omega \in \Omega$, and $A \in \mathscr{H}$:

$$K_S(\omega, A) = P(X \in A \mid Y = y, E \in S)$$

where $y$ is the component of $\omega$ corresponding to $Y$, and $E \in S$ means the environments indexed by $S$. So, in general,

$$K_S(\omega, A) = \begin{cases} P(C = R \mid Y = y, \Omega_{e_1} \times \Omega_{e_2}) \text{ if A corresponds to red images} \\ P(C = G \mid Y = y, \Omega_{e_1} \times \Omega_{e_2}) \text{ if A corresponds to green images} \end{cases}$$

5. For causal properties verification, we first check if $K_S(\omega, \{A|B\}) = K_S(\omega', \{A|B\})$ for all $\omega, \omega' \in \Omega$, $A \in \mathscr{H}_{\mathbf{V}_L}$, $B \in \mathscr{H}_Y$, $S \in \mathscr{P}(T)$ by comparing conditional probabilities across environments. This is verifying that the causal kernel $K_S$ is independent of the specific outcome $\omega$ when we condition on Y. It's checking if $P(X|Y)$ is the same for all instances, regardless of the environment, which reflects the anti-causal nature where Y causes X.

6. The we verify key properties:
   - $\Pr(\phi_L(\mathbf{V}_L) \mid Y)$ is invariant across $S \in \mathscr{P}(T)$
   - $\phi_H(\phi_L(\mathbf{V}_L)) \perp E \mid Y$

   To ensure the construction aligns with the anti-causal nature:
   - $K_S(\omega, \{Y \in B \mid X = x\})$ should be invariant across environments
   - $K_S^{do(X)}(\omega, \{Y \in B\}) = K_S(\omega, \{Y \in B\})$
   - $K_S^{do(Y)}(\omega, \{X \in A\}) \neq K_S(\omega, \{X \in A\})$

### E.1.2 Rotated MNIST (RMNIST)

Let $Y \in \{0, 1, ..., 9\}$ be the digit label, $X_0 \in \mathbb{R}^{28 \times 28}$ be the original grayscale MNIST image, $E \in \{0°, 15°, 30°, 45°, 60°, 75°\}$ be the environment variable, representing the rotation angle, and $R_\theta : \mathbb{R}^{28 \times 28} \to \mathbb{R}^{28 \times 28}$ be a rotation function that rotates an image by $\theta$ degrees. The generation process is as follows:

1. Sample $(Y, X_0)$ from the original MNIST dataset.
2. For each environment $E = \theta$, generate the rotated image: $X_\theta = R_\theta(X_0)$

To define the causal kernel $\mathbb{K}$: For RMNIST, construct $\mathbb{K}$ to reflect the causal chain:

$$K_S(\omega, A) = P(X_\theta \in A \mid Y = y, E = \theta)$$

where $\theta$ is the rotation angle from environment $E$.

**Why is this an anti-causal problem?**

1. The original relationship $Y \to X_0$ is anti-causal, as the digit label causes the original image.
2. The underlying causal graph is: $Y \to X_0 \to X_\theta \leftarrow E$, where rotation angle $E$ influences the final image.

According to Table 7, distribution of digits and rotations in RMNIST Environments $e_1$ and $e_2$ is observable. The task of predicting $Y$ from $X_\theta$ can be seen as an anti-causal problem (predicting cause from effect), but with an additional causal intervention ($E$) applied to the effect.

Table 7: Distribution of digits and rotations in Rotated MNIST Environments $e_1$ and $e_2$

| Digit | Environment $e_1$ | | | Environment $e_2$ | | |
|---|---|---|---|---|---|---|
| | Count | 15° | 75° | Count | 15° | 75° |
| 0 | 5923 | 4442 (75.00%) | 1481 (25.00%) | 980 | 245 (25.00%) | 735 (75.00%) |
| 1 | 6742 | 1686 (25.00%) | 5056 (75.00%) | 1135 | 851 (75.00%) | 284 (25.00%) |
| 2 | 5958 | 4469 (75.00%) | 1489 (25.00%) | 1032 | 258 (25.00%) | 774 (75.00%) |
| 3 | 6131 | 1533 (25.00%) | 4598 (75.00%) | 1010 | 758 (75.00%) | 252 (25.00%) |
| 4 | 5842 | 4382 (75.00%) | 1460 (25.00%) | 982 | 246 (25.00%) | 736 (75.00%) |
| 5 | 5421 | 1355 (25.00%) | 4066 (75.00%) | 892 | 669 (75.00%) | 223 (25.00%) |
| 6 | 5918 | 4439 (75.00%) | 1479 (25.00%) | 958 | 240 (25.00%) | 718 (75.00%) |
| 7 | 6265 | 1566 (25.00%) | 4699 (75.00%) | 1028 | 771 (75.00%) | 257 (25.00%) |
| 8 | 5851 | 4388 (75.00%) | 1463 (25.00%) | 974 | 244 (25.00%) | 730 (75.00%) |
| 9 | 5949 | 1487 (25.00%) | 4462 (75.00%) | 1009 | 757 (75.00%) | 252 (25.00%) |
| Total | 60000 | 29747 (49.58%) | 30253 (50.42%) | 10000 | 5039 (50.39%) | 4961 (49.61%) |

### E.1.3 Ball Agent

Following [9], let $Y \in \mathbb{R}^{2n}$ be ball positions (where $n$ is the number of balls), $X \in \mathbb{R}^{64 \times 64 \times 3}$ be the rendered image, and $E$ be the set of sparse interventions. The generation process is:

1. Sample initial ball positions $Y = [y_1, ..., y_{2n}]$ where each $y_i \sim U(0.1, 0.9)$.
2. Generate the base image $X$ by rendering colored balls at positions $Y$.
3. For each environment $e \in E$, apply sparse perturbations to specific ball coordinates.

**Why is this an anti-causal problem?**

1. Ball positions $Y$ cause the image appearance $X$
2. Interventions $E$ modify positions but not the underlying rendering process
3. The causal graph is: $Y \to X \leftarrow E$

**How to create the causal space from Ball Environment**

1. Define the sample space $\Omega$:

$$\Omega = (Y, E, X) \mid Y \in [0, 1]^{2n}, E \in \mathcal{P}(1, ..., 2n), X \in \mathbb{R}^{64 \times 64 \times 3}$$

where $\mathcal{P}(1, ..., 2n)$ represents possible intervention subsets.

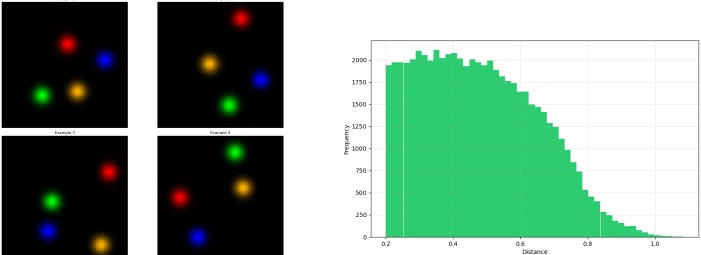

Figure 8: Visualization of the Ball Agent environment with 4 balls and 10,000 samples. (Left) Example configurations showing the colored balls spatial distribution. (Right) Analysis of inter-ball distances for the environment spatial constraints

2. Define the causal kernel $\mathbb{K}$:

$$K_S(\omega, A) = P(X \in A \mid Y = y, E = e)$$

where $e$ represents specific intervention coordinates.

3. For sparse interventions, we should verify that:

- Changes in $X$ are localized to intervened coordinates
- Non-intervened ball positions remain unchanged
- $K_S^{do(Y_i)}(\omega, \{X \in A\}) \neq K_S(\omega, \{X \in A\})$ for intervened coordinates

Fig.8 shows that the Ball Agent dataset demonstrates a remarkably balanced intervention structure across 10,000 samples with 4 balls. Each ball receives interventions approximately 50% of the time (ranging from 49.7% to 50.8%), indicating uniform intervention probability. The distribution of simultaneous interventions is uniform, with each possible number of interventions (0 to 4 balls) occurring in roughly equal proportions (19.6% to 20.4%). This uniformity is reflected in the high intervention pattern entropy of 1.609 bits, approaching the theoretical maximum for this scenario. Spatially, the balls maintain a minimum distance of 0.200 units, with a mean distance of 0.478 units between pairs, demonstrating effective enforcement of proximity constraints while allowing significant positional variation. This structured randomness makes the dataset particularly suitable for studying anti-causal relationships between ball positions, rendered images, and interventions.

**Intervention types**

In the Ball Agent environment, intervention types are categorized according to the number of ball coordinates simultaneously affected:

- **None**: No intervention applied to any ball coordinates ($|I| = 0$)
- **Single**: Intervention affects exactly one ball coordinate ($|I| = 1$)
- **Double**: Interventions affect exactly two ball coordinates ($|I| = 2$)
- **Multiple**: Interventions affect three or more ball coordinates ($|I| \geq 3$)

These intervention patterns follow the intervention distribution defined in [9], where the intervention set $I \subseteq \{1, 2, ..., 2n\}$ represents the indices of coordinates receiving interventions. The visualization reveals how ACIA's learned representations are organized based on these intervention patterns, demonstrating ACIAl's ability to identify invariant features despite varying intervention complexity.

### E.1.4 Camelyon17

Let $Y \in 0, 1$ be the tumor label, $X \in \mathbb{R}^{96 \times 96 \times 3}$ be the tissue image patch, and $E \in h_1, h_2, h_3, h_4, h_5$ be the hospital identifier. The generation process follows the natural data collection:

1. Sample tumor tissue ($Y = 1$) or normal tissue ($Y = 0$)
2. Process tissue according to hospital protocol $E = h_i$
3. Generate patch $X$ from the processed tissue slide

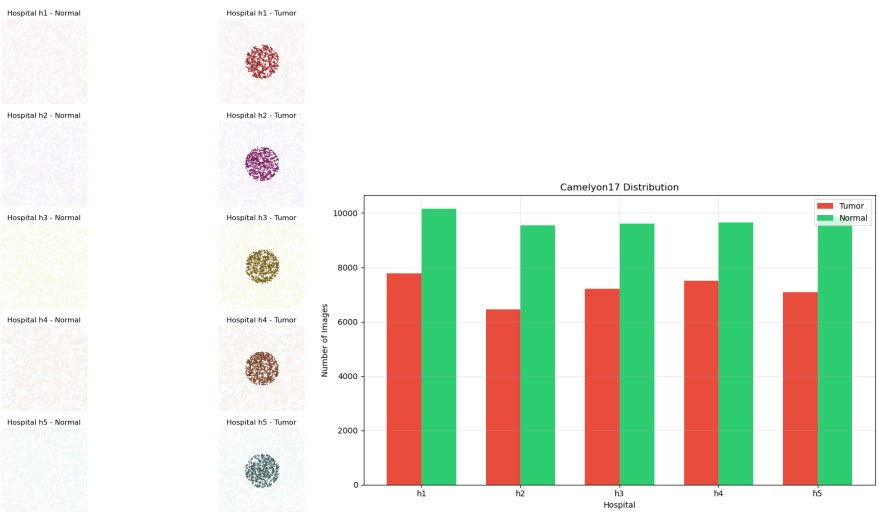

Figure 9: Camelyon17 shows hospital-specific tissue imaging variations. (Left) Examples of normal and tumor tissue images across five hospitals (h1-h5), illustrating distinct staining patterns. (Right) Distribution of normal and tumor samples across hospitals, showing the dataset's balanced nature.

**Why is this an anti-causal problem?**

1. The presence of tumor ($Y$) causes specific visual patterns in the tissue ($X$).

2. Hospital scanning and staining protocols ($E$) affect the image appearance but not the tumor status.

3. The task of predicting tumor presence from images is inherently anti-causal. The causal graph is: $Y \rightarrow X \leftarrow E$

**How to create the causal space from Camelyon17**

1. Define the sample space $\Omega$:
$$\Omega = (Y, E, X) \mid Y \in 0, 1, E \in h_1, ..., h_5, X \in \mathbb{R}^{96 \times 96 \times 3}$$

2. Define the causal kernel $\mathbb{K}$ for each hospital:
$$K_S(\omega, A) = P(X \in A \mid Y = y, E = h_i)$$

where $h_i$ represents the specific hospital protocol.

Table 8: Distribution of images across hospitals in Camelyon17 dataset

| Hospital | Total Images | Tumor | Normal |
|----------|--------------|-------|--------|
| $h_1$ | 17,934 | 7,786 (43.41%) | 10,148 (56.59%) |
| $h_2$ | 15,987 | 6,446 (40.32%) | 9,541 (59.68%) |
| $h_3$ | 16,828 | 7,212 (42.86%) | 9,616 (57.14%) |
| $h_4$ | 17,155 | 7,502 (43.73%) | 9,653 (56.27%) |
| $h_5$ | 16,960 | 7,089 (41.80%) | 9,871 (58.20%) |
| Total | 84,864 | 36,035 (42.46%) | 48,829 (57.54%) |

Visualization of Camelyon17 and its distribution is in Fig.9, which exhibits systematic variations in tissue imaging across 5 hospitals while maintaining consistent tumor detection challenges. Each hospital maintains a relatively balanced distribution between tumor and normal samples, with tumor prevalence ranging from 40.32% (h2) to 43.73% (h4). The dataset encompasses a substantial total of 84,864 images, with individual hospital contributions ranging from 15,987 to 17,934 samples. These statistics are available in Table 8. The hospital-specific staining patterns are visually distinct, as evidenced by the color variations in the example images, yet the underlying tumor patterns remain consistent. This structure makes Camelyon17 an example of an anti-causal learning problem, where hospital-specific imaging protocols (E) affect the image appearance (X) but not the ground truth tumor status (Y).

For the Camelyon17 results (Figure 6), the fourth panel shows a confidence visualization that maps the model's certainty levels across the representation space. The confidence levels are color-coded as:

- **Green/Yellow**: Regions where the model maintains high confidence in tumor classification
- **Orange**: Intermediate confidence regions, typically at class boundaries
- **Red**: Low confidence regions requiring additional evidence for reliable classification

This uncertainty quantification follows [14] and extends the predictive uncertainty estimation. The visualization reveals how ACIA preserves diagnostically relevant tissue features while discarding hospital-specific staining variations, which aligns with findings [55] on stain normalization effects in computational pathology. This confidence mapping is particularly important for medical applications where uncertainty awareness is critical for clinical decision support.

## E.2 Model Architecture and Hyperparameter Setting

Table 9: Dataset configuration details and their properties.

| Dataset | Type | Environments | Dimensionality | Label Space | Spurious Corr. | Train Size | Test Size |
|---|---|---|---|---|---|---|---|
| CMNIST | Synthetic | 2 ($e_1$, $e_2$) | 28x28x3 | {0,...,9} | Color-digit | 60,000 | 10,000 |
| RMNIST | Synthetic | 2 train (15°, 75°), 3 test (30°, 45°, 60°) | 28x28x3 | {0,...,9} | Rotation-digit | 60,000 | 10,000 |
| Ball Agent | Synthetic | 4 balls with interventions | 64x64x3 | $[0, 1]^{2n}$ | Coord-coupling | 15,000 | 5,000 |
| Camelyon17 | Real | 3 train (hospitals 0-2), 2 test (hospitals 3-4) | 96x96x3 | {0,1} | Hospital-stain | 50,916 | 33,944 |

We provide detailed hyperparameter settings for ACIA across all datasets to ensure reproducibility. We discuss the datasets properties in Table 9.

**Model architectures:** We design the architecture with the following principles: the low-level representation should be sufficiently rich to capture both label-relevant and environment-dependent signals, while the high-level representation serves as an information bottleneck, filtering out environment-dependent variations, and thus is set to a lower dimension than the low-level representation.

We used consistent network architectures across all experiments in our ACIA. For low-level representation learner, we used ConvNet (3 layers) + FC layers $\rightarrow$ 32/256-dim latent space. For high-level representation learner, we used MLP (2 layers) $\rightarrow$ 128-dim latent space. For the on-top classifier, we used linear layer (output dimension varies by dataset).

**Hyperparameter setting:** For all datasets, the batch size is 32, the optimizer is Adam, the learning rate is 1e-4, and we use early stopping for Camelyon17 to avoid overfitting in the results. In our regularier, we chose $\lambda_1$ as $0.1/\sqrt{\text{batch\_size}}$ ($\approx 0.0177$) for CMNIST, RMNIST, and Ball Agent, and chose 0.5 for Camelyon17. In addition, we chose $\lambda_2$ as $0.5/\sqrt{\text{batch\_size}}$ ($\approx 0.0884$) for CMNIST, RMNIST, and Ball Agent, and chose 0.1 for Camelyon17. These hyperparameters were optimized based on validation performance and theoretical constraints from the ACIA framework requiring balanced regularization to achieve environment invariance without sacrificing predictive performance.

## E.3 Imperfect Intervention Implementation

### E.3.1 Imperfect Intervention Datasets Construction

In **CMNIST**, perfect interventions implement deterministic color assignment with probabilities restricted to be 0 or 1, creating a strict mapping between digits and colors. Imperfect interventions use continuous probabilities in [0,1], allowing for partial influence of the causal mechanism.

In **RMNIST**, perfect interventions apply fixed and deterministic rotation angles for each digit class. Imperfect interventions introduce variability through probabilistic angle distributions, creating a blend between digit-specific and environment-specific rotations.

In **Ball Agent**, perfect interventions produce deterministic position shifts, completely overriding natural ball dynamics. Imperfect interventions implement probabilistic dynamics changes, where original positions are partially preserved while incorporating intervention effects.

In **Camelyon17**, perfect interventions standardize tumor appearance through complete staining changes. Imperfect interventions apply partial staining changes, preserving some hospital-specific characteristics while normalizing others.

### E.3.2 Results on Imperfect Intervention

The perfect and imperfect interventions both achieve accuracy values that exceed 99%. The low values for environment independence and low-level invariance metrics indicate a successful disentanglement of label information from environment factors. Intervention robustness remains low for both types (0.025-0.028), suggesting that the model maintains consistent predictions under small perturbations regardless of the intervention mechanism. This supports the theoretical assertion that ACIA's measure-theoretic framework can effectively handle both perfect and imperfect interventions through its interventional kernel formulation.

For CMNIST, perfect interventions completely break the probabilistic relationship between digits and colors by implementing deterministic coloring. For example:

$$P(Color = red|Digit = even, Env = e_1) = 1.0 \tag{13}$$

$$P(Color = red|Digit = odd, Env = e_1) = 0.0 \tag{14}$$

Imperfect interventions preserve partial dependencies by blending original probabilities with the target values ($\alpha = 0.5$ in our experiment.):

$$P(Color = red|Digit, Env) = 0.5 \cdot P_{original}(Color = red|Digit, Env)$$
$$+ 0.5 \cdot P_{perfect}(Color = red|Digit, Env) \tag{15}$$

**RMNIST:** Table 2 shows ACIA's effectiveness with rotation-based interventions. Perfect interventions yield a test accuracy of 99.1%, marginally outperforming imperfect interventions (99.0%). The environment independence metric shows the model's ability to progressively remove rotation-specific information, with perfect interventions achieving better invariance (0.455 vs. 0.480). This confirms that deterministic angle assignments facilitate more complete abstraction of spurious correlations.

In RMNIST, perfect interventions implement precise angle assignments for digit classes:

$$\theta_{\text{perfect}}(d) = \begin{cases} \theta_{\text{base}} & \text{if } d \text{ is even} \\ \theta_{\text{base}} + 45° & \text{if } d \text{ is odd} \end{cases} \tag{16}$$

Imperfect interventions implement probabilistic angle distributions:

$$\theta_{\text{imperfect}}(d) = (1 - \alpha) \cdot \theta_{\text{base}} + \alpha \cdot \theta_{\text{perfect}}(d) \tag{17}$$

where $\alpha = 0.5$ represents the intervention strength.

The training dynamics reveal that R1 values (environment independence) decrease from initial values around 10.0 to final values of 0.46-0.48, while R2 values (causal structure alignment) decrease from 0.21 to 0.02-0.03. This convergence pattern quantitatively demonstrates ACIA's ability to progressively disentangle digit identity from rotation angle through its measure-theoretic framework.

**Ball Agent:** Table 2 demonstrates ACIA's exceptional performance in a continuous spatial regression task. Extended training reveals near-perfect position accuracy of 99.97% (position error 0.0003) for both perfect and imperfect interventions. The environment independence metric shows perfect interventions achieve substantially better invariance (0.372 vs. 0.468), confirming that deterministic position shifts enable more complete abstraction of intervention-specific effects.

In Ball Agent, perfect interventions implement deterministic position transformations:

$$P_{\text{perfect}}(x_i, y_i) = \begin{cases} (x_i, y_i) & \text{if no intervention} \\ (x_i', y_i') & \text{if intervention on coordinate} \end{cases} \tag{18}$$

Imperfect interventions implement probabilistic position shifts:

$$P_{\text{imperfect}}(x_i, y_i) = (1 - \alpha) \cdot (x_i, y_i) + \alpha \cdot (x_i', y_i') \tag{19}$$

where $\alpha = 0.5$ represents the intervention strength.

The R1 values are steadily decreasing from initial values of 1.02-1.20 to final values of 0.372-0.468, while R2 values decrease from 0.38-0.44 to 0.024-0.026. This progressive convergence pattern demonstrates ACIA's ability to effectively generalize from discrete classification to continuous regression tasks while achieving exceptional invariance properties and near-perfect positional accuracy.

Table 10: Runtime performance of ACIA across different datasets

| Dataset | #Samples | Input Dimension | Time (seconds) |
|---|---|---|---|
| CMNIST | 60K | 2,352 | 580.7 |
| RMNIST | 60K | 784 | 973.6 |
| Ball Agent | 15K | 12,288 | 204.7 |
| Camelyon17 | 84K | 6,912 | 799.6 |

### E.4 Computational Efficiency

Table 10 reports the runtime on a laptop with a single GPU (3.3 GHz, 32 GB RAM), using precision $\epsilon = 0.01$ and failure probability $\delta = 0.05$.

ACIA completes training in approximately 3–16 minutes under these settings, demonstrating practical scalability for real-world applications.

# F Discussion

### F.1 Why Measure Theory?

Traditional causal inference relies on Directed Acyclic Graphs (DAGs) and Structural Causal Models (SCMs), which require explicit specification of causal graph structure, parametric forms for structural equations, and knowledge of which variables are observed/latent.

These requirements are prohibitive when learning representations from raw data. The measure-theoretic framework offers three key advantages:

*(1) Direct learning from observations:* Probability kernels $K(\omega, A)$ encode causal mechanisms without requiring explicit structural equations, enabling end-to-end learning from $X$ to latent causal variables.

*(2) Unified intervention handling:* Both perfect ($\mathbb{Q}(A|\omega) = \mathbb{Q}(A)$) and imperfect ($\mathbb{Q}(A|\omega)$ depends on $\omega$) interventions are expressed through kernel modifications, avoiding the restrictive perfect intervention assumption.

*(3) Compositional structure:* Tensor product $\sigma$-algebras $\mathscr{H}_e = \otimes_{t \in T_e} \mathscr{A}_t$ naturally represent factorized causal structures without specifying DAG edges, allowing flexible architecture design.

### F.2 Why Kernel Formulation?

Interventional kernels play a pivotal role in distinguishing between causal and anti-causal relationships by capturing how distributions respond to interventions. In anti-causal structures, intervening on $X$ does not affect $Y$ (as shown in Corollary 2), while intervening on $Y$ changes the distribution of $X$. Interventional kernels formalize the counterfactual question: "How would features $X$ change if we intervened on label $Y$?" This enables learning representations that capture the generative process from $Y$ to $X$. In addition, the interventional kernel formulation enables both perfect and imperfect interventions, which is critical since real-world interventions are rarely perfect. The kernel independence property also helps separate the mechanisms $P(X|Y)$ from the environmental influences $P(X|E)$. This formalization through interventional kernels provides a rigorous foundation for causal representation learning that goes beyond mere statistical association.

### F.3 Connection to Existing Methods

Our measure-theoretic approach connects to and extends several several existing frameworks. For instance, under linear representations and squared loss, the optimization problem in Eqn.5 is an extended version of Invariant Risk Minimization (IRM) [5] that includes two levels of invariance:

$$\min_{\phi_L, \phi_H, \mathcal{C}} \sum_{e \in \mathcal{E}} R^e(\mathcal{C} \circ \phi_H \circ \phi_L) \quad \text{s.t.} \begin{cases} \phi_H \circ \phi_L \in \arg\min_{\phi} R^e(\mathcal{C} \circ \phi) & \forall e \in \mathcal{E} \\ \phi_L \in \arg\min_{\psi} D_{\text{KL}}(P^{e_i}(\psi(X)|Y) \| P^{e_j}(\psi(X)|Y)) & \forall e_i, e_j \in \mathcal{E} \end{cases}$$

(20)

where $R^e$ is the risk in environment $e$ and $D_{\mathrm{KL}}$ is the Kullback-Leibler divergence.

Further, the min-max formulation in Eqn.5 can be viewed as a form of Distributionally Robust Optimization (DRO) with an uncertainty set defined by the causal structure:

$$\min_{\phi_L, \phi_H, \mathcal{C}} \max_{P \in \mathcal{P}} \mathbb{E}_{(X,Y) \sim P}[\ell((\mathcal{C} \circ \phi_H \circ \phi_L)(X), Y)] \tag{21}$$

where $\mathcal{P}$ is the set of distributions consistent with the anti-causal structure. This connection implies that our approach inherits the worst-case performance guarantees of DRO while enforcing causal consistency through the regularization terms.

### F.4 Anti-Causal Toy Example vs. Kernels

We illustrate how kernels capture conditional probabilities with a simple structural causal model (SCM). See below:

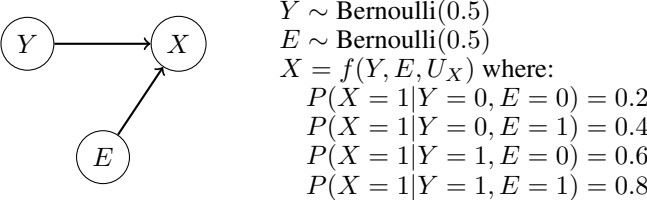

$$
\begin{aligned}
&Y \sim \text{Bernoulli}(0.5) \\
&E \sim \text{Bernoulli}(0.5) \\
&X = f(Y, E, U_X) \text{ where:} \\
&\quad P(X = 1 | Y = 0, E = 0) = 0.2 \\
&\quad P(X = 1 | Y = 0, E = 1) = 0.4 \\
&\quad P(X = 1 | Y = 1, E = 0) = 0.6 \\
&\quad P(X = 1 | Y = 1, E = 1) = 0.8
\end{aligned}
$$

Figure 10: Toy Anti-Causal SCM showing how $Y$ and $E$ determine $X$

In this model, $Y$ and $E$ both causally influence $X$. The environment $E$ shifts the probabilities, but the causal effect of $Y$ on $X$ remains consistent: $Y = 1$ always increases the probability of $X = 1$ by 0.4 compared to $Y = 0$.

This example illustrates a key property of anti-causal structures: intervening on $Y$ changes the distribution of $X$, while intervening on $E$ also changes $X$. Our kernels $K_S^{\mathcal{Z}_L}$ and $K_S^{\mathcal{Z}_H}$ capture these conditional probabilities. The role of ACIA is to extract the invariant relationship between $Y$ and $X$ (the consistent +0.4 effect) while removing the effect of environment $E$.

For the provided SCM where $Y \to X \leftarrow E$, the causal kernel $K_S(\omega, A)$ represents the conditional probability $P(X \in A | Y = y, E = e)$, where $\omega$ contains components $(y, e)$. For specific realizations:

$$
K_S(\omega, \{X = 1\}) = P(X = 1 | Y = y, E = e) = \begin{cases} 0.2 & \text{if } y = 0, e = 0 \\ 0.4 & \text{if } y = 0, e = 1 \\ 0.6 & \text{if } y = 1, e = 0 \\ 0.8 & \text{if } y = 1, e = 1 \end{cases}
$$

The product causal space combines environments by defining $\Omega = \Omega_{e_0} \times \Omega_{e_1}$ with $S \subseteq \{e_0, e_1\}$. For $S = \{e_0, e_1\}$, our kernel becomes:

$$
\begin{aligned}
K_{\{e_0, e_1\}}(\omega, \{X = 1\}) &= P(X = 1 | Y = y, E \in \{e_0, e_1\}) = \frac{P(X = 1, E \in \{e_0, e_1\} | Y = y)}{P(E \in \{e_0, e_1\} | Y = y)} \\
&= \frac{P(X = 1, E = e_0 | Y = y) + P(X = 1, E = e_1 | Y = y)}{P(E = e_0 | Y = y) + P(E = e_1 | Y = y)} \\
&= \frac{P(X = 1 | Y = y, E = e_0) P(E = e_0) + P(X = 1 | Y = y, E = e_1) P(E = e_1)}{P(E = e_0) + P(E = e_1)}
\end{aligned}
$$

For $Y = 0$ with equal environment probabilities, we get:

$$
K_{\{e_0, e_1\}}(\omega, \{X = 1\}) = \frac{0.2 \cdot 0.5 + 0.4 \cdot 0.5}{0.5 + 0.5} = 0.3
$$

The interventional kernel under perfect intervention $do(Y = 1)$ would be:

$$
K_S^{do(Y=1)}(\omega, \{X = 1\}) = \int_\Omega K_S(\omega', \{X = 1\}) \mathbb{Q}(d\omega' | \omega) = \begin{cases} 0.6 & \text{if } e = 0 \\ 0.8 & \text{if } e = 1 \end{cases}
$$

For imperfect interventions with strength $\alpha \in [0, 1]$:

$$K_S^{soft\_do(Y)}(\omega, \{X = 1\}) = (1 - \alpha) \cdot K_S(\omega, \{X = 1\}) + \alpha \cdot K_S^{do(Y)}(\omega, \{X = 1\})$$

Our low-level representation kernel $K_S^{\mathcal{Z}_L}$ integrates over the empirical distribution:

$$K_S^{\mathcal{Z}_L}(\omega, \{X = 1\}) = \int_\Omega K_S(\omega', \{X = 1\})d\mathbb{Q}(\omega') \approx \frac{1}{n}\sum_{i=1}^{n} K_S(\omega_i, \{X = 1\})$$

The high-level kernel $K_S^{\mathcal{Z}_H}$ integrates over low-level representations to abstract away environment-specific effects:

$$K_S^{\mathcal{Z}_H}(\omega, \{X = 1\}) = \int_{\mathcal{D}_{\mathcal{Z}_L}} K_S^{\mathcal{Z}_L}(\omega, \{X = 1\})d\mu(z) \approx P(X = 1|Y = y)$$
$$= 0.5 \cdot P(X = 1|Y = y, E = e_0) + 0.5 \cdot P(X = 1|Y = y, E = e_1)$$

In this way, ACIA captures the invariant relationship: $Y = 1$ consistently increases $P(X = 1)$ by 0.4 compared to $Y = 0$, regardless of environment.

### F.5  Anti-Causal Representation Learning, Robustness, and Privacy

Anti-causal representation learning focuses on modeling settings where labels generate features rather than the reverse. This perspective is particularly relevant in domains such as healthcare, biology, and natural sciences, where the underlying causal process is generative (e.g., diseases causing observable symptoms or molecular mechanisms producing data patterns). By explicitly capturing these anti-causal dynamics, we conjecture that ACIA gains not only better generalization across environments but also stronger foundations for robustness and privacy.

**Relationship to Adversarial Robustness:** Adversarial vulnerabilities often arise because conventional predictive models latch onto spurious correlations or environment-specific artifacts that are easily perturbed. Anti-causal representation learning mitigates this by forcing the model to encode how labels give rise to features. As a result, learned representations emphasize stable causal pathways instead of fragile correlations. For instance, in medical diagnose, a robust anti-causal representation prioritizes disease-related structures rather than scanner-specific noise or hospital-specific imaging protocols. This alignment reduces the attack surface for adversaries, since perturbations that exploit non-causal features become less influential. In this sense, adversarial robustness can be viewed as an emergent property of faithfully modeling anti-causal structure.

**Relationship to Privacy Protection:** Privacy risks (e.g., membership inference, data reconstruction) in machine learning models often arise when models overfit to environment- or individual-specific details. Anti-causal learning frameworks such as ACIA pursue high-level causal abstractions that discard environment-specific and identity-revealing details while preserving task-relevant causal signals. By abstracting away non-essential variations, anti-causal representations naturally suppress features tied to identity or context, thereby reducing information leakage. For instance, an anti-causal model for health prediction may learn "disease $\rightarrow$ symptom" relationships while ignoring hospital IDs or demographic quirks, which protects patient privacy.

**Relationship to Adversarially Robust and Privacy-Preserving Representation Learning:** Adversarially robust representation learning [77, 74, 72] seeks to construct representation spaces that remain stable under small but adversarially chosen perturbations, ensuring that the predicted labels remain unchanged across adversarial variants. A key limitation, however, is that such models may still rely on spurious correlations or environment-specific artifacts that are vulnerable to manipulation. In contrast, anti-causal representation learning explicitly models how labels generate features, thereby shifting learned representations toward invariant causal mechanisms rather than fragile correlations.

Privacy-preserving representation learning [60, 38, 4] aims to learn representations that prevent models from encoding or leaking sensitive, environment-specific, or identity-related attributes. These methods often employ regularization or information-theoretic constraints to balance utility and privacy. Anti-causal learning provides a principled complement: rather than obfuscating sensitive features post hoc, it avoids encoding them in the first place by aligning representations with label-to-feature generative mechanisms—retaining task-relevant causal features while discarding nuisance signals tied to individuals or contexts.

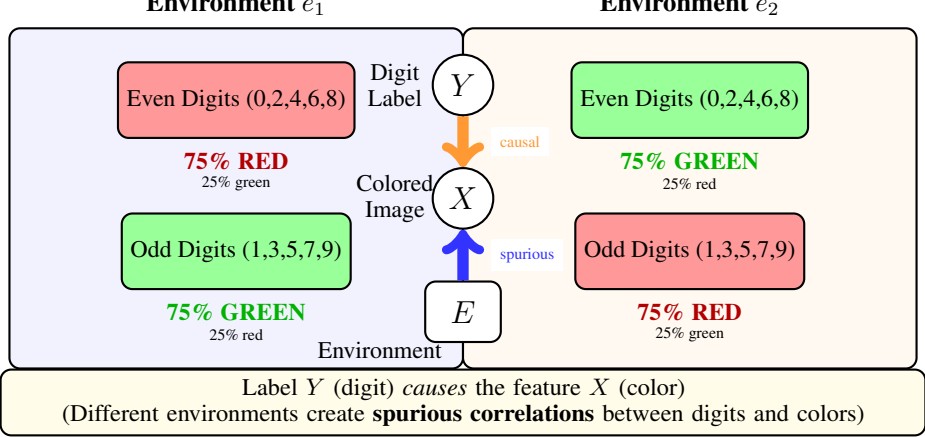

Figure 11: Anti-causal structure in Colored MNIST. The digit label $Y$ causally generates the observed colored image $X$, while environment $E$ introduces spurious digit-color correlations.

# G  A Concrete Example to Illustrate ACIA

In this section, we present an example step-by-step to illustrate how ACIA works in practice. We focus on a simplified version of the Colored MNIST (CMNIST) dataset to make the concepts accessible.

## G.1  Step 0 (Problem Setup): The Colored Digit Example

Imagine a scenario where we have images of handwritten digits that are colored either red or green. We have two environments with different digit-color correlations:

In causal learning, we predict causes from effects (e.g., predict disease from symptoms). But in anti-causal learning, we predict effects from causes (e.g., predict symptoms from disease). The digit label $Y$ (cause) generates the color feature in image $X$ (effect). The task is to predict $Y$ from $X$. The goal is to learn representations that capture the invariant causal relationship between $Y$ and $X$ while removing the spurious environment-specific correlations.

In environment $e_1$, even digits (0,2,4,6,8) are mostly colored red (75%), while odd digits (1,3,5,7,9) are mostly colored green (75%). In environment $e_2$, the correlation is reversed - even digits are mostly green (75%), while odd digits are mostly red (75%).

## G.2  Step 1: Define Causal Kernels for Each Environment

Causal kernels mathematically capture "how likely is a colored image given a digit label." They formalize the data generation process: $Y$ (digit) $\rightarrow$ color $\rightarrow$ $X$ (image).

First, we define causal kernels that capture the conditional probabilities in each environment. For environment $e_1$:

$$K_{e_1}(\omega, A) = \begin{cases} 0.75 & \text{if } A \text{ has red images and } Y \text{ is even} \\ 0.25 & \text{if } A \text{ has red images and } Y \text{ is odd} \\ 0.25 & \text{if } A \text{ has green images and } Y \text{ is even} \\ 0.75 & \text{if } A \text{ has green images and } Y \text{ is odd} \end{cases} \tag{22}$$

For environment $e_2$, the probabilities are reversed:

$$K_{e_2}(\omega, A) = \begin{cases} 0.25 & \text{if } A \text{ has red images and } Y \text{ is even} \\ 0.75 & \text{if } A \text{ has red images and } Y \text{ is odd} \\ 0.75 & \text{if } A \text{ has green images and } Y \text{ is even} \\ 0.25 & \text{if } A \text{ has green images and } Y \text{ is odd} \end{cases} \tag{23}$$

These kernels formalize how the probabilities of observable features (colors) depend on the label (digit) in each environment.

## G.3 Step 2: Construct the Low-Level Representation

While the causal kernels in Step 1 define the *theoretical* probabilities, we now need a *practical* neural network $\phi_L$ to learn these relationships from data. Our low-level representation $\phi_L$ captures raw relationships from input data. For each input image $x \in \mathcal{X}$, $\phi_L(x)$ extracts features that include both digit-related patterns and color information.

In the product causal space combining both environments, we integrate over the empirical distribution:

$$K_S^{\mathcal{Z}_L}(\omega, A) = \int_\Omega K_S(\omega', A)\, d\mathbb{Q}(\omega') \tag{24}$$

This integration averages the causal kernels across all observed samples, creating a low-level representation space that captures both digit shape information (invariant across environments), and color information (varies with environment). In practice, this integration is implemented as a weighted average over training samples:

$$K_S^{\mathcal{Z}_L}(\omega, A) \approx \frac{1}{|D_{e_1}|} \sum_{j=1}^{|D_{e_1}|} K_{e_1}(\omega_j, A) + \frac{1}{|D_{e_2}|} \sum_{j=1}^{|D_{e_2}|} K_{e_2}(\omega_j, A) \tag{25}$$

This gives us the low-level causal dynamics $\mathcal{Z}_L = \langle \mathcal{X}, \mathbb{Q}, \mathbb{K}_L = \{K_S^{\mathcal{Z}_L}(\omega, A) : S \in \mathscr{P}(T), A \in \mathscr{H}\}\rangle$ that captures both digit-specific and environment-specific information.

---

**Algorithm 1** Causal Dynamic: Construction of Low-Level Representations

---

1: **Input:** Dataset $\{(x_i, y_i, e_i)\}_{i=1}^n$ with images, labels, environments
2: **Output:** Low-level representation function $\phi_L$
3: Initialize convolutional neural network $\phi_L$ with parameters $\theta_L$
4: **for** each mini-batch $(x, y, e)$ **do**
5:   $z_L \leftarrow \phi_L(x; \theta_L)$                      ▷ Forward pass through low-level network
6:   Compute environment-conditioned distributions $P(z_L|y, e)$
7:   Compute causal kernel $K_S^{\mathcal{Z}_L}$ using:
8:   $K_S^{\mathcal{Z}_L}(\omega, A) \approx \frac{1}{|D_S|} \sum_{j \in D_S} P(z_L \in A|y_j, e_j)$
9:   Update $\theta_L$ using gradient descent
10: **end for**
11: **return** $\phi_L$

---

In practice, we extract a low-level representation $\phi_L(x)$ using a neural network as below:
$$\phi_L(x) = \text{Linear}(\text{Flatten}(\text{CNN}(x))) \tag{26}$$

## G.4 Step 3: Construct the High-Level Representation

The high-level representation $\phi_H$ builds on the low-level representation to extract only the invariant causal features. For our digit-color example, this means distilling information that is consistently related to the digit across environments, while discarding the spurious color-digit correlation.

The high-level kernel is defined by integrating over the domain of low-level representations:

$$K_S^{\mathcal{Z}_H}(\omega, A) = \int_{\mathcal{D}_{\mathcal{Z}_L}} K_S^{\mathcal{Z}_L}(\omega, A)\, d\mu(z) \tag{27}$$

This creates a causal abstraction $\mathcal{Z}_H = \langle \mathbf{V}_H, \mathbb{K}_H \rangle$ that focuses on the digit shape rather than its color. $\phi_H$ processes the low-level features to extract environment-invariant features: In practice, $\phi_H$ is implemented as a multi-layer perceptron:

$$\phi_H(z_L) = \text{MLP}(z_L) = W_2(\text{ReLU}(W_1 z_L + b_1)) + b_2 \tag{28}$$

The high-level kernel $K_S^{\mathcal{Z}_H}$ integrates over the domain of low-level representations to create a representation that is invariant to environment-specific features.

**Algorithm 2** Causal Abstraction: Construction of High-Level Representations

---

1: **Input:** Low-level representations $\{z_{L_i}\}_{i=1}^n$, labels $\{y_i\}_{i=1}^n$, environments $\{e_i\}_{i=1}^n$
2: **Output:** High-level representation function $\phi_H$
3: Initialize MLP network $\phi_H$ with parameters $\theta_H$
4: **for** each mini-batch $(z_L, y, e)$ **do**
5:    $z_H \leftarrow \phi_H(z_L; \theta_H)$                     ▷ Forward pass through high-level network
6:    **for** each label value $y_k$ **do**
7:       **for** environments $e_i, e_j$ where $i \neq j$ **do**
8:          Compute $z_{H,e_i,y_k} = \text{Avg}(\{z_H^{(l)} | y^{(l)} = y_k, e^{(l)} = e_i\})$
9:          Compute $z_{H,e_j,y_k} = \text{Avg}(\{z_H^{(l)} | y^{(l)} = y_k, e^{(l)} = e_j\})$
10:         Compute distance $d_{ij,k} = \|z_{H,e_i,y_k} - z_{H,e_j,y_k}\|_2$
11:       **end for**
12:    **end for**
13:    Update $\theta_H$ to minimize distances $d_{ij,k}$ (environment independence)
14: **end for**
15: **return** $\phi_H$

---

## G.5 Step 4: Joint Optimize on Loss and Regularizations

The complete optimization objective is:

$$\min_{\mathcal{C}, \phi_L, \phi_H} \max_{e_i \in \mathcal{E}} \left[ \mathbb{E}_{e_i}[\ell(f(X), Y)] + \lambda_1 R_1 + \lambda_2 R_2 \right] \tag{29}$$

where $f = \mathcal{C} \circ \phi_H \circ \phi_L$ is the complete predictive model.

$R_1$ **- Environment Independence Regularizer:** For each label $y_k$, we compute the mean high-level representation for each environment, then measure the distance between these means across environments:

$$R_1 = \sum_{y_k} \sum_{e_i \neq e_j} \|\mathbb{E}[z_H | y_k, e_i] - \mathbb{E}[z_H | y_k, e_j]\|_2 \tag{30}$$

Minimizing $R_1$ ensures that representations for the same digit are similar regardless of environment (color).

$R_2$ **- Causal Structure Consistency Regularizer:** We measure how well the high-level representation preserves the causal structure:

$$R_2 = \sum_{e_i} \|P(y | z_H, e_i) - P(y | \text{do}(z_H), e_i)\|_2 \tag{31}$$

This can be approximated by adding noise to the input (simulating intervention) and ensuring that the predicted output distribution remains consistent with the theoretical interventional distribution.

**Practical Implementation of $R_1$:** For each label $y_k$, we first compute the mean high-level representation for each environment, and then measure the distance between these means across environments. Minimizing this distance ensures that representations for the same digit are similar regardless of the environment.

**Practical Implementation of $R_2$:** We measure how well the high-level representation preserves the causal structure. In practice, this can be approximated by adding noise to the input (simulating intervention) and ensuring that the predicted output distribution remains consistent with the theoretical interventional distribution.

## G.6 Step 5 (Outcome): What ACIA Achieves?

Figure 12 illustrates the outcome visually. Given a new colored digit image $x$ from a potentially unseen environment, the low-level representation $\phi_L(x)$ extracts features related to both digit shape and color. The high-level representation $\phi_H(\phi_L(x))$ focuses only on features invariantly associated with the digit label, discarding the spurious color correlation. Finally, the classifier $\mathcal{C}$ predicts the digit based only on these invariant features.

**Algorithm 3** ACIA: Anti-Causal Representation Learning

---

1: **Input:** Dataset $\{(x_i, y_i, e_i)\}_{i=1}^n$, initialized $\phi_L$ and $\phi_H$
2: **Output:** Optimized $\phi_L$, $\phi_H$, and classifier $\mathcal{C}$
3: Initialize classifier $\mathcal{C}$ with parameters $\theta_C$
4: **for** each epoch **do**
5:    **for** each mini-batch $(x, y, e)$ **do**
6:       $z_L \leftarrow \phi_L(x)$                                 ▷ Low-level representations
7:       $z_H \leftarrow \phi_H(z_L)$                              ▷ High-level representations
8:       $\hat{y} \leftarrow \mathcal{C}(z_H)$                                       ▷ Predictions
9:       Compute classification loss $\mathcal{L}(\hat{y}, y)$
10:     Compute $R_1$ regularizer:                   ▷ Environment independence
11:       $R_1 = \sum_{y_k} \sum_{e_i \neq e_j} \|\mathbb{E}[z_H|y_k, e_i] - \mathbb{E}[z_H|y_k, e_j]\|_2$
12:     Compute $R_2$ regularizer:                   ▷ Causal structure consistency
13:       Estimate conditional distributions $P(y|z_H)$ and $P(y|\text{do}(z_H))$
14:       $R_2 = \sum_{e_i} \|P(y|z_H, e_i) - P(y|\text{do}(z_H), e_i)\|_2$
15:     Update all parameters using $\mathcal{L}_{total} = \mathcal{L}_{cls} + \lambda_1 R_1 + \lambda_2 R_2$
16:    **end for**
17: **end for**
18: **return** $\phi_L, \phi_H, \mathcal{C}$

---

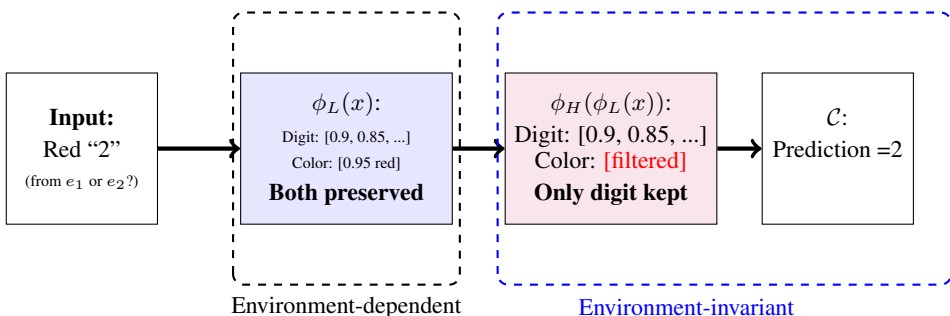

Figure 12: The outcome of ACIA with concrete values. The low-level representation captures both digit shape (0.9, 0.85) and color (0.95 for red), while the high-level representation filters out color and retains only digit features, enabling robust prediction regardless of environment.

## H   Impact Statement

**Societal Benefits.** Our work advances causality and machine learning by developing robust representations for anti-causal settings. Our experiments on real-world dataset Camelyon17 demonstrate potential for improving diagnostic tools that generalize across hospital systems. We recognize that in medical contexts, some hospital-specific variations may actually contain diagnostically relevant information. ACIA addresses this through its two-level representation architecture, where low-level features can preserve environment-specific details while high-level abstractions capture invariant causal relationships. The regularization parameters in our optimization problem in Equation 5 can be adjusted to balance between robustness and retention of clinically significant variations.

**Safety and Reliability.** ACIA's ability to disentangle causal mechanisms from spurious correlations has applications in safety-critical domains: autonomous vehicles must distinguish genuine hazards from environment-specific artifacts (lighting, weather), and climate models must separate fundamental physical processes from regional measurement biases. The regularization parameters in Eq. 5 allow practitioners to balance robustness with retention of contextually meaningful variations.

**Fairness Considerations.** While not our primary focus, ACIA's framework could help identify and mitigate dataset biases by distinguishing environment-specific associations from invariant causal relationships. For instance, if certain demographic groups cluster in specific environments, ACIA can separate spurious demographic correlations from genuine causal factors. However, we emphasize

that: (1) practitioners must validate that "spurious" features are truly non-causal in their domain, and (2) fairness requires additional constraints beyond environmental invariance.

**Limitations and Responsible Use.** ACIA assumes the anti-causal structure $Y \rightarrow X \leftarrow E$ is appropriate for the domain. Misapplication to causal settings ($X \rightarrow Y$) may discard informative features. In high-stakes applications, human domain experts should validate learned representations and intervention effects. We provide code and documentation to facilitate responsible deployment and reproducibility.

