# OpenReview forum: "Measure-Theoretic Anti-Causal Representation Learning"
_NeurIPS.cc/2025/Conference — NeurIPS 2025 poster_

### Official Review · Reviewer_qtK8 · 2025-07-01

**Clarity:** 2
**Significance:** 3
**Originality:** 3
**Rating:** 5
**Confidence:** 3

**Summary:**

This paper presents a measure-theoretic framework for anti-causal representation learning. The proposed framework handles both perfect and imperfect interventions. The proposed framework builds two levels of latent representations: a low-level "causal dynamic" that disengages the environment-specific variations and a high-level "causal abstraction" that extracts environment-invariant features.

**Questions:**

- How do we know the causal direction between the label and the observations in general?
- I'm a bit confused about the causal dynamic and causal abstraction part in the framework. What's the difference between "invariant across environments" and "environment-independent"? In l.137, if $P(\phi_L(\mathcal{X}|Y)$ is constant across environments $\mathcal{E}$, isn't that also means it is independent of $\mathcal{E}$ given $Y$?
- Could you explain more about how the causal-dynamic learns the low-level representations and disentangles the influence of the environment? How is it different from the "causal abstraction"? Also, in practice, for example, when using a neural network, how do you decide the part of high-level and low-level features?


broken reference in line 194

**Ethical Concerns:**

["NO or VERY MINOR ethics concerns only"]

**Final Justification:**

The paper proposed a novel approach with theoretically rigorous results. My concerns are adequately addressed during the rebuttal.

**Limitations:**

I can't find an explicit discussion about the limitations of the proposed work. Including a discussion on the practical scenario of anti-causal assumption might be helpful.

**Paper Formatting Concerns:**

No formatting concerns

**Quality:**

3

**Strengths And Weaknesses:**

# Strengths

- The paper proposed a novel approach based on a theoretically rigorous framework.
- The proposed framework, accommodating both soft and hard interventions, makes it more applicable and robust in real-world settings.
- The proposed method is evaluated through both synthetic and real-world experiments.

# Weaknesses

- Some part of the writing seems unclear. For example, the motivation for having two-level representations.
- It is not obvious when we can apply the proposed method by assuming the anti-causal relationship in the data.

---

> ### Author Rebuttal · Authors · 2025-07-31
>
> ###  W1: Motivation for having two-level representations.
>
> **Response:** Thanks for the comment!
>
> We first emphasize that **existing invariant or causal representation learning methods** typically focus on learning a **single-layer representation mapping** $\phi: X \rightarrow Z$. While effective in some settings, these approaches face several key challenges in the context of **anti-causal learning**, as outlined in our introduction:
>
> - They often assume **perfect interventions**, which are difficult or impractical to implement in real-world scenarios.
> - They typically require access to **explicit structural dependencies** (e.g., via Structural Causal Models), which are rarely available.
> - Many methods also assume **IID data** or require **knowledge of the test-time distribution**, limiting their applicability in realistic settings involving distribution shift.
>
>
> We propose to address all the challegnes through the two-level causal hierarchy framework.
> - **Low-level representation mapping** ($\phi_L: X \rightarrow Z_L$) extracts features directly from raw observations (without requiring explicit DAGs), capturing the **anti-causal structure** of the data, including both causal ($Y \rightarrow X$) and spurious ($E \rightarrow X$) patterns. The learned low-level representations support reasoning under both **perfect and imperfect interventions**, enabled by the introduced **interventional kernels**.
> - **High-level representation mapping** ($\phi_H: Z_L \rightarrow Z_H$) is applied on top of $\phi_L$. It serves as an information bottleneck that extracts **causally invariant features** — specifically, preserving the effect of $Y \rightarrow X$ while **discarding spurious environment-related signals** from $E \rightarrow X$.
> - Together, the two-level mappings cooperate to learn **robust anti-causal representations**, with $\phi_L$ capturing rich generative mechanisms and $\phi_H$ filtering out non-causal noise for improved generalization across environments.
>
>
>
>
> ###  Q1/W2: When to apply the proposed method/How to know anti-causal relationship in general.
>
> **Response:** An anti-causal relationship occurs *when the task label causes the observational features*. This structure is common in many real-world applications. For instance, we show some examples below:
>
> - *Medical diagnosis*: The presence of a disease causes specific symptoms or test results.
> - *Image classification*: The identity of an object determines its visual appearance.
> - *Quality control*: Defects in a product lead to anomalies in sensor measurements.
> - *Recommendation system*: Users' preferences drive their behavior patterns.
>
>
> Technically, we can identify anti-causal relationships through interventional tests: if intervening on features X does not change the label Y, but intervening on Y does affect X, then the relationship is anti-causal. Our **Corollary 2 (Interventional Kernel Invariance)** formalizes this intuition and provides a principled test using the interventional kernel we introduced.
>
>
>
>
> ### Q2: Difference between "Invariant Across Environments" and "Environment-Independent"
>
> **Response:** Thank you for the comment, and we apologize for the confusion.
>
> - **Invariant across environments** refers to **consistency under distribution shifts**. Specifically, it means that the learned representation remains **stable** across different environments/distributions.
> - **Environment-independent**, on the other hand, implies that the representation is **statistically independent of the environment variable**. That is, the features **do not contain any information** about the environment.
>
> Mapping this distinction to our two-layer representation learning setup:
>
> - The **low-level representation** $\phi_L$ learns features such that the distribution $\Pr(\phi_L(\mathcal{X}) \mid Y)$ is **invariant across environments**, but it may still encode **residual information** about the environment $E$.
> - The **high-level representation** $\phi_H$ then filters out this residual information to enforce **conditional environment-independence**, i.e., $\phi_H(\phi_L(\mathcal{X})) \perp E \mid Y$.
>
> **Illustrative example:** Two environments may exhibit identical $\Pr(\phi_L(\mathcal{X}) \mid Y)$, meaning the representations appear invariant across environments. However, the features $\phi_L(\mathcal{X})$ may still contain **environment-specific markers** (e.g., background texture or domain-specific artifacts). $\phi_H$ further removes these markers to achieve **true environment-independence**.
>
> We will revise the wording in the paper to better reflect this distinction.
>
>
> ### Q3: How does Causal Dynamic learn low-level representations? How is it different from Causal Abstraction? How are high-level and low-level feature parts determined in a neural network?
>
> **Response:** Thank you for the valuable question!
>
> Recall that the **low-level representation mapping** $\phi_L$ aims to capture the **anti-causal structure**, i.e., both $Y \rightarrow X$ and $E \rightarrow X$. The **Causal Dynamic** module facilitates this learning by optimizing a **supervised loss** in conjunction with the **$R_2$ regularizer**. Minimizing $R_2$ encourages the low-level representation to align with the true causal mechanisms underlying the data, promoting better generalization under intervention.
>
> Building on top of $\phi_L$, the **Causal Abstraction** module learns a **high-level representation mapping** $\phi_H$, guided by the **$R_1$ regularizer**. The goal is to further enforce **statistical independence from the environment variable** $E$. Specifically, $R_1$ measures the **discrepancy between the expected high-level representations across environments**, conditioned on the label $Y$, and minimizing it removes environment-specific information while retaining causal features.
>
> In practice, we design the architecture with the following principles:
> - The **low-level representation** should be **sufficiently rich** to capture both label-relevant and environment-dependent signals (i.e., $Y \rightarrow X$ and $E \rightarrow X$).
> - The **high-level representation** serves as an **information bottleneck**, filtering out environment-dependent variations, and thus is set to a **lower dimension** than the low-level representation.
>
> In our experiments with image data:
> - $\phi_L$ is implemented as a **CNN encoder**, producing a $d_L$-dimensional representation.
> - $\phi_H$ is a **small MLP bottleneck**, outputting a $d_H$-dimensional high-level representation, where $d_H < d_L$.
>
>
> ### L1: Discuss the limitations of the proposed work
>
> **Response:** Thanks for the suggestion! We outline the potential limitations below:
>
> - **Scalability**: Generalizing ACIA to large-scale, high-dimensional data may be computationally expensive and time-consuming.
> - **Optimal regularization selection**: The optimal hyperparameters $\lambda_1$ and  $\lambda_1$ in the objective function may dependent on the application domain. It requires careful tuning to obtain their optimal values.
> - **Complex causal structures**: ACIA is primarily designed for anti-causal settings. Its applicability to more complex causal structures—such as confounded-descendant or confounded-outcome [59] scenarios—remains unclear and warrants further investigation.

---

> > ### Author Response · Authors · 2025-08-07
> > **Comment by Authors**
> >
> > Dear Reviewer qtK8,
> >
> > Thank you again for your comments!
> >
> > We would like to kindly ask if our response has addressed your main concerns, or if there are any points you would like us to further clarify or elaborate on.
> >
> > Thanks,
> >
> > Authors

---

> > ### Comment · Reviewer_qtK8 · 2025-08-07
> >
> > Thanks for the detailed replies. I still do not fully understand the difference between "invariant across environments" and "environment-independent".
> >
> > You mentioned that the distribution invariant across environments means stable/constant across different environments, but may encode residual information about the environment. How is this represented in a mathematical definition? More specifically, if a variable E represents environments, does it mean $P(\phi_L(\mathcal{X})|Y, E) = P(\phi_L(\mathcal{X})|Y) $?
> >
> > My other questions about the anti-causal relationship and the causal abstraction have been well addressed.

---

> ### Comment · Area_Chair_SwRr · 2025-08-05
> **Please engage with discussion**
>
> Dear reviewer qtK8,
>
> I'd like to invite you to actively engage in the discussion with the authors. They provided a detailed rebuttal to your review, and it is important to know if this new information changes your mind.
>
> Best regards,
> AC

---

> ### Author Response · Authors · 2025-08-08
> **Response by Authors**
>
> Dear Reviewer qtK8,
>
> Thank you for the reply — we’re glad that our previous response has addressed all comments except for the distinction between **“Invariant across environments”** and **“Environment-independent.”**
>
> We apologize for the confusion, which we believe primarily stems from the imprecise use of the term *“invariant.”*
>
> As noted in our earlier rebuttal:
>
> > *“Low-level representation learns features that are invariant across environments, which refers to consistency/stability under distribution/environment shifts.”*
>
> This statement was **not** intended to imply that we achieve
> $$
> P(\Phi_L(X) \mid Y, E) = P(\Phi_L(X) \mid Y),
> $$ but rather that the learned conditional distribution $P(\Phi_L(X) \mid Y)$ is **stable with respect to environment shifts**, given that the label prior $P_e(Y)$ may vary across environments.
>
> In this context, we meant that the **low-level features may still encode residual information about the environment** (i.e., allow predicting $E$ from $\phi_L(X)$ when $Y$ is known), particularly due to the differences in $P_e(Y)$ across environments.
>
>
> Moreover, we emphasize that in the **anti-causal setting**, it is **impossible to learn a single-layer representation** $\Phi_L$ that satisfies  $P(\Phi_L(X) \mid Y, E) = P(\Phi_L(X) \mid Y)$, as demonstrated in Ref[59].
>
> This theoretical limitation further motivates our introduction of **high-level representation learning**, which aims to explicitly remove the influence of the environment. Or to say, forcing the **environment independence** in the high-level representation explicitly reflected by the conditional independence: $
> \phi_H(\phi_L(X)) \perp E \mid Y. $
>
> To prevent future confusion, we will revise the terminology and replace *“invariant”* with more precise phrases such as *“stable across environments”* or *“stable under environment shift.”*
>
> **We hope this clarification has addressed your last remaining concern—thank you again for your thoughtful feedback!**
>
> Authors

---

> > ### Comment · Reviewer_qtK8 · 2025-08-08
> >
> > Thanks for the detailed explanation and clarification! Now I understand it better, and my concerns and questions are adequately addressed.

---

> > > ### Author Response · Authors · 2025-08-09
> > > **Response by Authors**
> > >
> > > Dear Reviewer qtK8,
> > >
> > > We are glad that our response has addressed all final comment!
> > >
> > > Best,
> > >
> > > Authors

---

### Official Review · Reviewer_WVhG · 2025-07-02

**Clarity:** 2
**Significance:** 2
**Originality:** 3
**Rating:** 4
**Confidence:** 4

**Summary:**

This paper proposes a framework called ACIA for anti-causal representation, aiming at disentanglement between invariant causal features and environment variations. ACIA is a two-level representation framework, where the low-level representation captures the direct causal information from data and labels. And the high-level representation serves as an information bottleneck to filter environment-specific noise. The authors also prepare rather detailed experiments. Compared with different methods in four datasets, ACIA obtains several best results in several metrics including ACC, EI, LII, IR.

**Questions:**

1.As mentioned before, the experiment part is relatively sufficient, but the visualization only focus on the disentanglement of the representations using t-SNE, which is a very classical and common method to visualize representations. However, it can’t uncover the causal relationship between different features. Could you provide some visualization to uncover the correspondent relationship of the two-level representation learning?
2.Explicit DAG is abandoned during the optimization, which may weaken the causal properties and required some prior to complement. Could you provide a comparison of the prior information used in different methods? Experiments should be conducted at the same prior level to be fair.
3.ACIA captures intervention uncertainty through probability measures and avoid the assumption of perfect intervention. In addition to the experiment, can you provide some theoretical analysis of the intervention assumption?

**Ethical Concerns:**

["NO or VERY MINOR ethics concerns only"]

**Final Justification:**

I would like to give the borderline acc as the final justification. This paper proposes some interesting ideas of anti-causal representation learning, although some aspects can be further improved. I would not be upset if the final recommendation is acceptance, either.

**Limitations:**

The paper adopts an adversarial training approach to optimize the ACIA framework, but it does not explore alternative optimization strategies that could potentially achieve similar or better results. For instance, non-adversarial methods like variational inference or simpler gradient-based techniques might suffice, yet the paper lacks a comparative analysis to justify the exclusive reliance on adversarial training beyond its practical utility. This omission limits the understanding of the framework’s flexibility and robustness across different optimization paradigms.

**Paper Formatting Concerns:**

No formatting issues were found after checking against NeurIPS guidelines.

**Quality:**

3

**Strengths And Weaknesses:**

Strength:
1.The paper has thorough theoretical analysis of the performance of ACIA, e.g.,  convergence property, generalization bound in terms of sample complexity and interventional kernels in the anti-causal setting, and environmental robustness. These contribute to the scientific rigorousness of this paper and provide some reference meaning to peer work.
2.There are relatively sufficient experiments in this work including several classical datasets such as CMNIST and RMINST. It also obtains competitive results in more challenging scenario. From an experimental perspective, this work is relatively qualified.
Weakness:
1.Though the paper has long discussion about the ACIA, the two-level representation algorithm is still vague especially in the usage of the causal dynamic and its kernels. In appendix G, the pseudo code only briefly describes this crucial part which could be more detailed properly.
2.The paper proposes an adversarial method to train the model, it may be useful in practice. There are some theoretical analysis around the objective function in Section 3.2. But could you illustrate the necessity of the adversarial training method or it’s just a choice of methodology? Is there more analysis to bridge the gap between the theory and the training method?

---

> ### Author Rebuttal · Authors · 2025-07-31
>
> ### Q1: Visualize causal relationship between different features from the two-level representation learning.
>
> **Response:** Thank you for the valuable suggestion!
>
> To visualize the causal relationship between low-level and high-level features, we will employ **gradient-based attribution methods** (visualization via heatmap) to quantify how each dimension in the low-level representation contributes to each dimension in the high-level representation. For instance, on CMNIST, this approach will reveal how **low-level features**—such as digit **shape** and **color**—are transformed through the two-stage representation learning process. Specifically, we will generate **flow diagrams** illustrating:
> - Which low-level neurons are activated by shape vs. color,
> - And how only **shape-related dimensions** in low-level representations are used by high-level representations to form **causally invariant features** (i.e., digit identity).
>
> This visualization will help confirm that the high-level representation successfully filters out environment-specific information while preserving task-relevant signals.
>
>
>
> ### Q2: Explicit DAG is abandoned during the optimization. Comparison of the prior information used in different methods.
>
> **Response:** We highlight that **eliminating dependency on explicit DAG is actually one of the key advantages of ACIA**. This is due to the fact that obtaining the accurate and explicit DAG in real-world scenarios is often difficult and becomes our motivation.
>
> Please see below table the prior information of the compared SOTA causal representation learning methods.  Our experimental results in the paper demonstrate that, though requiring less information, ACIA still outperforms the compared SOTAs.
>
>
> | Method | Causal Structure           | SCM Knowledge | Intervention Type |
> |--------|----------------------------|---------------|-------------------|
> |ACTIR (NeurIPS'22) | Anti-causal Y -> X <- E | Variable roles only | Perfect only |
> | CausalDA (ICLR'23) | DAG structure              | Variable types | Perfect only |
> | LECI (NeurIPS'24) | Partial connectivity       | Variable relationships | Perfect only |
> | **ACIA** | **anti-causal Y -> X <- E** | **Variable roles only** | **Both perfect and imperfect** |
>
>
> **Note:**
> - **"Variable roles only"** refers to knowing the role of each variable in the causal model, i.e., identifying which variable is the **target** (Y), which are **observations** (X), and which represent **environmental factors** (E).
> - **"Variable types"** means knowing the **data type** of each variable, such as whether it is **binary**, **categorical**, or drawn from a **specific noise distribution**.
> - **"Variable relationship"** refers to knowing the **causal structure** (i.e., the **directionality** in the **causal DAG**) among the variables.
>
>
> ### Q3: Provide the theoretical analysis of the intervention assumption.
>
> **Response:** Perfect and imperfect intervention types are unified in the the introdued **interventional kernel framework**, stated below:
>
>
> Theorem 2 (Interventional Kernel): For any intervention $\mathbb{Q}: \mathscr{H} \times \Omega \rightarrow [0,1]$, there exists a unique interventional kernel:
>
> $K_S^{do(\mathcal{X}, \mathbb{Q})}(\omega, A) = \int_{\Omega} K_S(\omega, d\omega') \mathbb{Q}(A|\omega')$
>
> In *perfect interventions*, $\mathbb{Q}(A|\omega') = \mathbb{Q}(A)$, while $\mathbb{Q}(A|\omega')$ varies with $\omega'$ in *imperfect interventions*.
>
>
> Further, **Corollary 2** shows that **interventional kernel invariance** properties $K_S^{do(X)}(\omega, \{Y \in B\}) = K_S(\omega, \{Y \in B\})$ hold regardless of intervention perfectness.
>
>
> ### W1: More details on the two-level representation algorithm
>
> **Response:** Due to space constraints, we have provided the full details of the two-level representation algorithm in Algorithms 1–3 (Pages 44 and 55) in Appendix. Do these pseudocode descriptions help clarify the method?
>
>
>
> ### W2/L1: Nessisity of the min-max adversarial training approach.
>
> **Response:** ACIA adopts a *min-max* optimization framework to ensure **worst-case** robustness across environments. This formulation is theoretically grounded in our OOD generalization bounds (Theorem 7), which require guarantees on worst-case performance across diverse environments.
>
>
> We emphasize that existing (environment-)invariant representation learning methods—such as IRM, Rex, and VRex—are designed to optimize for **worst-case performance across environments**. This worst-case objective is critical: without it, the learned representations often fail to disentangle environmental factors effectively, as has been **extensively validated in prior literature**.

---

> > ### Comment · Reviewer_WVhG · 2025-08-05
> > **Responses to authors**
> >
> > Thank you for the detailed comments. Several key aspects like the visualization of the causal features are well addressed. I still think that explicit evaluations of the correctness of the causal direction, rather than a comprehensive evaluation like some socres (t-SNE  and others), which can only measure the goodness of fit concerning a DAG implicitly, are helpful, although it is clearly very hard, even impossible in some real cases (maybe some toy examples to show the discovery causal graph). I thus keep my score as it is.

---

> > > ### Author Response · Authors · 2025-08-05
> > > **Response by Authors**
> > >
> > > Dear Reviewer WVhG,
> > >
> > > Thank you for your response!
> > >
> > > Regarding the correctness of the causal direction, we would like to offer the following clarifications:
> > > 1. The datasets we use are intentionally **anti-causal** in nature. This is by design, following prior works such as [59], as detailed in Section E.1 of the Appendix.
> > >
> > > 2. Our evaluation metrics (e.g., *intervention robustness*) are specifically designed to characterize **anti-causal properties** in learned representations.
> > >
> > > 3. While we agree that anti-causal structures may be impossible to establish in certain applications, **they are present in many real-world domains** [21, 44]. Examples include:
> > >    - *Image classification*: The identity of an object determines its visual appearance.
> > >    - *Medical diagnosis*: The presence of a disease causes specific symptoms or test results.
> > >    - *Quality control*: Defects in a product lead to anomalies in sensor measurements.
> > >    - *Recommendation systems*: Users’ preferences drive their behavioral patterns.
> > >
> > >    Technically, anti-causal relationships can be identified through **interventional tests**: If intervening on features does **not** change the label, but intervening on label does change features, then the relationship is anti-causal. Our **Corollary 2 (Interventional Kernel Invariance)** formalizes this intuition and provides a principled test using the interventional kernel we introduced.
> > >
> > > **We remain open to further discussion on this challenging but important problem**!

---

> ### Comment · Area_Chair_SwRr · 2025-08-05
>
> Dear reviewer wvHG,
>
> I'd like to invite you to actively engage in the discussion with the authors. They provided a detailed rebuttal to your review, and it is important to know if this new information changes your mind.
>
> Best regards,
> AC

---

### Official Review · Reviewer_FVuS · 2025-07-02

**Clarity:** 3
**Significance:** 4
**Originality:** 3
**Rating:** 5
**Confidence:** 3

**Summary:**

The paper “Measure-Theoretic Anti-Causal Representation Learning” addresses the challenging anti-causal representation learning problem, and anti-causal invariant abstractions (ACIA) that can accommodate both perfect and imperfect interventions through interventional kernels. It contains two-level of representation learning, where the low-level capture the causal dynamics and high-level learn invariant predictor. Theoretical results are given to guarantee the OOD generalization. The experiments on synthetic and real-world datasets demonstrate the effectiveness of ACIA, also validate the theoretical results.

**Questions:**

Please refer to the weaknesses.

**Ethical Concerns:**

["NO or VERY MINOR ethics concerns only"]

**Final Justification:**

Concerns are well addressed, I have raised the significance score to 4

**Limitations:**

yes

**Quality:**

4

**Strengths And Weaknesses:**

**Strengths**

1. The problem is well-defined and the theoretical results are promising

2. A unified theoretical framework is given to model the anti-causal learning problem, the motivation of ACIA is clear and the objective of ACIA has a solid theoretical foundation

3. The theoretical results (e.g., convergence of ACIA, OOD generalization bound, etc.) further demonstrate the effectiveness of ACIA

3. The experiments validate the proposed theoretical results

**Weaknesses**

1. Both low-level and high-level learning processes encourage an environment-invariant representation, and its seems the goals of these two process overlap to some extent, what are the main differences between these two process? Are these two processes towards the same goal in different ways?

2. The description of Figure 2 is insufficient, and Figures 1 and 2 overlaps, it is better to merge them into one figure

3. Typos: Line 194 "Appendix ??"; Title of section 3.3, “objection” -> “objective”; Footnote in page 3 "P(Y|do(X=x)" -> "P(Y|do(X=x))"

---

> ### Author Rebuttal · Authors · 2025-07-31
>
> ###  Q1: Clarify the roles of the two-level learning
>
> **Response:** The two-level representation learning process in our framework serves **distinct roles within the causal hierarchy**. We provide the following clarification:
>
> - **Low-level representation mapping** ($\phi_L$) aims to extract features directly from raw observations (without requiring explicit DAGs), capturing the **anti-causal structure** of the data. This includes how both the label variable $Y$ and the environment variable $E$ influence the observation $X$. The learned features support reasoning under both **perfect and imperfect interventions**, enabled by the introduced **interventional kernels**.
> - **High-level representation mapping** ($\phi_H$) is applied on top of $\phi_L$, and its purpose is to extract **causally invariant features** — specifically, preserving the effect of $Y \rightarrow X$ while **discarding spurious environment-related signals** from $E \rightarrow X$. Formally, it enforces the condition: $\phi_H(\phi_L(\mathcal{X})) \perp E \mid Y$, ensuring that the final representation is **invariant to environment-specific variations**.
>
> Together, the two-level mappings cooperate to learn **robust anti-causal representations**, with $\phi_L$ capturing rich generative mechanisms and $\phi_H$ filtering out non-causal noise for improved generalization across environments.
>
>
>
> ### Q2: Figure 1 and Figure 2 mergerd into one
>
> **Response:** Thanks for the suggestion! We will merge the two figures!
>
> ### Q3: Typos: Line 194 "Appendix ??"; “objection” -> “objective”; "P(Y|do(X=x)" -> "P(Y|do(X=x))"
>
>  **Response:** Thanks for pointing them out and correction!

---

> > ### Author Response · Authors · 2025-08-07
> > **Comments by Authors**
> >
> > Dear Reviewer FVuS
> >
> > Thank you again for your comments!
> >
> > We would like to kindly ask if our response has addressed your main concerns, or if there are any points you would like us to further clarify or elaborate on.
> >
> > Thanks,
> >
> > Authors

---

> > > ### Comment · Reviewer_FVuS · 2025-08-09
> > >
> > > Thank you for your reply, my concerns are well addressed, I have raised the significance score to 4

---

> > > > ### Author Response · Authors · 2025-08-09
> > > > **Response by Authors**
> > > >
> > > > Dear Reviewer FVuS,
> > > >
> > > > We are glad that our response has addressed all your comments and thank you for raising the "significance score"!
> > > >
> > > > Best,
> > > >
> > > > Authors

---

### Official Review · Reviewer_xrNC · 2025-07-03

**Clarity:** 3
**Significance:** 3
**Originality:** 3
**Rating:** 5
**Confidence:** 4

**Summary:**

The authors propose ACIA, a measure-theoretic framework for anti-causal representation learning. ACIA aims to disentangle invariant causal features from environment-specific variations while making minimal assumptions about the underlying causal structure of the data-generating process. The main contributions of the paper include: A generalized intervention model that accommodates both perfect and imperfect interventions via interventional kernels; A construction of causal dynamics and causal abstractions to model invariant causal relations; A well-motivated optimization objective designed to identify stable causal features across domains/environments without requiring explicit DAG specifications.Theoretical guarantees for out-of-distribution generalization, including bounds on performance gaps between training and unseen environments. Experimental results on both synthetic and real-world datasets demonstrate ACIA’s ability to capture and leverage anti-causal structures, outperforming state-of-the-art methods.

**Questions:**

1. While the paper discusses theoretical complexity, could the authors also provide empirical training time or computational cost across different datasets to better assess practical scalability?

2. Line 94: The reference to Appendix ?? appears to be a placeholder. Could the authors clarify or correct the appendix citation?

**Ethical Concerns:**

["NO or VERY MINOR ethics concerns only"]

**Final Justification:**

I have read the authors' response and provided final feedback.

**Limitations:**

yes

**Quality:**

3

**Strengths And Weaknesses:**

Strengths

1. The paper is well written and logically structured, facilitating reader comprehension.

2. The authors provide rigorous and in-depth theoretical analysis to support their proposed method.

3. Comprehensive experimental evaluations validate the effectiveness of ACIA across diverse datasets.

Weaknesses

The review currently does not list specific weaknesses, but a minor area for improvement might be in the empirical clarity regarding computational cost (see question below).

---

> ### Author Rebuttal · Authors · 2025-07-31
>
> ### Q1/W1: Empirical training time on the evaluated datasets
>
> **Response:** The table below reports the runtime on a laptop with a single GPU (3.3 GHz, 32 GB RAM), using a desired precision $\epsilon=0.01$ and failure probability $\delta=0.05$. We observe that ACIA completes in approximately 3–16 minutes under these settings.
>
>
> | Dataset | \#Samples | Input Dimension| Time (seconds) |
> |---------|-------------|----------------|----------------|
> | CMNIST | 60K | 2352  | 580.7|
> | RMNIST | 60K | 784  | 973.6|
> | Ball Agent | 15K | 12288  | 204.7|
> | Camelyon17 | 84K | 6912 | 799.6|
>
>
>  ### Q2: Appendix Placeholder in Line 194
>
>  **Response:** Thanks for pointing out. It is Appendix C.5.

---

> > ### Comment · Reviewer_xrNC · 2025-08-03
> >
> > Thank you for the response. Your answers have addressed my questions.

---

> > > ### Author Response · Authors · 2025-08-03
> > > **Response by Authors**
> > >
> > > Dear Reviewer xrNC, we are glad that our response has addressed all your comments!

---

### Note · Authors · 2025-08-12

We sincerely thank the AC and all four reviewers for their time, effort, and constructive feedback.

**All reviewers initially provided positive scores**, acknowledging the importance of the studied problem, the solid theoretical contributions, and the empirical validations. This is reflected in the fact that all reviewers gave average scores of **Originality**, **Quality**, and **Significance** no lower than 3, reflecting their recognition of the work’s value.

Following the rebuttal, we have addressed all clarification comments, except for one **open and challenging question** raised--and acknowledged--by **Reviewer WVhG**, which we recognize as an important direction for future research.

---

### Decision · Program_Chairs · 2025-09-17

**Decision:**

Accept (poster)

**Comment:**

The authors study the problem of causal representation learning in the anti-causal setting where labels cause features. They propose a theoretically sound approach based on measure-theory that allows them to present a number of interesting results. Although there was some confusion about the problem and its relevance at first, these were alleviated during the rebuttal and discussion. All reviewers agree that the problem and solution are relevant and interesting. There remain minor concerns about the clarity and empirical evaluation, which I share. Nevertheless, I'm happy to recommend acceptance.